# Cloud height measurement by a network of all-sky-imagers

Niklas Benedikt Blum<sup>1,2</sup>, Bijan Nouri<sup>1</sup>, Stefan Wilbert<sup>1</sup>, Thomas Schmidt<sup>2</sup>, Ontje Lünsdorf<sup>2</sup>, Jonas Stührenberg<sup>2</sup>, Detlev Heinemann<sup>2</sup>, Andreas Kazantzidis<sup>3</sup>, and Robert Pitz-Paal<sup>4</sup> <sup>1</sup>Deutsches Zentrum für Luft- und Raumfahrt (DLR), Institut für Solarforschung, Paseo de Almería, 73, 2, E-04001 Almeria, Spain <sup>2</sup>Deutsches Zentrum für Luft- und Raumfahrt (DLR), Institut für Vernetzte Energiesysteme, Carl-von-Ossietzky-Straße 15, 26129 Oldenburg, Germany <sup>3</sup>Laboratory of Atmospheric Physics, Department of Physics, University of Patras, 26500 Patras, Greece <sup>4</sup>DLR, Institut für Solarforschung, Linder Höhe, 51147 Köln, Germany **Correspondence:** Niklas Blum (niklas.blum@dlr.de)

Abstract. Cloud base height (CBH) is an important parameter for many applications such as aviation, climatology or solar irradiance nowcasting (forecasting for the next seconds to hours ahead). The latter application is of increasing importance to operate distribution grids as well as photovoltaic power plants, energy storage systems and flexible consumers.

- To nowcast solar irradiance, systems based on all-sky-imagers (ASIs), cameras monitoring the entire sky dome above their 5 point of installation, have been demonstrated. Accurate knowledge of CBH is required to nowcast the spatial distribution of solar irradiance around the ASI's location at a resolution down to 5 m. Two ASIs located at a distance of usually less than 6 km can be combined into an ASI-pair to measure CBH. However, the accuracy of such systems is limited. We present and validate a method to measure CBH using a network of ASIs to enhance accuracy. To the best of our knowledge, this is the first method to measure CBH by a network of ASIs which is demonstrated experimentally.
- In this study, the deviations of 42 ASI-pairs are studied in comparison to a ceilometer and characterized by camera distance. The ASI-pairs are formed from seven ASIs and feature camera distances of 0.8...5.7 km. Each of the 21 tuples of two ASIs formed from seven ASIs yields two independent ASI-pairs as the ASI used as main and auxiliary camera respectively is swapped. Deviations found are compiled into conditional probabilities telling how probable it is to receive a certain reading of CBH from an ASI-pair given that true CBH takes on some specific value. Based on such statistical knowledge, in the inference
- 15 the likeliest actual CBH is estimated from the readings of all 42 ASI-pairs.

Based on the validation results, ASI-pairs with small camera distance (especially if 

In this study, seven of the ASIs included in the Eye2Sky ASI network (Schmidt et al., 2019; Blum et al., 2019a, b) are used. The selected ASIs are located in the city of Oldenburg. At the moment of writing, Eye2Sky contains 24 ASIs in Oldenburg and a region of about  $110 \text{ km} \times 100 \text{ km}$  to the west of Oldenburg. Eye2Sky is mainly dedicated to nowcasting of solar irradiance at high spatial and temporal resolution. The forecasting procedure, which will be described in more detail in a future publication,

- first recognizes clouds from the images of the ASIs. Cloud observations are then projected into a horizontal plane at the current CBH. These georeferenced cloud observations of multiple ASIs are merged and cloud properties are estimated. The angular velocities of clouds, as recognized by the individual ASIs, are transformed into absolute velocities over ground relying on an accurate estimation of CBH. Clouds are tracked along received cloud motion vectors to predict the clouds' future positions. Prior works studying ASI-based forecasting systems with up to four cameras (e.g. Nouri et al., 2019b) suggested that CBH
- is an essential component when predicting maps of solar irradiance based on cloud observations from ASIs, as the current and future positions of cloud shadows on the ground can only be predicted accurately if the clouds' height and velocity are determined accurately. Thus, in this publication an important component of this nowcasting system, namely the estimation of CBH, is presented. Our approach allows to use multiple ASI-pairs organized as ASI network and located in proximity, to estimate CBH. 42 ASI-pairs are formed from the seven ASIs and CBH is estimated by each ASI-pair based on the method
- presented by Nouri et al. (2019a). In a period of three months, the accuracy of the included ASI-pairs is evaluated for distinct conditions. Gained knowledge about the deviations of each ASI-pair is applied to merge the measurements of CBH from all 42 ASI-pairs into a more reliable measurement.

This publication is structured as follows. First, Eye2Sky, the ASI network used in the experiments, is introduced (Sect. 2). Then, the measurement procedure of CBH using the ASI network is presented (Sect. 3). Here, the properties of CBH measured

by reference ceilometer and by 42 ASI-pairs are discussed (Sect. 3.1). The meteorological conditions at the site are studied next (Sect. 3.2). In Sect. 3.4 and Sect. 3.3, a novel procedure to combine CBH measurements from multiple ASI-pairs of the ASI network is presented. Section 4 analyzes CBH measurement by the ASI network in comparison to the individual ASI-pairs for all relevant conditions. A summary of the presented findings closes the study in Sect. 5.

### 2 Eye2Sky network and experimental setup

- The so called Eye2Sky ASI network is being set up in the region of Oldenburg (Fig. 1, left). At its full extent, Eye2Sky will include 38 stations distributed over an area of roughly 110 km  $\times$  100 km equipped with ASIs. 13 of these stations will be supported by additional meteorological measurements to provide beam, diffuse and global irradiance via rotating shadowband irradiometers as well as ambient temperature and relative humidity. Eight ceilometers will be included in the network. Six of these are operated by the meteorological service Deutscher Wetterdienst (DWD). Five of these ceilometers are in the region
- viewed in Fig. 1. Several PV plants and numerous smaller distributed PV installations are also present in the study area. With its regional coverage, Eye2Sky aims to achieve nowcasts for individual PV installations from some minutes to multiple hours ahead. In the urban area of Oldenburg, the network will feature a high density of 14 ASIs in an area of  $13 \text{ km} \times 12 \text{ km}$ . This dense setup aims to provide ASI-based nowcasts of high accuracy across the urban area and reliable estimation of CBH under all conditions is an important contribution to achieve this scope.

Figure 1. Overview of the Eye2Sky ASI network including operational ASIs (ASI), radiometric measurements (Meteo) as well as planned stations (left) and ASIs in the city of Oldenburg included in this study (right). The ceilometer used as reference (marked by a red circle in the right figure) is located near the northwest-most ASI UOL. (background: © OpenStreetMap contributors 2020. Distributed under a Creative Commons BY-SA License.)

This work utilizes seven ASIs and one ceilometer located in the city of Oldenburg (Fig. 1, right). The ceilometer is located 133 m southeast of to the most northwestern ASI UOL. All included ASIs except for UOL are located east and south of the ceilometer. ASIs are placed at most 5.7 km from this ceilometer.

For this study, these ASIs are arranged into several ASI-pairs as defined by iteratively selecting a tuple of two ASIs out of the 21 tuples available and forming two independent ASI-pairs from each tuple by swapping its main camera. The main camera
of an ASI-pair is central to the measurement of CBH through an ASI-pair, described in more detail in Sect. 3.1, and defines the center of the area for which CBH is estimated. From 21 tuples of 2 ASIs, 42 ASI-pairs are received. All 42 ASI-pairs are included in the estimation procedure. The paired cameras' distance and the orientation of the ASI-pair's axis characterize the ASI-pairs. The orientation of an ASI-pair's axis is defined as seen from the main ASI and given in degree north. Figure 2 shows the distribution of orientations of ASI-pair's axes (left) and camera distances (right) in the set of available ASI-pairs. This set
covers almost all possible orientations of ASI-pair's axes. Available camera distances 0.8...5.7 km cover most of the range

0.02...5.5 km that is used in literature (Kuhn et al., 2019). Only towards small camera distances below 0.8 km, the present set lacks further ASI-pairs.