# Peer review of "Cloud height measurement by a network of all-sky-imagers"

_Atmospheric Measurement Techniques, 2020_

## Referee Comment (RC1) · Anonymous Referee #2 · 19 Jan 2021

The Authors present and evaluate an approach to derive cloud-base height (CBH) from a network of seven upward looking all-sky imagers (ASIs). The analysis focusses on a region in NW Germany during summer and shoulder seasons. The authors demonstrate that a network approach outperforms individual pairs of ASIs.

The manuscript is generally well-written, and the figures complement the main text appropriately. I recommend publication of this article after resolving several general and few minor comments.

General Comments

The Authors motivate their work as it allows to better nowcast downwelling solar fluxes (e.g., for photovoltaic power plants) and it is said that "accurate knowledge of CBH is

required". It is not perfectly obvious why better knowledge of CBH itself improves nowcasting. I'm assuming CBH is only one piece of information - apart from knowledge of each cloud's horizontal extend, cloud-top height, and geolocation (derived from satellite?) as well as the wind vector in cloudy altitudes (from meteorological forecasts or from ASIs?). Section 1 (ll. 26-32, ll. 48-53) touches on this topic but leaves open questions of how exactly this work fits into a larger picture. It is also unclear to me if voxel carving (ll. 58-59) is a competing approach or if this work could be used for voxel carving efforts – the Authors should clarify this in Section 1.

When using a network of ASIs over an area of (100km)2 to obtain a single CBH, do the Authors inherently assume a cloud (or a field of clouds) of unique base height? The Authors should make this more explicit (perhaps in Section 3) and discuss the realism of this assumption (perhaps in Section 4.4).

To obtain CBH probabilities (Section 3.3) the Authors use a subset of available data points. It is unclear what portion of the data was excluded. Did this selection mostly affect samples of high-altitude clouds? Perhaps the Authors could add a column to Table 1 that lists the fraction of data points excluded per altitude group?

The Authors measure accuracy of their approach by using a three-month dataset, shown in Fig. 9 and elaborated in Section 4.3. From a machine-learning standpoint is would be important to know if these were "training samples" (i.e., used to prepare CBH probabilities, etc.) or whether these data points were withheld from algorithm preparation.

The Authors introduce the Maximum Likelihood Estimation (MLE) approach in Section 3.4 and – before in Section 3.3 – provide information on conditional CBH probability. This arrangement seems confusing to me and recommend that Section 3.3 follows 3.4 (or is a subsection of 3.4).

Section 3.3 lists a variety of filters that were applied (ll. 240ff). The Authors should revise Section 3.3 and reference the use of these filters - if applied in the past – and

explain their intended effect.

The Authors list high temporal and spatial resolutions ("30 s or 5 m", l. 6) of state-of-the-art nowcasts. It is not obvious if chosen CBH intervals ("100m", l. 231) are fine enough to provide such high resolution. Perhaps the Authors could expand on this in Section 3.3 or in their discussion to address this question.

To help the reader appreciate the scientific advance in the work, the Authors should stress wherever (in Section 3.3 or 3.4) new techniques were developed or combined.

Minor Comments

Fig. 2: The plot seems to contain redundant information (by switching perspectives between two ASIs). The Authors could color code each perspective or exclude one redundant half.

ll. 140-144: Please provide the minimum optical thickness for ceilometer detection.

ll. 145-151: Is there a maximum solar zenith angle that limits CBH retrieval?

ll. 171-173: Please substitute "most dominant in features, driven by area and optical thickness" instead of "most dominant in the sense of area and optical thickness".

ll. 193-195: Please link to reference or plot(s) or else put "not shown".

Equ. 1: What is "j"?

Fig. 8: Please provide performance metrics (e.g., correlation coefficient, bias, and RMSD) to each panel.

---

## Referee Comment (RC2) · Anonymous Referee #3 · 1 May 2021

The manuscript presents an interesting use of a network of all-sky-imagers (ASIs) to derive mean cloud-base-height over a wide area. The method presented is interesting and, overall, the proposed system seems robust of probable practical use. The authors offer practical suggestions about the optimal layout of future ASIs installations, thus providing some useful information to the user. Reading the manuscript, it is clear that a lot of interesting work has been done, but unfortunately this has not been distilled enough yet to be clearly presented to the scientific community. The new algorithm is poorly presented, the novel contributions are not clearly identified, and the discussion of the results lacks focus. The authors should drastically revise the manuscript, trying to clearly present the essence and motivation of their work and separate it from implementation details.
To my understanding, there are three technical aspects presented: a) Implementation of three different approaches to calculate CBH from a pair of ASIs b) Evaluation of CBH retrievals from ASI-pairs. c) The use of a network of multiple ASI-pairs to derive a robust CBH estimate for the region.

Each of these aspects should be discussed and evaluated one by one, or references should be given in studies evaluating their performance. Otherwise, the reader cannot properly interpret the results.

Major comments:

Sections 3.3 and 3.4 should be rewritten. The sections seem like a direct translation of computer code into words, with no effort to describe why each step was implemented, what is essential, and what is just an implementation detail or even an experiment that happened to work. E.g. why use the three-gaussian filters? Why use the specific $\sigma$ thresholds? Why add an offset of 0.5 in low frequency bins (why not 0.01 or 1)? Implementation details could be even moved into an appendix.

A similar comment goes also for the discussion part: It should be made much more concise, focusing on key results. Moreover, the stated aim of the proposed method is to assist nowcasting, and thus the authors should add an evaluation of the single measurement accuracy of the network. I.e. if the network outputs a CBH value of h, what is the uncertainty of this estimate? It is good that the network shows small overall biases in a three month period, but it wouldn't be of much use if the correct CBHs were measured at the wrong times.

Specific comments:

Line 85: Is the 3-month period enough to monitor all available conditions? What would be a suggestion to other users about the range of conditions that needs to be captured for good training?

Line 108: "by arbitrary selecting a tuple of ASIs". From the text, you seem to be

selecting all possible combinations of ASIs not only some arbitrary pairs. Moreover, I am not sure if tuple is the propre name as, in my mind, a tuple could include more than 2 objects. Consider rephrasing.

Line 111: "Camera axis". Does this refer to the line connecting the two ASIs that form a pair? If yes, then it should be called "pair-axis" or similar. "Camera axis" sound to me as the name for the direction that a single camera is looking.

Line 116: For completeness, please provide some more information about the instrument: E.g. Is the instrument part of DWD network you mentioned before? How is the CBH calculated from the data? Are you using the manufacturer's algorithm or a custom one? What is the minimum overlap height? What is the minimum height that CBH can be detected? References?

Line 116: Are you using the color or B&W version of Q25?

Line 130: Please mention what is the total time required to get a processed image (including data transfer and processing)?

Line 141: *optically* thick clouds.

Line 141-144: How exactly do you distinguish if the first cloud layer is thick, to exclude the other detected cloud layers? Do you always keep only the first layer when multiple layers are detected?

Line 143-144: The accuracy discussion is not enough for an instrument sued as reference. The differences reported in Martucci et al. 2010 seem to be coming from different algorithm or even definition of CBH used by each instrument. Moreover, the bias they find is not only 160 meters, but also has a range component (Y=0.925X + 160). Finally, Martucci et al used a rather old model of the instrument you are using here. Therefore, you should give more details about the CBH algorithm used with Ceilometer data and discuss the possible differences in definition of CBH as used for ceilometer and for ASIs. Lines 161 - 176: The description of the algorithm is not very clear. Please add a

new figure (or add a panel in Fig.) showing the image of the second ASI, highlighting the matched window. Also, a small flowchart could be helpful.

Lines 161 – 176: Have you compared the results from the three method (center box, side boxes, full image) to validate your expectation that they yield similar results?

Lines 161 – 176: Please provide the relations connecting a) the ASI-pair distance with b) the minimum altitude that each method can be applied, due to purely geometric considerations.

Line 186: Specify that this analysis is based on the ceilometer. Is the CBH analysis based only on the lowest layer detected by the ceilometer?

Line 200: How are TanDEM-X data used in this study? This seems the wrong place of the manuscript to introduce a new dataset.

Line 187-206: How is this analysis of CBH stability relevant to this study? Does your algorithm work only in these conditions? Maybe the stability excludes some possible errors in transition periods? Please mention the context and usefulness of this part of the manuscript.

Line 228: ". . . , where N is the number of vertical bins used for the analysis" or similar.

Line 303: Why use theta for true CBH and not a symbol based on h?

Lines 350-354: The uniformity constraint is very reasonable during algorithm training, not so during evaluation! It is very interesting to evaluate the algorithm in variable cases and understand what the outputs are, if it is biased towards the low or high clouds etc.

Line 360: Why not reverse the two plots in Figure 6, to discuss them in order?

Line 377: As shown from the two pairs, in cloud-free conditions some ASI-pairs output the value of 12km, while others 2km (probably due to local low clouds). Why do you suggest that the 4km output of the network is a reasonable prediction of a layer coming at least 30 minutes later? Is this layer captured by any pair in the network? It could

also be a lucky combinations of these two extreme values? In general, how does the network handle cloud-free conditions?

Line 395: The main ASI-based CBH retrieval limits the instrument to a maximum zenith angle of 67 degrees. For the CLO-FLE pair, given the 4.2km distance of the instruments, the minimum detectable clouds should be around 1.4 km (if I calculate correctly). In the September cases many clouds are below this limit, so probably the second or third sub-algorithm was used (using e.g. the complete FOV of the camera). Could this be the reason of the overestimation? If yes, does the full-FOV retrieval add anything to the estimate or could just be skipped?

Line 405: What I understand from the plot is that the low clouds are detected by the ceilometer and not by the ASI-pair, not the other way around. If this is true, the ceilometer site should have persistent low not present over the ASI-pair. Is this reasonable from the local meteorological conditions? What seems more reasonable is that ASI-pair cannot detect low clouds, e.g. due to geometric and algorithm considerations. Please provide more details.

Line 408-410: This doesn't sound very surprising since the minimum altitude where your ASI have overlapping images at 67deg FOV should be around 1.7km. Please discuss such issues, preferable in a previous section, before presenting the results.

Line 417: "in the dataset used for modelling" ?

Line 418-422: The text is not well written, and it is not clear what you mean. Please rephrase.

Technical comments:

Lines 42-54: As written now, the paragraph starts as if to present ceilometers but ends up presenting various CBH estimation techniques and ends up with ASI-based forecasting requirements. A slight editing is needed to make the text clearer.

Line 71: Better use "Most ASI-based monitoring systems. . ." or similar.

Line 202: "For example, Tabernas, ..."

Line 355: "Then, the coincidence, ..." . The sentence needs rewording.

---

## Author Comment (AC2) · 1 Jun 2021

**0.0.1   Reviewer:**

The manuscript presents an interesting use of a network of all-sky-imagers (ASIs) to derive mean cloud-base-height over a wide area. The method presented is interesting and, overall, the proposed system seems robust of probable practical use. The authors offer practical suggestions about the optimal layout of future ASIs installations, thus providing some useful information to the user. Reading the manuscript, it is clear that a lot of interesting work has been done, but unfortunately this has not been distilled enough yet to be clearly presented to the scientific community. The new algorithm is

poorly presented, the novel contributions are not clearly identified, and the discussion of the results lacks focus. The authors should drastically revise the manuscript, trying to clearly present the essence and motivation of their work and separate it from implementation details. To my understanding, there are three technical aspects presented: a) Implementation of three different approaches to calculate CBH from a pair of ASIs b) Evaluation of CBH retrievals from ASI-pairs. c) The use of a network of multiple ASI-pairs to derive a robust CBH estimate for the region. Each of these aspects should be discussed and evaluated one by one, or references should be given in studies evaluating their performance. Otherwise, the reader cannot properly interpret the results.

0.0.2 Authors' response:

We really appreciate the reviewer's time and effort spent on reviewing our manuscript, their insightful comments and suggestions. We have addressed all comments and incorporated the suggestions as good as possible to us and we believe, these changes led to valuable improvements of our manuscript. In particular, we have strongly revised the sections on modelling and validation.

In the following, we will address the reviewer's comments point-by-point. Changes to the manuscript are extracted from the adapted manuscript within which changes were highlighted using latexdiff. Blue indicates insertions, red indicates deletions. Please note, that the order of Sect. 3.3 and Sect. 3.4 has been reversed as suggested by Reviewer Comment 1, General Comment 5. This change has been excluded from the markup, as it would have obscured all other changes.

**We recommend viewing the PDF version of this response which contains the changes in the manuscript. See bottom of this document. It was not possible to include all changes as figures, here.**

0.0.3   Changes in manuscript:

See below.

**1   Major comments**

**1.1   Major comment**

**1.1.1   Reviewer:**

Sections 3.3 and 3.4 should be rewritten. The sections seem like a direct translation of computer code into words, with no effort to describe why each step was implemented, what is essential, and what is just an implementation detail or even an experiment that happened to work. E.g. why use the three-gaussian filters? Why use the specific $\sigma$ thresholds? Why add an offset of 0.5 in low frequency bins (why not 0.01 or 1)? Implementation details could be even moved into an appendix.

**1.1.2   Authors' response:**

Based on the reviewer's feedback we revised Sect. 3.3 and 3.4 drastically. Especially regarding the modelling of conditional probability distributions, we moved the exact description of the procedure to the appendix and focused more on describing the idea behind the procedure and every filter. We further gave a reason for the value assigned to each of the parameters. However, we also stated that these parameters may still be optimized in a future work and are so far only rough approximations. Similarly, we reworked Sect. 3.3. In particular, we focused on pointing out for each step of the procedure, what the intention of each equation/ calculation step was.

**1.1.3 Changes in manuscript:**

Sections 3.3 and 3.4: pp. 12-20,

Appendix A, pp. 35-37.

**1.2 Major comment**

**1.2.1 Reviewer:**

A similar comment goes also for the discussion part: It should be made much more concise, focusing on key results. Moreover, the stated aim of the proposed method is to assist nowcasting, and thus the authors should add an evaluation of the single measurement accuracy of the network. I.e. if the network outputs a CBH value of h, what is the uncertainty of this estimate? It is good that the network shows small overall biases in a three month period, but it wouldn't be of much use if the correct CBHs were measured at the wrong times.

**1.2.2 Authors' response:**

We reworked the validation part strongly, intending most of all to focus on the key results. Still, we also needed to add passages at some points as further discussions or clarifications were suggested by the reviewers. As suggested, we included an additional subsection which evaluates the accuracy of an ASI-pair and of the ASI network for the nowcasting application as suggested above. We agree with the reviewer that this is an interesting aspect, which attests a certain advantage of the ASI network for this application. In particular, we made the following larger changes to the discussion part (Sect. 4): The behavior of the ASI network during mostly clear periods has been

detailed and described more precisely. The discussion of exemplary time series of CBH has been limited to a single day. Descriptions and visualizations of 06 August 2019 have been moved to Appendix B. To give the reader a faster overview of the results (and also based on Reviewer Comment 1, Minor Comment 7) scatter-density plots have been enhanced to include performance metrics and quantiles Discussion of minimum CBH has been condensed and has been placed in Sect. 4.2.1 also relating it to the expectation from geometry. Sect 4.3, evaluating accuracy in a nowcasting context, was added Based on Reviewer Comment 1, General comment 3, we described which portion of the validation data set was filtered out Based on Reviewer Comment 1, General comment 7, we discussed the relationship between the accuracy of CBH and the resolution of irradiance maps created by a nowcasting procedure. We merged the prior "Sect. 4.4" with the present Sect. 4.4, intending to shorten discussions as far as possible. We hope to have found a reasonable trade-off regarding the length of this section. As another measure to shorten the discussion part, Sect. 4.1, which analyzes time series of CBH from the different sources, could still be moved to the Appendix. This section is majorly intended to give the reader concrete examples of the effects discussed thereafter by statistical tools.

**1.2.3 Changes in manuscript:**

pp. 20-34,

Appendix B, pp. 38-39, ll. 943-953.

**2 Specific comments**

**2.1 Specific comment**

**2.1.1 Reviewer:**

Line 85: Is the 3-month period enough to monitor all available conditions? What would be a suggestion to other users about the range of conditions that needs to be captured for good training?

**2.1.2 Authors' response:**

One idea of our method was that it should not be necessary to train the model based on a dataset which represents the conditions during the operation or validation. The method should work best if the ASI-pairs exhibit the same behavior at a given reference CBH during model development and validation. I.e. distributions of conditional probabilities at a given reference CBH should be comparable for both data sets. Apart from that the data sets used for modeling and validation are both considered to be qualitatively representative of the months which are of greatest interest to solar applications at the studied latitude, as they may provide the greatest energy yield, based on sky conditions and sun elevation.

**2.1.3 Changes in manuscript:**

p. 17, ll. 412-418.
**2.2 Specific comment**

**2.2.1 Reviewer:**

Line 108: "by arbitrary selecting a tuple of ASIs". From the text, you seem to be selecting all possible combinations of ASIs not only some arbitrary pairs. Moreover, I am not sure if tuple is the proper name as, in my mind, a tuple could include more than2 objects. Consider rephrasing.

**2.2.2 Authors' response:**

The description we used was misleading and should be understood as indicated by the reviewer. We replaced the term by "iteratively". Further we now pointed out in the same paragraph, that all 42 ASI-pairs are considered for the estimation procedure. Indeed, the term tuple is not precise. Throughout the text, tuples are intended to have only two members. We replaced the term in general by "tuple of 2 ASIs".

**2.2.3 Changes in manuscript:**

p.1 l.10,

p. 5 ll. 127-131,

p. 6, caption of Fig. 6,

p. 17 l. 419.
**2.3  Specific comment**

**2.3.1  Reviewer:**

Line 111: "Camera axis". Does this refer to the line connecting the two ASIs that form a pair? If yes, then it should be called "pair-axis" or similar. "Camera axis" sound to me as the name for the direction that a single camera is looking.

**2.3.2  Authors' response:**

Indeed, the term is ambiguous and was adapted now to "ASI-pair's axis". The nomenclature was originally motivated by the one used by Kuhn, P., B. Nouri, S. Wilbert, N. Hanrieder, C. Prahl, L. Ramirez, L. Zarzalejo, T. Schmidt, Z. Yasser, D. Heinemann, P. Tzoumanikas, A. Kazantzidis, J. Kleissl, P. Blanc and R. Pitz-Paal (2019). "Determination of the optimal camera distance for cloud height measurements with two all-sky imagers." Solar Energy 179: 74-88.

**2.3.3  Changes in manuscript:**

pp. 5-6 ll. 132-135, caption Fig. 2,

p. 33 ll. 798, 800, 803, 805.

**2.4 Specific comment**

**2.4.1 Reviewer:**

Line 116: For completeness, please provide some more information about the instrument: E.g. Is the instrument part of DWD network you mentioned before? How is the CBH calculated from the data? Are you using the manufacturer's algorithm or a custom one? What is the minimum overlap height? What is the minimum height that CBH can be detected? References?

**2.4.2 Authors' response:**

We added a short description about the used instrument. It is operated by DLR since 2018. The manufacturer's algorithm is used with the default configuration. The algorithm is outlined in the instrument's manual. The firmware version is v0.747. A prior study stated that full overlap is given at a CBH of 1500 m and above. Based on an overlap correction, the manufacturer allows to set a minimum CBH down to 0 m. We use the default setting of 45 m. We also contacted the manufacturer for further information on the used algorithms in the meantime but did not receive a response, yet. If required, this can be handed in at a later time.

**2.4.3 Changes in manuscript:**

p. 6, ll. 137-141.

2.5   Specific comment

2.5.1   Reviewer:

Line 116: Are you using the color or B&W version of Q25?

2.5.2   Authors' response:

We use the daylight version of Mobotix Q25 6MP. This is the RGB/color version. We now also attached a reference to the instrument's specification.

2.5.3   Changes in manuscript:

p. 6, l. 142.

2.6   Specific comment

2.6.1   Reviewer:

Line 130: Please mention what is the total time required to get a processed image (including data transfer and processing)?

2.6.2   Authors' response:

We now specified the overall time required for image acquisition, transfer and processing, as suggested. Note that a further addition was made here based on Reviewer Comment 1, Minor comment 3.

**2.6.3 Changes in manuscript:**

p. 7, ll. 156-158.

**2.7 Specific comment**

**2.7.1 Reviewer:**

Line 141: *optically* thick clouds.

**2.7.2 Authors' response:**

As suggested, we now specified this more accurately.

**2.7.3 Changes in manuscript:**

p. 7, l. 171.

**2.8 Specific comment**

**2.8.1 Reviewer:**

Line 141-144: How exactly do you distinguish if the first cloud layer is thick, to exclude the other detected cloud layers? Do you always keep only the first layer when multiple layers are detected?

2.8.2   Authors' response:

This is the case. We now specified this more clearly.

2.8.3   Changes in manuscript:

p. 7, l. 172-173.

2.9   Specific comment

2.9.1   Reviewer:

Line 143-144: The accuracy discussion is not enough for an instrument sued as reference. The differences reported in Martucci et al. 2010 seem to be coming from different algorithm or even definition of CBH used by each instrument. Moreover, the bias they find is not only 160 meters, but also has a range component (Y=0.925X + 160). Finally, Martucci et al used a rather old model of the instrument you are using here. Therefore, you should give more details about the CBH algorithm used with Ceilometer data and discuss the possible differences in definition of CBH as used for ceilometer and for ASIs.

2.9.2   Authors' response:

Indeed, there have been several updates to the firmware after 2010. Some of these indicate changes to the algorithm of CBH measurement. We now summarized the results of two more recent studies which evaluated the CBH measurement by the ceilometer type used here. Based on these authors' findings, we also provided differences in the

algorithms used by the manufacturers. Further, we explained that prior validations of the method used in this study to measure CBH by the ASI-pairs, were performed by an instrument of the same type. This may avoid inconsistencies when comparing the results of the present study to those prior ones.

**2.9.3   Changes in manuscript:**

pp. 7-9, ll. 174-203,

pp. 9, l. 210.

**2.10   Specific comment**

**2.10.1   Reviewer:**

Lines 161 - 176: The description of the algorithm is not very clear. Please add a new figure (or add a panel in Fig.) showing the image of the second ASI, highlighting the matched window. Also, a small flowchart could be helpful.

**2.10.2   Authors' response:**

As suggested by the reviewer, we added another row to Fig. 3 showing the raw and processed image simultaneously recorded by ASI FLE. For a flow chart of the method we would like to refer to Nouri, B., P. Kuhn, S. Wilbert, N. Hanrieder, C. Prahl, L. Zarzalejo, A. Kazantzidis, P. Blanc and R. Pitz-Paal (2019). "Cloud height and tracking accuracy of three all sky imager systems for individual clouds." Solar Energy 177: 213-228. We revised the description of the algorithm and hope that it is clearer now. Further, we aimed to point out clearer that the CBH measurement of the ASI-pairs is only modified

very slightly over the one described and validated in the publication given above. We further provided validation results of that study.

**2.10.3 Changes in manuscript:**

p. 8, Fig. 3,

pp. 9-10, ll. 204-251.

**2.11 Specific comment**

**2.11.1 Reviewer:**

Lines 161 – 176: Have you compared the results from the three method (center box, side boxes, full image) to validate your expectation that they yield similar results?

**2.11.2 Authors' response:**

Unfortunately, we did not validate these sub-algorithms separately. As described in our response to Specific Comment 9 we now pointed out clearer that the method to estimate CBH, used by the ASI-pairs, is only modified very slightly over the publication which introduced this method and implementation: Nouri, B., P. Kuhn, S. Wilbert, N. Hanrieder, C. Prahl, L. Zarzalejo, A. Kazantzidis, P. Blanc and R. Pitz-Paal (2019). "Cloud height and tracking accuracy of three all sky imager systems for individual clouds." Solar Energy 177: 213-228. Therefore, we would like to refer to this study for further validation results. As part of a future study, it would be interesting to investigate the characteristics of these sub-algorithms. In our expectation, CBH will be measured more accurately by matches which are detected at small zenith angles.

Further, CBH measurement received for this central image area, which is used here, were also in the focus of the validation carried out in the publication named above, due to the cloud conditions at that site and due to the positions chosen for ASIs and ceilometer.

**2.11.3 Changes in manuscript:**

p. 10, ll. 243-248,

p. 11, l. 266-269.

**2.12 Specific comment**

**2.12.1 Reviewer:**

Lines 161 – 176: Please provide the relations connecting a) the ASI-pair distance with b) the minimum altitude that each method can be applied, due to purely geometric considerations.

**2.12.2 Authors' response:**

We calculated the expected minimum CBH which can be detected relying on the central window as well as when relying on all of the nine windows inside the cropped image of the main ASI. We additionally calculated the minimum CBH which is achieved if matches only succeed if matched windows cover zenith angles not larger than $67°$. We further stated that the third iteration of the matching procedure in which the ASI image is evaluated up to a zenith angle of $77.8°$ is not expected to reduce minimum CBH as this step matches a very large windows.

**2.12.3 Changes in manuscript:**

p. 10, ll. 236-242,

p. 10-11, ll. 252-265.

**2.13 Specific comment**

**2.13.1 Reviewer:**

Line 186: Specify that this analysis is based on the ceilometer. Is the CBH analysis based only on the lowest layer detected by the ceilometer?

**2.13.2 Authors' response:**

We added the statement as suggested. As in the complete study, we carried out this analysis only based on the lowest recognized cloud layer.

**2.13.3 Changes in manuscript:**

p. 12, ll. 284-285.

**2.14   Specific comment**

**2.14.1   Reviewer:**

Line 200: How are TanDEM-X data used in this study? This seems the wrong place of the manuscript to introduce a new dataset.

**2.14.2   Authors' response:**

TanDEM-X data are needed by the nowcasting system to create irradiance maps. For this study the data set is only relevant for this estimation of the maximum elevation of the topography. We rephrased as shown below.

**2.14.3   Changes in manuscript:**

p. 12, ll. 299-300.

**2.15   Specific comment**

**2.15.1   Reviewer:**

Line 187-206: How is this analysis of CBH stability relevant to this study? Does your algorithm work only in these conditions? Maybe the stability excludes some possible errors in transition periods? Please mention the context and usefulness of this part of the manuscript.

**2.15.2 Authors' response:**

The meteorological conditions described in this paragraph motivated the development of a method which aims to estimate CBH of the most dominant cloud layer more accurately. We added a conclusion to this paragraph which puts the analysis into the context of this study. Further as suggested by Reviewer Comment 1, General comment 2, we outlined at this point the scope of the method to estimate CBH and how it may be enhanced in the future.

**2.15.3 Changes in manuscript:**

p. 12, ll. 305-310.

**2.16 Specific comment**

**2.16.1 Reviewer:**

Line 228: "..., where N is the number of vertical bins used for the analysis" or similar.

**2.16.2 Authors' response:**

We adapted this statement as suggested.

**2.16.3 Changes in manuscript:**

Appendix A, p. 35, l.886.

**2.17 Specific comment**

**2.17.1 Reviewer:**

Line 303: Why use theta for true CBH and not a symbol based on h?

**2.17.2 Authors' response:**

We adapted the nomenclature as suggested, as it may be clearer (replacing $\theta, \theta, \hat{\theta}_{likeliest}, \hat{\theta}_{refined}$ by $h_{true}, \hat{h}_{true}, h_{likeliest}, h_{refined}$). Theta was used as this symbol may be used frequently with maximum likelihood estimation for the true/ estimated parameter.

**2.17.3 Changes in manuscript:**

p. 14-15, ll. 337-350,

pp.15-16, ll. 364-401.

**2.18 Specific comment**

**2.18.1 Reviewer:**

Lines 350-354: The uniformity constraint is very reasonable during algorithm training, not so during evaluation! It is very interesting to evaluate the algorithm in variable cases and understand what the outputs are, if it is biased towards the low or high clouds etc.

2.18.2   Authors' response:

First of all, we would like to apologize as a statement in the manuscript was misleading. The filter excluding variable situations is applied in the modelling of conditional probabilities (now Sect. 3.4) and in Sect. 4.4 to compare performance metrics from ASI-pairs and ASI network. In Sect. 4.1-4.3 this filter is not applied. We now corrected this statement and moved it from Sect. 4.1 to Sect. 4.4. The scatter-density plots shown in Sect. 4.2 may provide insights regarding effects occurring in variable cases. Based on Reviewer Comment 1, minor comment 7 we also added performance metrics to these plots. To enable the reader to evaluate the performance of the ASI-based estimation of CBH under these conditions (e.g. concerning biases) more quickly, we also added percentiles to all scatter-density plots.

2.18.3   Changes in manuscript:

pp. 20-21, ll. 531-535,

p. 28-29, ll. 691-695,

p. 24, Fig. 8.

2.19   Specific comment

2.19.1   Reviewer:

Line 360: Why not reverse the two plots in Figure 6, to discuss them in order?

2.19.2   Authors' response:

We appreciate the suggestion. However, as suggested by Major Comment 2, to shorten Sect. 4.1, this figure and related descriptions have been moved to Appendix B.

2.19.3   Changes in manuscript:

p. 21, Fig. 6,

p. 21, ll. 544-545,

p. 22, Fig. 7,

p. 22, l. 553,

p. 23, ll. 572-581,

p. 23, ll. 588-589,

Appendix B, pp. 38-39, ll. 943-953.

2.20   Specific comment

2.20.1   Reviewer:

Line 377: As shown from the two pairs, in cloud-free conditions some ASI-pairs output the value of 12km, while others 2km (probably due to local low clouds). Why do you suggest that the 4km output of the network is a reasonable prediction of a layer coming at least 30 minutes later? Is this layer captured by any pair in the network? It could also be a lucky combination of these two extreme values? In general, how does the

network handle cloud-free conditions?

2.20.2 Authors' response:

Our description may not have been precise in this point. We now added a plot (Fig. B1) in the appendix which shows the measurements of CBH from the ASI-pairs and from the ASI network as well as from the ceilometer during this clear period in more detail. We also added a short passage in Sect. 4.1 to describe closer which period we referred to. From Fig. B1 it is visible that the ASI-pairs measure a broad range of values between the extreme values of 2 km and 12 km, before around 17:00. Most ASI-pairs measure an intermediate CBH. After 17:00 the spread between the measurements of the ASI-pairs reduces. From around 17:05 the ASI network and some of the ASI-pairs measure a CBH of around 3 km. This CBH (3.1 km) is later also measured by the ceilometer. During this period the approaching cloud layer may be detected before its arrival in the urban area. During very clear periods, the ASI network is likely to return a CBH which is very large, in the range of 10 km. For an application this is not problematic, in our opinion, because another image processing step is used which is able to detect the absence of clouds. We added a short explanation on this.

2.20.3 Changes in manuscript:

p. 22, ll. 564-571,

Appendix B, p. 38, ll. 940-942, Fig B1.
**2.21 Specific comment**

**2.21.1 Reviewer:**

Line 395: The main ASI-based CBH retrieval limits the instrument to a maximum zenith angle of 67 degrees. For the CLO-FLE pair, given the 4.2km distance of the instruments, the minimum detectable clouds should be around 1.4 km (if I calculate correctly). In the September cases many clouds are below this limit, so probably the second or third sub-algorithm was used (using e.g. the complete FOV of the camera). Could this be the reason of the overestimation? If yes, does the full-FOV retrieval add anything to the estimate or could just be skipped?

**2.21.2 Authors' response:**

We share the reviewer's opinion, that the behavior seen for CLO-FLE in situations with CBH much smaller than 2 km is connected to the minimum CBH which this system can detect. This minimum CBH may indeed be determined by the sub-algorithm relying on the main ASI's cropped orthogonal image. The usage of the full FOV to retrieve CBH is not expected to improve an ASI-pairs capability to detect very low clouds noticeably. We now pointed out in Sect. 3.1, that this sub-algorithm is mainly intended to increase the robustness of the method. It may yield a valid measurement in some cases when the first sub-algorithms failed. We condensed the discussion of the minimum CBH in Sect. 4.2 as shown below. See also our response to Specific Comment 22.

**2.21.3 Changes in manuscript:**

p. 10, ll. 236-242,

p. 10-11, ll. 252-265,

p. 25 ll. 614-632.

**2.22 Specific comment**

**2.22.1 Reviewer:**

Line 405: What I understand from the plot is that the low clouds are detected by the ceilometer and not by the ASI-pair, not the other way around. If this is true, the ceilometer site should have persistent low not present over the ASI-pair. Is this reasonable from the local meteorological conditions? What seems more reasonable is that ASI-pair cannot detect low clouds, e.g. due to geometric and algorithm considerations. Please provide more details.

**2.22.2 Authors' response:**

We assume that the reviewer refers to the areas on the far left of the scatter-density plots (e.g. reference CBH < 0.5 km for DON-MAR and reference CBH < 2 km for UOL-HOL) and we agree with the analysis of the reviewer. At this point, we intended to discuss another area of these plots and now indicated these areas more precisely in the manuscript. When reference CBH ranges around 3...12 km, the ASI-based systems frequently detect low clouds close to the 5-percentile line, i.e. far below the main diagonal of the plot. In these cases, the ASI-based systems provide a CBH which is too small. As described in previous sections, we expect that in these cases the ASI-based systems recognize low clouds present in their field of view. At the same time there might be a gap in the low cloud layer at the location of the ceilometer. Therefore, the ceilometer may recognize a larger CBH.

**2.22.3 Changes in manuscript:**

pp. 24-25, ll. 605-610, Fig. 8.

**2.23 Specific comment**

**2.23.1 Reviewer:**

Line 408-410: This doesn't sound very surprising since the minimum altitude where your ASI have overlapping images at 67deg FOV should be around 1.7km. Please discuss such issues, preferable in a previous section, before presenting the results.

**2.23.2 Authors' response:**

As suggested by Specific Comment 11, we now calculated the minimum CBH which may be related to the sub-algorithms in Sect. 3.1. We reworked the discussion of minimum CBH in Sect. 4.2. and adapted it to refer to these values of minimum CBH expected from geometry.

**2.23.3 Changes in manuscript:**

p. 10, ll. 236-242,

p. 10-11, ll. 252-265,

p. 25 ll. 614-632.

2.24   Specific comment

2.24.1   Reviewer:

Line 417: "in the dataset used for modelling"?

2.24.2   Authors' response:

As we understand the comment, it is not clear at this point why the "dataset used modelling" is discussed in this context.  As also suggested by the following Special Comment 24, we rephrased this passage. We hope this makes the intended statement clearer, also in this perspective.

2.24.3   Changes in manuscript:

p. 26, ll. 642-650.

2.25   Specific comment

2.25.1   Reviewer:

Line 418-422: The text is not well written, and it is not clear what you mean.  Please rephrase.

2.25.2   Authors' response:

We rephrased the passage as shown below.

2.25.3   Changes in manuscript:

p. 26, ll. 642-650.

**3   Technical comments**

**3.1   Technical comment**

**3.1.1   Reviewer:**

Lines 42-54: As written now, the paragraph starts as if to present ceilometers but ends up presenting various CBH estimation techniques and ends up with ASI-based forecasting requirements. A slight editing is needed to make the text clearer.

**3.1.2   Authors' response:**

We rewrote this paragraph in part to put a stronger focus on possible sources of CBH to be considered for nowcasting. We moved this specification of nowcasts up.

**3.1.3   Changes in manuscript:**

p. 2, ll. 34-35,

p. 2-3, ll. 45-58.

**3.2 Technical comment**

**3.2.1 Reviewer:**

Line 71: Better use "Most ASI-based monitoring systems..." or similar.

**3.2.2 Authors' response:**

We adapted ASI system to ASI-based nowcasting system

**3.2.3 Changes in manuscript:**

p. 3, l. 79.

**3.3 Technical comment**

**3.3.1 Reviewer:**

Line 202: "For example, Tabernas,..."

**3.3.2 Authors' response:**

We inserted accordingly.
**3.3.3 Changes in manuscript:**

p. 12, l. 302.

**3.4 Technical comment**

**3.4.1 Reviewer:**

Line 355: "Then, the coincidence,...". The sentence needs rewording.

**3.4.2 Authors' response:**

We reformulated as shown below.

**3.4.3 Changes in manuscript:**

p. 21, ll. 537-539.

Please also note the supplement to this comment:
https://amt.copernicus.org/preprints/amt-2020-430/amt-2020-430-AC2-supplement.pdf

**Supplement:**

**Authors' response to Reviewer Comment 2**

**Reviewer:**

The manuscript presents an interesting use of a network of all-sky-imagers (ASIs) to derive mean cloud-base-height over a wide area. The method presented is interesting and, overall, the proposed system seems robust of probable practical use. The authors offer practical suggestions about the optimal layout of future ASIs installations, thus providing some useful information to the user. Reading the manuscript, it is clear that a lot of interesting work has been done, but unfortunately this has not been distilled enough yet to be clearly presented to the scientific community. The new algorithm is poorly presented, the novel contributions are not clearly identified, and the discussion of the results lacks focus. The authors should drastically revise the manuscript, trying to clearly present the essence and motivation of their work and separate it from implementation details.

To my understanding, there are three technical aspects presented: a) Implementation of three different approaches to calculate CBH from a pair of ASIs b) Evaluation of CBH retrievals from ASI-pairs. c) The use of a network of multiple ASI-pairs to derive a robust CBH estimate for the region.

Each of these aspects should be discussed and evaluated one by one, or references should be given in studies evaluating their performance. Otherwise, the reader cannot properly interpret the results.

**Authors' response:**

We really appreciate the reviewer's time and effort spent on reviewing our manuscript, their insightful comments and suggestions. We have addressed all comments and incorporated the suggestions as good as possible to us and we believe, these changes led to valuable improvements of our manuscript. In particular, we have strongly revised the sections on modelling and validation.

In the following, we will address the reviewer's comments point-by-point. Changes to the manuscript are extracted from the adapted manuscript within which changes were highlighted using latexdiff. Blue indicates insertions, red indicates deletions. Please note, that the order of Sect. 3.3 and Sect. 3.4 has been reversed as suggested by Reviewer Comment 1, General Comment 5. This change has been excluded from the markup, as it would have obscured all other changes.

Changes in manuscript:

See below.

**Major comment 1**

**Reviewer:**

Sections 3.3 and 3.4 should be rewritten. The sections seem like a direct translation of computer code into words, with no effort to describe why each step was implemented, what is essential, and

what is just an implementation detail or even an experiment that happened to work. E.g. why use the three-gaussian filters? Why use the specific  $\sigma$  thresholds? Why add an offset of 0.5 in low frequency bins (why not 0.01 or 1)? Implementation details could be even moved into an appendix.

**Authors' response:**

Based on the reviewer's feedback we revised Sect. 3.3 and 3.4 drastically. Especially regarding the modelling of conditional probability distributions, we moved the exact description of the procedure to the appendix and focused more on describing the idea behind the procedure and every filter. We further gave a reason for the value assigned to each of the parameters. However, we also stated that these parameters may still be optimized in a future work and are so far only rough approximations.

Similarly, we reworked Sect. 3.3. In particular, we focused on pointing out for each step of the procedure, what the intention of each equation/ calculation step was.

Changes in manuscript:

Sections 3.3 and 3.4: pp. 12-20

3.3 Estimating CBH in the ASI network (ORDER OF SECTIONS 3.3 AND 3.4 WAS EXCHANGED)

The estimation procedure presented here is motivated by In this section we present our method to combine the measurements of CBH from a large number ASI-pairs organized as network. Prior works estimated CBH by a small number of two or in some cases four ASIs (Nouri et al., 2019a). However, with a large number of ASI-pairs, we consider a statistical method

315 promising, which analyzes the CBH samples received and, based on the known characteristics of each ASI-pair, determines

Figure 5. Inference procedure — Step 1: For each range i of camera distance CBH1 is computed as mean CBH from the respective ASI-pairs. Conditional probability is evaluated that CBH1 would be received if true CBH (at the ceilometer) took on a value  $\{0...0.1, 0.1...0.2, ..., 11.9...12\}$  km (red boxes). Step 1 yields a likelihood function for each range of camera distance. Step 2: Cumulative and complementary cumulative likelihood are calculated for each range of camera distance. Step 3: These functions are logarithmized and then summed over all ranges i of camera distance yielding overall cumulative and complementary cumulative likelihood. Step 4: The Intersection of both functions gives the estimated likeliest CBH.

the CBH which is most likely to be present. The characteristics of each ASI-pair are in the following described by conditional probability distributions, which will by retrieved in Sect. 3.4. These distributions provide the probability of receiving a certain CBH reading from an ASI-pair, given that actually a specific reference CBH is present. Our estimation procedure then uses principles from Maximum Likelihood Estimation (MLE) - Figure and modifies them for the specific case. To the best of our

320 knowledge, the usage of a statistical method and in particular one relying on conditional probability distributions is novel to the task of estimating CBH from the observations of a multitude of ASIs.

To give an overview, Fig. 5 shows the inference process used to estimate CBH by the network based on the 42 CBH readings provided by the individual ASI-pairs. For each range *i* of camera distance, conditional probabilities estimated in Sect. 3.4, conditional probability distributions will be estimated. These conditional probabilities are translated into the likelihood that

- 325 actually certain values of (reference) CBH are present (step 1) based on the readings of CBH received for from ASI-pairs in this range *i* of camera distance. After calculating the cumulative likelihood for each range of camera distance (step 2), these are combined yielding the overall cumulative and complementary cumulative likelihood from all ASIs ASI-pairs (step 3). Finally, the value of CBH which is most likely to be present at the site and at the evaluated time, given the readings from all involved ASI-pairs, is estimated (step 4). These steps are explained-presented in more detail in the following.
- 330 Step 1: For each ASI-pair, the median value of all valid CBH readings of the previous 10 min is calculated. If an ASI-pair does not provide any valid CBH within this period, it is excluded from the prediction for the instance in time evaluated. The ranges of camera distance 1...2.5 km and 3...4 km are represented by a larger number of ASI-pairs than the remaining distances. To-Thus, the readings of ASI-pairs in these ranges of camera distance may prevail in the estimation of CBH. As the variety of camera distances is considered to bring a benefit to the procedure, we intend to represent all camera distances as uniformly
- as possible. For this, we define ranges of camera distance are defined, using the range limits {0.5, 1, 1.5, ..., 6} kmand. CBH readings of all ASI-pairs with camera distance in range *i* are averaged to yield CBHi. Consecutively, the conditional probability  $P(CBH_i | \theta) P(CBH_i | h_{true})$  is evaluated that the found CBHi would be received for a given true CBH  $\theta h_{true}$  (red marked box prior to step 1 in Fig. 5). Note that  $P(CBH_i | \theta)$  was  $P(CBH_i | h_{true})$  will be modeled in Sect. 3.4 measuring CBH  $h_{Ref}$ by a ceilometer which provided  $h_{Ref} \approx \theta$  provides  $h_{Ref} \approx h_{true}$ . Thus, the likelihood  $\mathcal{L}_i(\theta) \mathcal{L}_i(h_{true})$  is obtained (Fig. 5, 240 event of step 1):
- 340 output of step 1):

$$\mathcal{L}_i(\underline{\theta}h_{true}) = P(\text{CBH}_i \mid \underline{\theta}h_{true}).$$

(1)

Step 2: Likelihood is We define cumulative likelihood  $C_i(\hat{h}_{true})$  as the likelihood of receiving the present reading CBHi given that  $h_{true}$  is smaller or equal to an estimation of true CBH  $\hat{h}_{true}$ . Accordingly in the implementation, likelihood is summed cumulatively over all bins of reference CBH  $\theta$  to define cumulative likelihood  $h_{true}$  (Fig. 5, step 2):

 $345 \quad \mathcal{C}_{i}(\hat{h}_{true}) = \sum_{\underline{\theta \leq \hat{\theta}} h_{true} \leq \hat{h}_{true}} \mathcal{L}_{i}(\underline{\theta} h_{true}).$

(2)

Likewise, a complementary cumulative likelihood is defined

$$\bar{\mathcal{C}}_i(\hat{\theta}) = \sum_{\theta > \hat{\theta}} \mathcal{L}_i(\theta).$$

as the likelihood of receiving the present reading CBHt given that  $h_{true}$  is greater than an estimation of true CBH  $\dot{h}_{true}$ :

$$\bar{C}_i(\hat{h}_{true}) = \sum_{\substack{h_{true} > \hat{h}_{true}}} \mathcal{L}_i(h_{true}).$$

(3)

- 350  $C_{t}(\hat{\theta})$  and  $C_{t}(\hat{\theta})$  are used here as measures how likely it is that actual CBH $\theta$  is in the interval  $]0 \text{ km}, \hat{\theta}]$  or  $]\hat{\theta}, 12 \text{ km}]$  respectively. It is mainly the In particular, the use of these cumulative functions that and the estimation of likelihood functions from measurement data distinguishes the present approach from a regular Maximum-Likelihood-Estimation (MLE). This modification is used as in MLE typically smooth analytical tikelihood functions are assumed as likelihood function. In contrast, likelihood functions here are will be estimated based on empirical conditional probabilities. These approximated likelihood-
- 355 functions, derived from a dataset of finite size, may therefore be less smooth and may not be completely representable. representative. When using cumulative distributions, it is expected that the method still works robustly if the conditional probabilities are not estimated accurately for each grid cell of the discrete distribution if at least the cumulative value over a range of CBH is appropriate. In spite of the modification, the presented approach may adopt beneficial properties of MLE: The use of appropriate conditional probabilities (described determined in Sect. 3.4) reduces systematic deviations of estimated
- CBH compared to the measurement of a single ASI-pair. Moreover, applied conditional probabilities are in general not specific to the studied site and its meteorological conditions which allows to apply the method at other sites. When using cumulative distributions, it is expected that the method still works robustly if the conditional probabilities are not estimated accurately for each joint frequency grid cell but at least the cumulative value over a range of CBH is appropriate. Both functions C1(ĥ) and C1(ĥ) true) and C1(ĥ) true) are shown for three exemplary intervals of camera distance in Fig. 5 as output
- 365 of step 2.

Step 3: The natural logarithm is then applied to  $C_i(\hat{\theta})$  and summed over all. We aim to determine the likelihood of receiving the combination of readings CBHi from all the intervals *i* of camera distance to yield the given that  $h_{true} \leq \hat{h}_{true}$ . This can be expressed as product of  $C_i(\hat{h}_{true})$  from all intervals *i*. As this product would often become zero in our numerical treatment, we instead calculate its natural logarithm, which we refer to as overall logarithmized cumulative likelihood log  $C_n(\hat{h}_{true})$ . This

370 operation also allows to replace the product by a sum (Fig. 5, step 3)given the readings CBH, per interval *i* of camera distance

$$\log C_n(\hat{h}_{\underline{true}}) = \sum_i \log C_i(\hat{h}_{\underline{true}}).$$
(4)

Analogously, an overall complementary logarithmized cumulative likelihood is computed given all readings  $CBH_i$  per interval i of camera distance

$$\quad \log \bar{\mathcal{C}}_n(\hat{h}_{\underline{true}}) = \sum_i \log \bar{\mathcal{C}}_i(\hat{h}_{\underline{true}}). \tag{5}$$

Both functions are visualized exemplarily as output of step 3 in Fig. 5. In theory, the method could do without the application of a logarithm to  $C_1$  and  $\overline{C_1}$  in Eq. and Eq. respectively. In that case, the sum would be replaced by a multiplication in the respective equations. However, this would induce numerical problems regularly as handled products approach zero.

Step 4: The left hand sides in Eq. and Eq.  $\log C_n(\hat{h}_{true})$  and  $\log \bar{C}_n(\hat{h}_{true})$  are only known at discrete points. Linear interpolation yields continuous representations of these. An estimation of the likeliest actual CBH  $\theta_{tikeliest}$  is selected for which  $\log \bar{C}_n(\hat{\theta})$  and  $\log C_n(\hat{\theta})$ . Then finally, we aim to select the true CBH  $h_{tikeliest}$ , which makes it likeliest to receive the given combination of CBHi. In our formulation of the problem, this means we intend to find a  $\hat{h}_{tikeliest}$  which simultaneously maximizes  $\log C_n(\hat{h}_{true})$  and  $\log \bar{C}_n(\hat{h}_{true})$ . Consequently, we accept  $h_{tikeliest}$ , for which  $\log C_n(\hat{h}_{true})$  and  $\log \bar{C}_n(\hat{h}_{true})$ are equal (Fig. 5, step 4):

385
$$\underline{\theta}\underline{h}_{likeliest} = \operatorname{argmin}_{\hat{\theta}\hat{h}_{true}} \left| \log \bar{C}_n(\hat{h}_{true}) - \log C_n(\hat{h}_{true}) \right|.$$
(6)

Besides this estimation of CBH, a version of this procedure will be discussed that includes further refinements (in the following referred to as *refined* estimation). The refinement is motivated by the finding that some As a first observation from the generation of conditional probabilities, ASI-pairs are already accurate if actually a certain range of CBH is present as we will discuss in Sect. 4. First, the procedure presented above is modified to exclude ASI-pairs with camera distance greater

- 390 than 4.5 km as these ASI-pairs cause large deviations for CBH < 4 km and only provide a limited benefit exhibit only a moderate advantage at greater CBH. Results from this procedure are accepted as refined estimation  $\theta_{refined}$  if estimated CBH is within 3...12 km. Otherwise, the arithmetic average of CBH measured by These ASI-pairs are excluded from the refined estimation of  $h_{likeliest}$ . On the other hand, ASI-pairs with specific camera distance is used. The most appropriate small camera distance are already accurate if only small CBH occur, as we will discuss in Sect. 4. We inspected conditional probabilities
- 395 of the ASI-pairs for an interval of CBH are identified by an inspection of the conditional probabilities (exemplarily viewed as input to step 1 in Fig. 5). This and identified the ASI-pairs which are most appropriate for an interval of CBH. Based on this, the refined estimation is restricted to remain within the specific interval of CBH from the unrefined estimation in which it is applied received from the arithmetic average of CBH measured by ASI-pairs with corresponding small camera distance, if the first iteration of *h*likelicst yielded a sufficiently small CBH. In summary, the refinement procedure to receive the final estimation of CBH <del>front reads</del>.

|                                             | $h_{likeliest},$                                                          | $h_{likeliest} \in ]3, 12] \text{ km}$                                                                                  |     |
|---------------------------------------------|---------------------------------------------------------------------------|-------------------------------------------------------------------------------------------------------------------------|-----|
| $\underline{\theta}\underline{h}_{refined}$ | $\min(3 \text{ km}, \max(h_{i \in \{i \mid d_i < 1.6 \text{ km}\}}))),$   | $h_{likeliest} \leq 3\;\mathrm{km} \wedge \mathrm{mean}(h_{i \in \{i \mid d_i < 1.6~\mathrm{km}\}}) > 1.5\;\mathrm{km}$ | (7) |
|                                             | $\min(1.5 \text{ km}, \max(h_{i \in \{i \mid d_i < 1.2 \text{ km}\}}))),$ | $h_{likeliest} \leq 3 \text{ km} \wedge \text{mean}(h_{i \in \{i d_i < 1.6 \text{ km}\}}) \leq 1.5 \text{ km}.$         |     |

3.4 Estimation of conditional probabilities of CBH (ORDER OF SECTIONS 3.3 AND 3.4 WAS EXCHANGED)

The procedure to combine CBH-measurements from independent ASI-pairs, which are organized as a network, requires knowledge of the (conditional) probability to receive a certain reading of CBH from an ASI-pair given the true CBH takes on some specific value. The method itself will be presented in Sect. 3.3. Here we discuss the probability distributions used. The required distribution aims to answer the following question: If true CBH ranges in between 1.8...1.9 km, how large will be the probability that an ASI-pair with camera distance 2.2 km delivers a certain CBH e.g. within 0...0.1 km or 1.8...1.9 km or 11.9...12 km? In the following, these conditional probabilities are estimated not only for the range of true CBH between 1.8...1.9 km but

for each range {0...0.1, 0.1...0.2, 0.2...0.3, ..., 11.9...12} km of true CBH. Conditional probability distributions of this kind are

- 410 not available so far for ASI-pairs. Therefore, we aim to approximate them from the measurement data of a modelling period. Estimations of CBH from the available ASI-pairs and measurements from the ceilometer during the period 01 April 2019 to 29 June 2019 are used. CBH measured by the ceilometer serves as reference CBH. It is considered not to be essential that the training period is representative of the period to which the method is applied. However, we expect that the method works best if the included ASI-pairs exhibit a similar distribution of measurement deviations given the same reference CBH in both
- 415 periods. For solar applications and the latitude of this study, we consider the used dataset and its split reasonable. The summer and shoulder months provide the main share of the annual solar yield at the site and are therefore in the focus of the nowcasting system under development. In that sense, the training dataset is considered to be for the large part representative of conditions relevant to solar applications at similar latitudes.

The seven ASIs available in the urban area are arranged into 42 ASI-pairs. Each tuple of two ASIs, that is selected from the 420 set of seven ASIs, yields 2 independent ASI-pairs by swapping the ASI used as main camera (see Sect. 3.1).

The procedure is developed based on periods in which valid measurements from ceilometer and the respective ASI-pair are available and in which the variability of CBH is moderate: For each time stamp a window of 30 min centered at this time stamp is defined. A time stamp is only included if standard deviation of reference CBH within the window is less than 30% of the mean value of reference CBH within the same window. As discussed before, ASI-pairs and ceilometer measure CBH

425 as spatial median and point-wise respectively. Therefore, this filter intends to assure that ceilometer and ASI-pair measure CBH of the same layer. CBH from the respective ASI-pair and from the ceilometer are processed by a moving-median filter with a window of 10 min. The joint frequency distribution of CBH measured by ceilometer  $h_{Ref}$  and the respective ASI-pair  $h_{ASI}$  is computed from these simultaneously acquired time series. That means the In other words, the domain of reasonable values,  $[0, 12 \text{ km}] \times [0, 12 \text{ km}]$ , which the pair  $(h_{Ref}, h_{ASI})$  can take on, is discretized into a mesh of square grid cells with

430 side lengths  $\Delta h$ . Then the frequency is calculated with which  $(h_{Ref}, h_{ASI})$  is observed in a discrete grid cell defined by the interval  $[j\Delta h, (j+1)\Delta h]$  for  $h_{Ref}$  and the interval  $[k\Delta h, (k+1)\Delta h]$  for  $h_{ASI}$ , where  $j,k \in \{0,1,2,...,N-1\}$  each of the discrete grid cells. A bin size  $\Delta h = 100$  m is chosen in a trade-off between sources of error. Finer bins will allow to represent the distributions at higher resolution and will thus allow for higher resolved measurements of CBH in the network. However, the size of the used data set is limited which makes it difficult to model these distributions at highest resolution. The bin size

435 chosen here is expected to limit the achievable uncertainty of the measurement to a minimum level of 100 m. Joint frequency distributions modeled here are restricted to a maximum CBH of 12 km. This yields N = 120.

Joint frequency distributions were inspected and found to be well reproduced among the studied independent ASI-pairs, if only the corresponding camera distances are similar. This meets the expectation from literature discussed in Sect. 3.1. Moreover, we conclude that the distributions modeled here will be transferable to other setups that use camera distances in the

440 studied range. Local climate is expected to influence the transferability to a minor extentas will be discussed later. To further support this transferability to.

The limited size and representativeness of the data set used in model development are expected to cause random features in the joint frequency distributions which are not useful to the estimation procedure, when it is applied to other setups, sites and

times ... we aim to suppress (such as represented by the validation data set). To suppress such random features of received joint

- 445 frequency distributions. For this, the original joint frequency distribution  $F_1$  of ASI pair *l* is transformed by a first filter into  $F_{1,filter,1}$  and by a consecutively applied filter into  $F_{1,filter,2}$ , we introduce a filtering procedure with two consecutive steps described here and in more detail in Appendix A. The parameter values set in the filtering procedure are approximate to this point and are based on a visual comparison of unfiltered and filtered distributions, evaluating the degree to which noise but also reasonable features were suppressed. The parameter values may be optimized in a future study.
- 450 First, a weighted mean filter is applied between the original joint frequency distributions *Ft* received from all received for ASI-pairs with arbitrary camera distanced-

$$F_{l,filter \ 1} = \frac{\sum_j w_{l,m} F_m}{\sum_j w_{l,m}}.$$

For the joint frequency distribution  $F_l$  of each respective ASI-pair *l*, weights  $w_{l,m}$  are used that include similar camera distance. As discussed above, ASI-pairs with similar camera distance - More precisely, a triangular window, based on the difference of eamera distance  $\Delta d_{Lm}$  of ASI-pair *m* compared to ASI-pair *l*, is used that is defined by-

 $w_{l,m} = max(0, 1 - \Delta d_{l,m}/0.5 \text{ km}).$

Then are expected to perform similarly in the measurement of CBH and should consequently also exhibit similar joint frequency distributions of CBH. Thus, the filter aims to suppress differences between the joint frequency distributions of ASI-pairs which may result from disturbances in the estimation rather than from a difference in the systems' characteristics.

- 460 To each filtered distribution resulting from the prior step, a composite of three Gaussian filters is applied to  $F_{l,filter,1}$  of each ASI-pair *l*. We first decompose each distribution  $F_{l,filter,1}$  by conditional filters into three separate modes. In the second step, which correspond to parts of the joint frequency distributions which are estimated with descending precision. Thereafter, we apply to each mode a Gaussian filter  $g_{\sigma}$  with distinct standard deviation  $\sigma_{mode}$  to each mode. The standard deviation of the Gaussian kernel. The subscript mode indicates the specific mode for which  $\sigma_{mode}$  is applied. filter applied to each mode
- 465 corresponds qualitatively to the uncertainty with which the prior joint frequency distribution is estimated within grid cells of that mode. Consecutively, the three filtered modes are summed to receive the smoothened joint frequency distribution. The first mode is constituted by all outlier observations. Outliers are defined here as grid cells ( $h_{Ref}$ ,  $h_{ASI}$ ) for which grid cells for which the ASI-pair based measurement of CBH  $h_{ASI}$ -deviates by more than 1.5 km from the ceilometer reading $h_{Ref}$ .

470
$$F_{l,outlier}(h_{Ref}, h_{ASI}) = \begin{cases} F_{l,filter \ 1}(h_{Ref}, h_{ASI}), & |h_{ASI} - h_{Ref}| > 1.5 \text{ km} \\ 0, & \text{else.} \end{cases}$$

,

Such outliers will contain a large random component. We expect that in a reproduction of the experiment, a similar number of outliers will be received, while. The large deviations represented by this mode occur less frequently which is why the joint frequency distribution will be estimated less precisely for the respective grid cells. On the other hand, apart from such

scattering effects, the joint frequency found for a single grid cell  $(h_{ReI}, h_{ASI})$  may vary significantly. Therefore, the strongest

475 filter distributions are found to be comparably smooth in the grid cells of this mode. A Gaussian filter with a large standard deviation of 1 km is applied to this mode using  $\sigma_{outliver} = 1$  km, which is considered to be apt to preserve the expected distribution while suppressing random features.

The second mode is constituted by grid cells that are not part of the first mode and for which the ASI-pair based measurement of CBH deviates by less than 1.5 km from the ceilometer reading and which feature a joint frequency less than the average 480 over below the average of all grid cells of the joint frequency distribution:

|                                           | $F_{l,filter\ 1}(h_{Ref},h_{ASI}),$ | $ h_{ASI}-h_{Ref}  \leq 1.5 \ \rm km$                                          |
|-------------------------------------------|-------------------------------------|--------------------------------------------------------------------------------|
| $F_{l,inconfident}(h_{Ref}, h_{ASI}) = c$ |                                     | $\wedge \ F_{l,filter \ 1}(h_{Ref},h_{ASI}) < \mathrm{mean}(F_{l,filter \ 1})$ |
|                                           | 0,                                  | else.                                                                          |

The . These grid cells typically exhibit a larger joint frequency, i.e. more observations, than grid cells in the first mode. Still the comparably small number of observations in these grid cells is expected to cause an increased uncertainty of the estimated joint frequencies. For this mode,  $\sigma_{inconfident} = 0.5$  km Consequently in a trade-off between suppressing random scattering and preserving meaningful variations a Gaussian filter with standard deviation 0.5 km is applied.

The third mode  $F_{l,confident}(h_{Ref},h_{AST})$ -makes up the complementary of the first and second mode. It contains grid cells that are observed with an at least average joint frequency and which are not classified as outliers:-

|                                               | $F_{l,filter\ 1}(h_{Ref},h_{ASI}),$ | $ h_{ASI} - h_{Ref}  \le 1.5 \; \mathrm{km}$                                  |
|-----------------------------------------------|-------------------------------------|-------------------------------------------------------------------------------|
| $F_{l,confident}(h_{Ref}, h_{ASI}) = \langle$ |                                     | $\wedge F_{l,filter\ 1}(h_{Ref},h_{ASI}) \geq \mathrm{mean}(F_{l,filter\ 1})$ |
|                                               | 0,                                  | else.                                                                         |

Joint frequencies in these grid cells are considered to have be estimated with a comparably high accuracy. To avoid a loss

490 of precision and ultimately a loss of accuracy in the estimation of CBH, a small value of  $\sigma_{confident} = 0.1$  km Gaussian filter with a standard deviation of 0.1 km is used. The three filtered modes  $g_{\sigma}$  are summed to receive the smoothened joint frequency distribution-

 $F_{l,filter\ 2} = g_{\sigma_{outlier}}(F_{l,outlier}) + g_{\sigma_{inconfident}}(F_{l,inconfident}) + g_{\sigma_{confident}}(F_{l,confident}).$

Hence, only neighboring grid cells have a significant influence on this filter.

- In many joint frequency distributions, there are grid cells with joint frequency close to zero. Especially for these grid cells, a greater dataset data set would be required to receive more representative values. For all grid cells, joint frequency is increased to a minimum value of 0.5 to avoid underestimations of joint frequency. This value corresponds to half of the joint frequency associated with a single actual observation in a grid-cell. For the estimation procedure of CBH, this such a minimum value leads to slightly reduced precision for most readings but increased robustness in the case that these grid cells  $(h_{Ref}, h_{ASI})$  are
- 500 indeed observed in the measurement.

485

Finally, from each joint frequency distributionis normalized with the sum of all joint frequency grid cells. In this way, a probability mass function (also known as discrete density function) to measure a certain CBH with the respective ASI-pair and to coincidentally measure a certain CBH with the ecolometer is yielded. The , the conditional probability  $P(h_{ASI} | h_{Ref})$ to receive a certain CBH reading from an ASI-pair, given that the ceilometer measures some certain CBH, is calculated by

- 505 dividing the respective probability mass function by the marginal distribution of CBH measured by the ceilometer. The latter distribution gives the probability to receive CBH from the ceilometer within a certain bin  $h_{Ref}$  regardless of which CBH reading is simultaneously received from an ASI-pair. The distrbution can be derived from any of the probability mass functions by summing all grid cells of the probability mass function which correspond to the respective bin  $h_{Ref}$  of CBH measured by the ceilometer, derived (see Appendix A for a more detailed description).
- 510 Inference procedure Step 1: For each range *i* of camera distance CBH1 is computed as mean CBH from the respective ASI-pairs. Conditional probability is evaluated that CBH1 would be received if true CBH (at the ceilometer) took on a value {0...0.1,0.1...0.2,...,11.9...12} (red boxes). Step 1 yields a likelihood function for each range of camera distance. Step 2: Cumulative and complementary cumulative likelihood are calculated for each range of camera distance. Step 3: These functions are logarithmized and then summed over all ranges *i* of camera distance yielding overall cumulative and complementary cumulative likelihood functions gives the estimated likeliest CBH.
- The inference procedure, which is was introduced in Sect. 3.3, represents each range i of camera distance bounded by the limits  $\{0.5, 1, 1.5, ..., 6\}$  km by a single distribution of conditional probability. For each range of camera distance, the distribution of conditional probability, which corresponds to the camera distance closest to the center of this range, is selected . For example, for the range i 2 representing camera distances 1...1.5 km, the center of the range would be 1.25 km. For
- 520 the camera distances 1.081, 1.247 and 1.352 km, conditional probabilities have been modeled. Consequently, for this range of camera distance, the distribution of conditional probability corresponding to the camera distance 1.247 km is used. (example provided in Appendix A). Figure 5 (above Step 1) shows exemplary conditional probabilities for three ASI-pairs with camera distances 0.8, 2.2, 5.7 km representing the ranges of camera distance i = 1, 4, 11 respectively. The further content of Fig. 5 is explained in the next sectionBIAS and precision, with which ASI pairs of distinct camera distances measure CBH, given a
- 525 certain reference CBH, are visible in these conditional probabilities. Such characteristics will be evaluated in more detail in the following, based on a separate validation data set.

**Appendix A, pp. 35-37:**

880 Appendix A: Details on the retrieval of conditional probabilities

**A1 Retrieval of raw joint frequency distributions**

CBH from the respective ASI-pair and from the ceilometer are processed by a moving-median filter with a window of 10 min. The joint frequency distribution of CBH measured by ceilometer  $h_{Bef}$  and the respective ASI-pair  $h_{ASI}$  is computed from these simultaneously acquired time series. That means, the frequency is calculated with which  $(h_{Bef}, h_{ASI})$  is observed in

- a discrete grid cell defined by the interval  $[i\Delta h, (j \pm 1)\Delta h]$  for  $h_{Ref}$  and the interval  $[k\Delta h, (k \pm 1)\Delta h]$  for  $h_{ASI}$ , where  $j, k \in \{0, 1, 2, ..., N 1\}$ , where N is the number of bins used for CBH in the analysis. A bin size  $\Delta h = 100$  m is chosen in a trade-off between sources of error. Finer bins will allow to represent the distributions at higher resolution and will thus allow for higher resolved measurements of CBH in the network. However, the size of the used data set is limited which makes it difficult to model these distributions at highest resolution. The bin size chosen here is expected to limit the achievable uncertainty of
- 890 the measurement to a minimum level of 100 m. Joint frequency distributions modeled here are restricted to a maximum CBH of 12 km. This yields N = 120.

**A2 Filtering operations applied**

First, a weighted mean filter is applied between original joint frequency distributions  $F_l$  received from all ASI-pairs with camera distance d, this yields  $F_{l,fulter,1}$ ;

$$F_{l,filter 1} = \frac{\sum_{m} w_{l,m} F_{m}}{\sum_{m} w_{l,m}}.$$
(A1)

For the joint frequency distribution  $F_l$  of each respective ASI-pair l, weights  $w_{l,m}$  are used that include ASI-pairs with similar camera distance. More precisely, a triangular window, based on the difference of camera distance  $\Delta d_{l,m}$  of ASI-pair m compared to ASI-pair l, is used that is defined by

$$w_{l,m} = max(0, 1 - \Delta d_{l,m}/0.5 \text{ km}). \tag{A2}$$

900 We decompose each distribution  $F_{L,filter,1}$  by conditional filters into three separate *modes*. In the second step, we apply to each mode a Gaussian filter  $g_{\sigma}$  with distinct standard deviation  $\sigma_{mode}$  of the Gaussian kernel. The subscript *mode* indicates the specific mode for which  $\sigma_{mode}$  is applied. The first mode is constituted by all outlier observations. Outliers are defined here as grid cells ( $h_{Bef}$ ,  $h_{ASI}$ ) for which ASI-pair measurement of CBH  $h_{ASI}$  deviates by more than 1.5 km from the ceilometer reading  $h_{Ref}$ :

905
$$F_{l,outlier}(h_{Ref}, h_{ASI}) = \begin{cases} F_{l,filter\ 1}(h_{Ref}, h_{ASI}), & |h_{ASI} - h_{Ref}| > 1.5 \text{ km} \\ 0, & \text{else.} \end{cases}$$
 (A3)

Such outliers will contain a large random component. We expect that in a reproduction of the experiment, a similar number of outliers will be received, while the joint frequency found for a single grid cell  $(h_{Bcfa}, h_{ASL})$  may vary significantly. Therefore, the strongest filter is applied to this mode using  $\sigma_{outlier} = 1$  km.

The second mode is constituted by grid cells that are not part of the first mode and feature a joint frequency less than the 910 average over all grid cells of the joint frequency distribution:

$$F_{l,inconfident}(h_{Ref}, h_{ASI}) = \begin{cases} F_{l,filter\ 1}(h_{Ref}, h_{ASI}), & |h_{ASI} - h_{Ref}| \le 1.5 \text{ km} \\ & \wedge F_{l,filter\ 1}(h_{Ref}, h_{ASI})

---

## Author Response (AR1)

**Authors' response to Reviewer Comment 1**

Reviewer:

The Authors present and evaluate an approach to derive cloud-base height (CBH) from a network of seven upward looking all-sky imagers (ASIs). The analysis focusses on a region in NW Germany during summer and shoulder seasons. The authors demonstrate that a network approach outperforms individual pairs of ASIs.

The manuscript is generally well-written, and the figures complement the main text appropriately. I recommend publication of this article after resolving several general and few minor comments.

Authors' response:

We would like to thank the reviewer a lot for the time and effort spent on providing feedback to our manuscript and for the insightful comments. We believe that these led to valuable improvements of our manuscript. We addressed all comments and have incorporated all of the suggestions made by the reviewer as good as it was possible to us.

In the following, we will address the reviewer's further comments point-by-point. Changes are extracted from the adapted manuscript within which changes were highlighted using latexdiff. Blue indicates insertions, red indicates deletions. Please note, that the order of Sect. 3.3 and Sect. 3.4 has been reversed as suggested by General Comment 5. This change has been excluded from the markup, as it would have obscured all other changes. Further, please note, that Sect. 3 and Sect. 4 have been reworked strongly, based on Reviewer Comment 2, Major Comments 1, 2.

Changes in manuscript:

See below.

**General comment 1:**

The Authors motivate their work as it allows to better nowcast downwelling solar fluxes (e.g., for photovoltaic power plants) and it is said that "accurate knowledge of CBH is required". It is not perfectly obvious why better knowledge of CBH itself improves nowcasting. I'm assuming CBH is only one piece of information - apart from knowledge of each cloud's horizontal extend, cloud-top height, and geolocation (derived from satellite?) as well as the wind vector in cloudy altitudes (from meteorological forecasts or from ASIs?). Section 1 (ll. 26-32, ll. 48-53) touches on this topic but leaves open questions of how exactly this work fits into a larger picture. It is also unclear to me if voxel carving (ll. 58-59) is a competing approach or if this work could be used for voxel carving efforts – the Authors should clarify this in Section 1.

Authors' response:

We agree with the reviewer, that indeed nowcasting of solar irradiance is a complex task which includes a number of subtasks, which may all bring uncertainties to the method. Based on previous works, e.g.

Nouri, B., S. Wilbert, P. Kuhn, N. Hanrieder, M. Schroedter-Homscheidt, A. Kazantzidis, L. Zarzalejo, P. Blanc, S. Kumar and N. Goswami (2019). "Real-Time Uncertainty Specification of All Sky Imager Derived Irradiance Nowcasts." Remote Sensing **11**(9): 1059.,

knowledge of CBH was identified as critical, especially if the accurate position of cloud edges is of interest. We addressed this by a summary of the nowcasting procedure, pointing out the importance of cloud base height (CBH).

We also added a short explanation on the relationship of stereoscopic and voxel carving approaches. From our perspective these approaches are in principle competing. However, previous works have shown that voxel carving approaches can be improved, if CBH is received from a stereoscopic approach.

Changes in manuscript:

p. 3, ll. 76-78:

75 here (Nouri et al., 2019a) enhances the approach by Kuhn et al. (2018b) and works completely independently from cloud recognition which is considered to bring a greater robustness. While stereoscopic and voxel carving/ tomographic approaches are in principle competing techniques, Nouri et al. (2019a) demonstrated, that voxel carving-based cloud modelling can be enhanced by incorporating CBH from a stereoscopic procedure.

p.4 ll. 91-102:

90 The selected ASIs are located in the city of Oldenburg. At the moment of writing, Eye2Sky contains 24 ASIs in Oldenburg and a region of about $110\,\mathrm{km}\times100\,\mathrm{km}$ to the west of Oldenburg.  Eye2Sky is mainly dedicated to nowcasting of solar irradiance at high spatial and temporal resolution. The forecasting procedure, which will be described in more detail in a future publication, first recognizes clouds from the images of the ASIs. Cloud observations are then projected into a horizontal plane at the current CBH. These georeferenced cloud

95 observations of multiple ASIs are merged and cloud properties are estimated. The angular velocities of clouds, as recognized by the individual ASIs, are transformed into absolute velocities over ground relying on an accurate estimation of CBH. Clouds are tracked along received cloud motion vectors to predict the clouds' future positions. Prior works studying ASI-based forecasting systems with up to four cameras (e.g. Nouri et al., 2019b) suggested that CBH is an essential component when predicting maps of solar irradiance based on cloud observations from ASIs, as the current and future positions of cloud shadows on the ground

100 can only be predicted accurately if the clouds' height and velocity are determined accurately. Thus, in this publication an important component of this nowcasting system, namely the estimation of CBH, is presented. Our approach allows to use multiple ASI-pairs  organized as ASI network and located in proximity, to estimate CBH. 42 ASI-pairs are formed from the seven ASIs and CBH is estimated by each ASI-pair based on the method presented by Nouri

**General comment 2:**
Reviewer:

When using a network of ASIs over an area of (100km)2 to obtain a single CBH, do the Authors inherently assume a cloud (or a field of clouds) of unique base height? The Authors should make this more explicit (perhaps in Section 3) and discuss the realism of this assumption (perhaps in Section 4.4)

Authors' response:

We share the reviewer's opinion that the use of CBH assessed in the urban area for the whole region of Eye2Sky measuring 100 km x 100 km is a strong simplification. We now tried to outline, which scope our method may fulfill and how the method can be enhanced for a broader scope in future. For this, we added an explanation to Sect. 3.2, as in this section the conditions at the studied site are analyzed. More precisely our expectation from the conditions on site is, that our method is suited to provide an estimation of CBH which is useful to nowcast solar irradiance in the urban area of Oldenburg. For the task of providing nowcasts for the whole of Eye2Sky, we intend to classify the cloud conditions at larger distances from the urban area by a number of ASIs which are dispersed over the region and then to assign CBH from the urban area, by looking up which CBH was observed recently in the urban area for similar cloud conditions.

Changes in manuscript:

p. 12, ll. 305-310:

305    characteristics are expected to cause greater temporal and spatial variability of CBH. To conclude, a procedure, which estimates CBH of the cloud layer most dominant in the urban area of Oldenburg accurately, is considered beneficial to assess and model clouds in the same area (depicted in Fig. 1, right). Still, if clouds over the whole region covered by Eye2Sky (depicted in Fig. 1, left) are assessed, this method alone may not be sufficient. In the future, local cloud conditions may be classified by image processing techniques (e.g. Fabel et al., 2021) and CBH may be assigned to local clouds from clouds of the same type, which
310    were recently observed in the urban area.

**General comment 3:**
Reviewer:

To obtain CBH probabilities (Section 3.3) the Authors use a subset of available datapoints. It is unclear what portion of the data was excluded. Did this selection mostly affect samples of high-altitude clouds? Perhaps the Authors could add a column to Table 1 that lists the fraction of data points excluded per altitude group?

Authors' response:

First of all, we would like to apologize as a statement in the manuscript was misleading. The filter excluding variable situations is applied in the modelling of conditional probabilities (now Sect. 3.4) and in Sect. 4.4 to compare performance metrics from ASI-pairs and ASI network. In Sect. 4.1-4.3 this filter is not applied.

As suggested, we added a column to Table 1, indicating the excluded fraction of time stamps per interval of CBH and added a description, how the filter influences the distribution of CBH in the validation data set. The filter excludes observations from all ranges of CBH in a similar way. However, for the lowest range of CBH a larger fraction is excluded. As this range is still represented by a large number of observations in the filtered data set, this was accepted.

Changes in manuscript:

Description on filtering of validation data was corrected, moved; additional description of the filter effect added (Sect. 4.4):

pp. 20-21, ll. 531-535:

530    from 30 June 2019 to 27 September 2019. This dataset was excluded from the model development described in Sect. 3. The analyzed quantity is 10 min-median CBH. ~~The evaluations are restricted to times in which the variability of CBH is small. More precisely, the standard deviation of CBH within a window 15 min before and after the analyzed time is required to be less than 30% of mean CBH within the same window. As discussed above, the ASI-pairs and the ASI network are expected to measure a spatial median CBH whereas the ceilometer measures CBH at the point of its installation. This restriction aims to~~

535

pp. 28-29, ll. 691-702:

The statistical evaluations are now restricted to times in which the variability of CBH is small. More precisely, the standard deviation of CBH within a window 15 min before and after the analyzed time is required to be less than 30% of mean CBH

within the same window. As discussed above, the ASI-pairs and the ASI network are expected to measure a spatial median CBH whereas the ceilometer measures CBH at the point of its installation. This restriction aims to assure a good comparability

695    of both measurements. Further, this way our results are more comparable to a prior study by Kuhn et al. (2019).

Accuracies of CBH measurement by ASI-pairs and ASI network are analyzed separately for five ranges of reference CBH defined by the bounds $\{0, 1, 2, 4, 8, 12\}$ km. The number of CBH measurements included in this evaluation is given in Table 1 for each of these ranges. The interval bounds are spaced irregularly to correspond better to the distribution of CBH at the site (see also Fig. 4). Table 1 also shows the number of observations excluded from the validation as significant temporal variability

700    of CBH was detected for these observations. While a significant fraction of the readings is sorted out, the representation of the CBH ranges remains widely comparable to the original data set (see Fig. 2, left). Only the range of lowest CBH $< 1000$ m is represented by a notably smaller share of the validation data set.

p. 28, Table 1:

**Table 1.** Frequency of measurements from the validation data set (period 30 June 2019 to 27 September 2019) per range of cloud base height (CBH) used in the evaluations described in Sect. 4.4 (retained) and frequency of those filtered from the evaluation due to increased variability of CBH (rejected).

| CBH range [km] | Observations retained | Observations rejected |
|---|---|---|
| $0 < \text{CBH} \leq 1$ | 11844 | 13255 |
| $1 < \text{CBH} \leq 2$ | 14130 | 9120 |
| $2 < \text{CBH} \leq 4$ | 9962 | 5923 |
| $4 < \text{CBH} \leq 8$ | 5559 | 3570 |
| $8 < \text{CBH} \leq 12$ | 4935 | 1355 |

**General comment 4:**

Reviewer:

The Authors measure accuracy of their approach by using a three-month dataset, shown in Fig. 9 and elaborated in Section 4.3. From a machine-learning stand point is would be important to know if these were "training samples" (i.e., used to prepare CBH probabilities, etc.) or whether these data points were withheld from algorithm preparation.

Authors' response:

In the study we use two separate data sets: one for training/development of the method and one for the test/validation. The training period is 01 April 2019 to 29 June 2019. The validation period is 30 June 2019 to 27 September 2019. We hope that this answers the reviewer's question satisfyingly. We revised passages, by which we intended to describe this split of the used dataset, as shown below, for more clarity.

Changes in manuscript:

p. 20, l. 530:

> In this section, the accuracy of CBH measurement by the ASI network and by 42 independent ASI-pairs set up at a wide variety of camera distances and alignments is compared. This section is based on a validation data set including the days
> 530 from 30 June 2019 to 27 September 2019. This dataset was excluded from the model development described in Sect. 3. The

p. 7, ll. 160-162:

> 160 used for deriving the method (until 29 June 2019) and a period used for validations (starting from 30 June 2019). Time stamps from the validation period 30 June 2019 to 27 September 2019 are excluded from the model development and also from the estimation of conditional probabilities.

**General comment 5:**

Reviewer:

The Authors introduce the Maximum Likelihood Estimation (MLE) approach in Section3.4 and – before in Section 3.3 – provide information on conditional CBH probability. This arrangement seems confusing to me and recommend that Section 3.3 follows 3.4(or is a subsection of 3.4).

Authors' response:

We changed the order of the sections accordingly. As this change would obscure the markup of other changes in the red-line version, we excluded this exchange from the markup (by applying the change before comparing with latexdiff).

These sections have additionally been revised strongly based on Reviewer Comment 2, Major Comment 1.

Changes in manuscript:

Order of 3.3 and 3.4 is exchanged (p. 12-20, ll. 311-526):

> **3.3 Estimating CBH in the ASI network (ORDER OF SECTIONS 3.3 AND 3.4 WAS EXCHANGED)**
>
>  In this section we present our method to combine the measurements of CBH from a large number ASI-pairs organized as network. Prior works estimated CBH by a small number of two or in some cases four ASIs (Nouri et al., 2019a). However, with a large number of ASI-pairs, we consider a statistical method
> 315 promising, which analyzes the CBH samples received and, based on the known characteristics of each ASI-pair, determines

> **3.4 Estimation of conditional probabilities of CBH (ORDER OF SECTIONS 3.3 AND 3.4 WAS EXCHANGED)**
>
> The procedure to combine CBH-measurements from independent ASI-pairs, which are organized as a network, requires knowledge of the (conditional) probability to receive a certain reading of CBH from an ASI-pair given the true CBH takes on some
> 405 specific value. The  required

**General comment 6:**

Reviewer:

Section 3.3 lists a variety of filters that were applied (ll. 240ff). The Authors should revise Section 3.3 and reference the use of these filters - if applied in the past – and explain their intended effect.

Authors' response:

We revised Sect. 3.4 (in new order) strongly according to the reviewer's feedback but also based on the Reviewer Comment 2, Major Comment 1. We now pointed out, why a method to estimate the distributions of conditional probability from measurement data, was developed, which was new in our perspective at least to this application. Such distributions were so far not available for stereoscopic CBH measurements.

We moved details on the implementation of the filters to the appendix and focused on the intended effects of the filters and motivated the value assigned to the parameters of the filters.

Changes in manuscript:

pp. 16-20, ll. 403-526

**3.4 Estimation of conditional probabilities of CBH (ORDER OF SECTIONS 3.3 AND 3.4 WAS EXCHANGED)**

The procedure to combine CBH-measurements from independent ASI-pairs, which are organized as a network, requires knowledge of the (conditional) probability to receive a certain reading of CBH from an ASI-pair given the true CBH takes on some

405    specific value. The  required distribution aims to answer the following question: If true CBH ranges in between 1.8...1.9 km, how large will be the probability that an ASI-pair with camera distance 2.2 km delivers a certain CBH e.g. within 0...0.1 km or 1.8...1.9 km or 11.9...12 km? In the following, these conditional probabilities are estimated not only for the range of true CBH between 1.8...1.9 km but

times  (such as represented by the validation data set). To suppress such random features of received joint

445 frequency distributions, we introduce a filtering procedure with two consecutive steps described here and in more detail in Appendix A. The parameter values set in the filtering procedure are approximate to this point and are based on a visual comparison of unfiltered and filtered distributions, evaluating the degree to which noise but also reasonable features were suppressed. The parameters values may be optimized in a future study.

450 First, a weighted mean filter is applied between the original joint frequency distributions  received for ASI-pairs with $d$

$$F_{l,filter\ 1} = \frac{\sum_j w_{l,m} F_m}{\sum_j w_{l,m}}.$$

 similar camera distance. As discussed above, ASI-pairs with similar camera distance

455

$$w_{l,m} = max(0, 1 - \Delta d_{l,m}/0.5 \text{ km}).$$

 are expected to perform similarly in the measurement of CBH and should consequently also exhibit similar joint frequency distributions of CBH. Thus, the filter aims to suppress differences between the joint frequency distributions of ASI-pairs which may result from disturbances in the estimation rather than from a difference in the systems' characteristics.

460 To each filtered distribution resulting from the prior step, a composite of three Gaussian filters is applied. We first decompose each distribution  by conditional filters into three separate *modes*.  which correspond to parts of the joint frequency distributions which are estimated with descending precision. Thereafter, we apply  a Gaussian filter  to each mode. The standard deviation of the Gaussian  filter applied to each mode

465 corresponds qualitatively to the uncertainty with which the prior joint frequency distribution is estimated within grid cells of that mode. Consequently, the three filtered modes are summed to receive the smoothened joint frequency distribution.

The first mode is constituted by  grid cells for which the ASI-pair based measurement of CBH  deviates by more than 1.5 km from the ceilometer reading

470 $$F_{l,outlier}(h_{Ref}, h_{ASI}) = \begin{cases} F_{l,filter\ 1}(h_{Ref}, h_{ASI}), & |h_{ASI} - h_{Ref}| > 1.5 \text{ km} \\ 0, & \text{else.} \end{cases}$$

. The large deviations represented by this mode occur less frequently which is why the joint frequency distribution will be estimated less precisely for the respective grid cells. On the other hand, apart from such

scattering effects, the joint frequency  distributions are found to be comparably smooth in the grid cells of this mode. A Gaussian filter with a large standard deviation of 1 km is applied to this mode  which is considered to be apt to preserve the expected distribution while suppressing random features.

The second mode is constituted by grid cells  for which the ASI-pair based measurement of CBH deviates by less than 1.5 km from the ceilometer reading and which feature a joint frequency  below the average of all grid cells

$$F_{l,inconfident}(h_{Ref}, h_{ASI}) = \begin{cases} F_{l,filter\ 1}(h_{Ref}, h_{ASI}), & |h_{ASI} - h_{Ref}| \leq 1.5 \text{ km} \\ & \wedge F_{l,filter\ 1}(h_{Ref}, h_{ASI}) < \text{mean}(F_{l,filter\ 1}) \\ 0, & \text{else.} \end{cases}$$

. These grid cells typically exhibit a larger joint frequency, i.e. more observations, than grid cells in the first mode.  comparably small number of observations in these grid cells is expected to cause an increased uncertainty of the estimated joint frequencies.  Consequently in a trade-off between suppressing random scattering and preserving meaningful variations a Gaussian filter with standard deviation 0.5 km is applied.

The third mode  makes up the complementary of the first and second mode. It contains grid cells that are observed with an at least average joint frequency and which are not classified as outliers

$$F_{l,confident}(h_{Ref}, h_{ASI}) = \begin{cases} F_{l,filter\ 1}(h_{Ref}, h_{ASI}), & |h_{ASI} - h_{Ref}| \leq 1.5 \text{ km} \\ & \wedge F_{l,filter\ 1}(h_{Ref}, h_{ASI}) \geq \text{mean}(F_{l,filter\ 1}) \\ 0, & \text{else.} \end{cases}$$

. Joint frequencies in these grid cells are considered to  be estimated with a comparably high accuracy. To avoid a loss of precision and ultimately a loss of accuracy in the estimation of CBH, a  Gaussian filter with a standard deviation of 0.1 km is used.

$$F_{l,filter\ 2} = g_{\sigma_{outlier}}(F_{l,outlier}) + g_{\sigma_{inconfident}}(F_{l,inconfident}) + g_{\sigma_{confident}}(F_{l,confident}).$$

Hence, only neighboring grid cells have a significant influence on this filter.

In many joint frequency distributions, there are grid cells with joint frequency close to zero. Especially for these grid cells, a greater  data set would be required to receive more representative values. For all grid cells, joint frequency is increased to a minimum value of 0.5 to avoid underestimations of joint frequency. This value corresponds to half of the joint frequency associated with a single actual observation in a grid-cell. For the estimation procedure of CBH,  such a minimum value leads to slightly reduced precision for most readings but increased robustness in the case that these grid cells $(h_{Ref}, h_{ASI})$ are indeed observed in the measurement.

 Finally, from each joint frequency distribution, the conditional probability $P(h_{ASI} \mid h_{Ref})$ to receive a certain CBH reading from an ASI-pair, given that the ceilometer measures some certain CBH, is

505 ~~dividing the respective probability mass function by the marginal distribution of CBH measured by the ceilometer. The latter distribution gives the probability to receive CBH from the ceilometer within a certain bin $h_{Ref}$ regardless of which CBH reading is simultaneously received from an ASI-pair. The distrbution can be derived from any of the probability mass functions by summing all grid cells of the probability mass function which correspond to the respective bin $h_{Ref}$ of CBH measured by the ceilometer.~~ derived (see Appendix A for a more detailed description).

510 ~~Inference procedure — Step 1: For each range $i$ of camera distance $CBH_r$ is computed as mean CBH from the respective ASI-pairs. Conditional probability is evaluated that $CBH_r$ would be received if true CBH (at the ceilometer) took on a value $\{0...0.1, 0.1...0.2, ..., 11.9...12\}$ (red boxes). Step 1 yields a likelihood function for each range of camera distance. Step 2: Cumulative and complementary cumulative likelihood are calculated for each range of camera distance. Step 3: These functions are logarithmized and then summed over all ranges $i$ of camera distance yielding overall cumulative and complementary~~

515

The inference procedure, which  was introduced in Sect. 3.3, represents each range $i$ of camera distance bounded by the limits $\{0.5, 1, 1.5, ..., 6\}$ km by a single distribution of conditional probability. For each range of camera distance, the distribution of conditional probability, which corresponds to the camera distance closest to the center of this range, is selected

520  (example provided in Appendix A). Figure 5 (above Step 1) shows exemplary conditional probabilities for three ASI-pairs with camera distances 0.8, 2.2, 5.7 km representing the ranges of camera distance $i = 1$, 4, 11 respectively. BIAS and precision, with which ASI pairs of distinct camera distances measure CBH, given a

525 certain reference CBH, are visible in these conditional probabilities. Such characteristics will be evaluated in more detail in the following, based on a separate validation data set.

Appendix A, pp. 35-37:

880 **Appendix A: Details on the retrieval of conditional probabilities**

**A1 Retrieval of raw joint frequency distributions**

CBH from the respective ASI-pair and from the ceilometer are processed by a moving-median filter with a window of 10 min. The joint frequency distribution of CBH measured by ceilometer $h_{Ref}$ and the respective ASI-pair $h_{ASI}$ is computed from these simultaneously acquired time series. That means, the frequency is calculated with which $(h_{Ref}, h_{ASI})$ is observed in

885 a discrete grid cell defined by the interval $[j\Delta h, (j+1)\Delta h[$ for $h_{Ref}$ and the interval $[k\Delta h, (k+1)\Delta h[$ for $h_{ASI}$, where $j, k \in \{0, 1, 2, ..., N-1\}$, where N is the number of bins used for CBH in the analysis. A bin size $\Delta h = 100$ m is chosen in a trade-off between sources of error. Finer bins will allow to represent the distributions at higher resolution and will thus allow for higher resolved measurements of CBH in the network. However, the size of the used data set is limited which makes it difficult to model these distributions at highest resolution. The bin size chosen here is expected to limit the achievable uncertainty of

890 the measurement to a minimum level of 100 m. Joint frequency distributions modeled here are restricted to a maximum CBH of 12 km. This yields $N = 120$.

**A2    Filtering operations applied**

First, a weighted mean filter is applied between original joint frequency distributions $F_l$ received from all ASI-pairs with camera distance $d$, this yields $F_{l,filter\ 1}$:

$$F_{l,filter\ 1} = \frac{\sum_m w_{l,m} F_m}{\sum_m w_{l,m}}. \tag{A1}$$

For the joint frequency distribution $F_l$ of each respective ASI-pair $l$, weights $w_{l,m}$ are used that include ASI-pairs with similar camera distance. More precisely, a triangular window, based on the difference of camera distance $\Delta d_{l,m}$ of ASI-pair $m$ compared to ASI-pair $l$, is used that is defined by

$$w_{l,m} = max(0, 1 - \Delta d_{l,m}/0.5\ \text{km}). \tag{A2}$$

We decompose each distribution $F_{l,filter\ 1}$ by conditional filters into three separate *modes*. In the second step, we apply to each mode a Gaussian filter $g_\sigma$ with distinct standard deviation $\sigma_{mode}$ of the Gaussian kernel. The subscript *mode* indicates the specific mode for which $\sigma_{mode}$ is applied. The first mode is constituted by all outlier observations. Outliers are defined here as grid cells $(h_{Ref}, h_{ASI})$ for which ASI-pair measurement of CBH $h_{ASI}$ deviates by more than 1.5 km from the ceilometer reading $h_{Ref}$:

$$F_{l,outlier}(h_{Ref}, h_{ASI}) = \begin{cases} F_{l,filter\ 1}(h_{Ref}, h_{ASI}), & |h_{ASI} - h_{Ref}| > 1.5\ \text{km} \\ 0, & \text{else.} \end{cases} \tag{A3}$$

Such outliers will contain a large random component. We expect that in a reproduction of the experiment, a similar number of outliers will be received, while the joint frequency found for a single grid cell $(h_{Ref}, h_{ASI})$ may vary significantly. Therefore, the strongest filter is applied to this mode using $\sigma_{outlier} = 1$ km.

The second mode is constituted by grid cells that are not part of the first mode and feature a joint frequency less than the average over all grid cells of the joint frequency distribution:

$$F_{l,inconfident}(h_{Ref}, h_{ASI}) = \begin{cases} F_{l,filter\ 1}(h_{Ref}, h_{ASI}), & |h_{ASI} - h_{Ref}| \leq 1.5\ \text{km} \\ & \wedge F_{l,filter\ 1}(h_{Ref}, h_{ASI}) < \text{mean}(F_{l,filter\ 1}) \\ 0, & \text{else.} \end{cases} \tag{A4}$$

The comparably small number of observations in these grid cells is expected to cause an increased uncertainty of the estimated joint frequencies. For this mode, $\sigma_{inconfident} = 0.5$ km is applied.

**General comment 7:**

Reviewer:

The Authors list high temporal and spatial resolutions ("30 s or 5 m", l. 6) of state-of-the-art nowcasts. It is not obvious if chosen CBH intervals ("100m", l. 231) are fine enough to provide such high resolution. Perhaps the Authors could expand on this in Section 3.3 or in their discussion to address this question.

Authors' response:

We agree with the reviewer that indeed the specification of the used state-of-the-art ASI-pairs may appear contradictory to the accuracy of the CBH estimation attested in this study for all of the studied ASI-based CBH measurements. As suggested, we added a short discussion to give an

explanation why ASI-based nowcasts may be provided at a resolution which is by far finer than the deviations of cloud shadow positions induced by deviations in the estimation of CBH. The source cited in this discussion was additionally added to the introduction. Note that Sect. 4.4 was also reworked based on Reviewer Comment 2, Major Comment 2.

Changes in manuscript:

p. 2, ll. 37-38:

nowcasts can reduce the uncertainty of supply from solar power plants and can support efficient balancing of energy supply and demand (Law et al., 2014; Kaur et al., 2016). Further, they can be applied to control concentrating solar power plants (Nouri et al., 2020a) more efficiently. The coordination of renewable production and energy consumption at a local scale is a way to minimize requirements on grid-infrastructure while keeping curtailment of feed-ins from renewable sources at a low
40  level. Ghosh et al. (2016) use nowcasts (15 s ahead) to control PV-feed in and provide reactive power. In this context, spatially

p. 30, ll. 740-754:

740  statistics and from an inspection of the ASI images observations of higher CBH layers are likely to be found in the presence of a lower layer. As discussed in Sect. 3.1 the ASI-based estimation of CBH in this study. Meanwhile, the underlying ASI-pairs measure CBH of the most dominant cloud layer in the sense of optical thickness and area in the analyzed field of view of the sky. When for example, a CBH of 10 km is present the corresponding spatial area included has side lengths of 15.7 km. For multi-cloud-layer conditions it is likely that within this window lower clouds are present which are recognized by an ASI-pair
745  instead. More aggressive filtering of such multilayer situations included in the evaluation could reduce this influence but would further limit the database. The distance between the cameras used by an ASI-pair and the reference ceilometer are not found to have a significant influence on received accuracy in the evaluated data set. This was expected in part from the previously discussed effect that the can nowcast 30 s-averages of solar irradiance at a spatial resolution of 5 m × 5 m. According to the considerations of Nouri et al. (2019b) and with the sun elevations occuring at the site, deviations in CBH may cause deviations
750  in the positions of cloud shadow edges of at least 100 m under favorable conditions for the ASI-pairs measure CBH of the most dominant cloud layer and also for the ASI-network. This deviation is much larger than the spatial resolution of these maps of solar irradiance. For certain applications, e.g. to control solar power plants (Nouri et al., 2020a), it may still be advantageous to provide maps of solar irradiance at a resolution finer than the uncertainty of cloud shadow edge positions, as the statistical properties of spatial variability may still be captured in these maps.

**General comment 8:**
Reviewer:

To help the reader appreciate the scientific advance in the work, the Authors should stress wherever (in Section 3.3 or 3.4) new techniques were developed or combined.

Authors' response:

To emphasize the novelty of the method used in the study we added a short introduction to the MLE-based method in 3.4. Our combination method allows to combine the CBH measurements from a large number of ASI-pairs. Additionally, the method takes account of the individual characteristics of

the ASI-pairs by the use of conditional probabilities. Finally, the use of MLE is to the best of our knowledge not known to this application.

Further we also pointed out that the required distributions of conditional probability were so far not available for CBH measurement by ASI-pairs.

Changes in manuscript:

pp. 12-13, ll. 312-321:

>  In this section we present our method to combine the measurements of CBH from a large number ASI-pairs organized as network. Prior works estimated CBH by a small number of two or in some cases four ASIs (Nouri et al., 2019a). However, with a large number of ASI-pairs, we consider a statistical method
> 315 promising, which analyzes the CBH samples received and, based on the known characteristics of each ASI-pair, determines the CBH which is most likely to be present. The characteristics of each ASI-pair are in the following described by conditional probability distributions, which will by retrieved in Sect. 3.4. These distributions provide the probability of receiving a certain CBH reading from an ASI-pair, given that actually a specific reference CBH is present. Our estimation procedure then uses principles from Maximum Likelihood Estimation (MLE)  and modifies them for the specific case. To the best of our
> 320 knowledge, the usage of a statistical method and in particular one relying on conditional probability distributions is novel to the task of estimating CBH from the observations of a multitude of ASIs.

p. 17, ll. 409-410:

> for each range $\{0...0.1, 0.1...0.2, 0.2...0.3, ..., 11.9...12\}$ km of true CBH. Conditional probability distributions of this kind are
> 410 not available so far for ASI-pairs. Therefore, we aim to approximate them from the measurement data of a modelling period. Estimations of CBH from the available ASI-pairs and measurements from the ceilometer during the period 01 April 2019 to

**Minor comment 1:**
Reviewer:

Fig. 2: The plot seems to contain redundant information (by switching perspectives between two ASIs). The Authors could color code each perspective or exclude one redundant half.

Authors' response:

We adapted Fig 2 (left) and for consistency also Fig. 2 (right). As suggested, the plots now make it clearer which ASI-pairs' axes orientations and also distances were yielded by switching the used main camera.

Changes in manuscript:

p. 6, Fig. 2:

[Figure]

[Figure]

**Figure 2.** Frequency distribution of  the bearing angles of the ASI-pairs' axes in the set of available ASI-pairs (over north, left) and of available camera distances (right) resulting when arranging the seven ASIs in the urban area into 42 ASI-pairs (from each  2-ASI-tuple two different ASI-pairs result by switching the main camera, counts of ASI-pairs with switched main camera are marked orange, striped)

**Minor comment 2:**

Reviewer:

ll. 140-144: Please provide the minimum optical thickness for ceilometer detection.

Authors' response:

We requested this information from the manufacturer but did not receive a response yet. As requested by Reviewer Comment 2, Specific Comment 4, 9, we added a more detailed description of the algorithm used by the ceilometer and extended the description of how this reference instrument was validated in previous studies. We hope that this may be helpful to a possible reader. Otherwise, we hope that we can add this information in a response to be handed in later.

Changes in manuscript:

p. 6, ll. 137-141:

> The used ceilometer  of type Lufft CHM 15 k Nimbus  (firmware v0.747) is operated by DLR since 2018. CBH is measured by the manufacturer's *Sky Condition Algorithm* (Lufft, 2018) in the default configuration. Heese et al. (2010) specifies for a ceilometer of the same type, that full overlap of the laser's and the receiver's field of view is reached at a height of 1500 m.
> 140 However relying on an overlap correction, the manufacturer specifies a minimum CBH of down to 0 m. In this study the manufacturer's default minimum CBH of 45 m is used.

pp. 7-9, ll. 174-203:

Regarding the accuracy of  ceilometers in general,
de Haij et al. (2016) and Görsdorf et al. (2016) noted that there is no generally excepted, quantifiable definition of CBH, yet.
Further, due to a lack of reference measurements, benchmarks may typically focus on the consistency of CBH measurements
by different types of ceilometers. In a benchmark performed by Martucci et al. (2010), the measurement of a Vaisala CL31
ceilometer $CBH_{CL}$ showed a significant deviation from the reading $CBH_{CHM}$ of the instrument

used here. This trend was given by $CBH_{CL} = 160.315\,\mathrm{m} + 0.925 * CBH_{CHM}$. However, the measurement procedure, of the
instrument used here, was modified by firmware updates in the meantime. Görsdorf et al. (2016) presented results from a more
recent measurement campaign, CeiLinEx2015, which took place in 2015. In this experiment the measurements of six types of
ceilometers were compared. For stratus and stratocumulus clouds as well as for fog, deviations between the instruments of up
to 70 m were observed. For each of these conditions, the CHM 15 k, used here, provided the smallest measurements of CBH

in terms of mean deviation from the median of all tested instruments. More severe deviations of several kilometers between the
instrument types were observed during conditions with heavy rain.

In an acceptance test, de Haij et al. (2016) measured CBH by two CHM 15 k, by a Vaisala LD40 ceilometer, by a UV lidar
(Leosphere ALS450) and by visibility sensors mounted in various altitudes on a tower of 213 m height. For CBH of up to
200 m, the CHM 15 k typically measured a CBH 30...50 m smaller than the one of the LD40. However, the CHM 15 k was in
better agreement with the estimate based on visibility sensors. Görsdorf et al. (2016) and de Haij et al. (2016) suggest, that the
negative mean deviation of the CHM 15 k attested by all these studies, for clouds in the range $CBH < 3\,\mathrm{km}$, is mostly caused by
the manufacturers' algorithms to detect CBH from backscatter profiles. Whereas, according to the manufacturer (Lufft, 2018),
the CHM 15 k detects the rising edge of a backscatter peak that exceeds a threshold, other manufacturers' devices may rather
recognize the peak's maximum.

For the range of CBH in 3...12 km, an inspection of timeseries depicted by de Haij et al. (2016) indicates very good
agreement of the measurements from CHM 15 k and the UV lidar, used there. As a further test of de Haij et al. (2016),
performed at a resolution of 60 s, high clouds, detected by the UV lidar in a range of 6...7.5 km, were to be detected by
the CHM 15 k within a tolerance of ±3 classes in hh code (WMO Table 1677). This tolerance corresponds to a CBH-range
of ±1050 m centered around the discretized reference CBH. CHM 15 k was attested a probability of detection of $> 98\%$ and
a false alarm rate of 0%. Based on these studies, the accuracy of the reference instrument is expected to be adequate for the
range of $CBH < 3\,\mathrm{km}$ and also for the range of $CBH > 3\,\mathrm{km}$, a rather good performance of the instrument is indicated. The
experimental results of this study will in particular be compared to prior studies which used a ceilometer of the same type. This
is expected to avoid possible inconsistencies related to the used reference.

**Minor comment 3:**

Reviewer:

ll. 145-151: Is there a maximum solar zenith angle that limits CBH retrieval?

Authors' response:

The measurement of CBH by the ASI-pairs is in principle only limited by the illuminance of the scene.
In this study we included zenith angles smaller than 90 degree. We added a description on this. Note
that a further addition was made here based on Reviewer Comment 2, Specific comment 6.

Changes in manuscript:

p. 7, ll. 157-158:

155   avoiding redundant calculations. In this way, computational cost scales mostly linear with the number of ASIs used instead of with the number of ASI combinations so that execution in real time is possible. In total, including computation time, the estimation of CBH by the ASI network can be retrieved within 10 s after image acquisition. CBH is computed by the ASI-pairs and by the ASI network during daytime, i.e. if the sun elevation at the time of image acquisition is greater than $0°$.

**Minor comment 4:**

Reviewer:

l. 171-173: Please substitute "most dominant in features, driven by area and optical thickness" instead of "most dominant in the sense of area and optical thickness".

Authors' response:

We appreciate the reviewer's suggestion and replaced the term accordingly.

Changes in manuscript:

p. 11, ll. 270-271:

270   measurement approximates the median CBH of the cloud layer that is locally most dominant in  features, driven by area and optical thickness.

**Minor comment 5:**

Reviewer:

l. 193-195: Please link to reference or plot(s) or else put "not shown".

Authors' response:

We added this note as suggested.

Changes in manuscript:

p.12, l. 292:

290    Within the range of high clouds, a roll-off of the frequency is seen for $CBH > 10$ km. A reliable estimation of CBH should therefore provide accurate readings for the range of $CBH \in ]0, 12[$ km.

A visual analysis and a k-means classification for the site of Oldenburg (not shown) suggested that local conditions predom-

**Minor comment 6:**

Reviewer:

Equ. 1: What is "j"?

Authors' response:

We would like to apologize for the mistake, the letter was indeed not intended. We corrected the equation.

Changes in manuscript:

Appendix, p. 36, l. 895:

First, a weighted mean filter is applied between original joint frequency distributions $F_l$ received from all ASI-pairs with camera distance $d$, this yields $F_{l, filter\ 1}$:

$$\qquad F_{l, filter\ 1} = \frac{\sum_m w_{l,m} F_m}{\sum_m w_{l,m}}. \qquad\qquad (A1)$$

For the joint frequency distribution $F_l$ of each respective ASI-pair $l$, weights $w_{l,m}$ are used that include ASI-pairs with similar

**Minor comment 7:**

Reviewer:

Fig. 8: Please provide performance metrics (e.g., correlation coefficient, bias, and RMSD) to each panel.

Authors' response:

We added performance metrics to all scatter-density plots shown in the publication. We further adapted the scatter-density plots as Sect. 4 was reworked based on Reviewer Comment 2, Major Comment 2.

Changes in manuscript:

p. 24, Fig. 8:

[Figure]

**Figure 8.** Relative frequency of ASI-based CBH estimation for given CBH from ceilometer. Evaluation for two of the ASI-pairs DON-MAR (upper row, left) and UOL-HOL (upper row, right) with respective camera distances of 0.8 and 5.7 km, and from the ASI network without (bottom row, left) and with refinements (bottom row, right). Relative frequency in each column adds up to 1. Additionally, median (50%-quartile, red dotted), limits to the interquartile range (IQR, red dashed) and 5−,95−percentiles (red solid line) based on floating 1000 m-bins of CBH from ceilometer are plotted.

**Authors' response to Reviewer Comment 2**

Reviewer:

The manuscript presents an interesting use of a network of all-sky-imagers (ASIs) to derive mean cloud-base-height over a wide area. The method presented is interesting and, overall, the proposed system seems robust of probable practical use. The authors offer practical suggestions about the optimal layout of future ASIs installations, thus providing some useful information to the user. Reading the manuscript, it is clear that a lot of interesting work has been done, but unfortunately this has not been distilled enough yet to be clearly presented to the scientific community. The new algorithm is poorly presented, the novel contributions are not clearly identified, and the discussion of the results lacks focus. The authors should drastically revise the manuscript, trying to clearly present the essence and motivation of their work and separate it from implementation details.

To my understanding, there are three technical aspects presented: a) Implementation of three different approaches to calculate CBH from a pair of ASIs b) Evaluation of CBH retrievals from ASI-pairs. c) The use of a network of multiple ASI-pairs to derive a robust CBH estimate for the region.

Each of these aspects should be discussed and evaluated one by one, or references should be given in studies evaluating their performance. Otherwise, the reader cannot properly interpret the results.

Authors' response:

We really appreciate the reviewer's time and effort spent on reviewing our manuscript, their insightful comments and suggestions. We have addressed all comments and incorporated the suggestions as good as possible to us and we believe, these changes led to valuable improvements of our manuscript. In particular, we have strongly revised the sections on modelling and validation.

In the following, we will address the reviewer's comments point-by-point. Changes to the manuscript are extracted from the adapted manuscript within which changes were highlighted using latexdiff. Blue indicates insertions, red indicates deletions. Please note, that the order of Sect. 3.3 and Sect. 3.4 has been reversed as suggested by Reviewer Comment 1, General Comment 5. This change has been excluded from the markup, as it would have obscured all other changes.

Changes in manuscript:

See below.

**Major comment 1**
Reviewer:

Sections 3.3 and 3.4 should be rewritten. The sections seem like a direct translation of computer code into words, with no effort to describe why each step was implemented, what is essential, and

what is just an implementation detail or even an experiment that happened to work. E.g. why use the three-gaussian filters? Why use the specific σ thresholds? Why add an offset of 0.5 in low frequency bins (why not 0.01 or 1)? Implementation details could be even moved into an appendix.

Authors' response:

Based on the reviewer's feedback we revised Sect. 3.3 and 3.4 drastically. Especially regarding the modelling of conditional probability distributions, we moved the exact description of the procedure to the appendix and focused more on describing the idea behind the procedure and every filter. We further gave a reason for the value assigned to each of the parameters. However, we also stated that these parameters may still be optimized in a future work and are so far only rough approximations.

Similarly, we reworked Sect. 3.3. In particular, we focused on pointing out for each step of the procedure, what the intention of each equation/ calculation step was.

Changes in manuscript:

Sections 3.3 and 3.4: pp. 12-20

**3.3 Estimating CBH in the ASI network (ORDER OF SECTIONS 3.3 AND 3.4 WAS EXCHANGED)**

 In this section we present our method to combine the measurements of CBH from a large number ASI-pairs organized as network. Prior works estimated CBH by a small number of two or in some cases four ASIs (Nouri et al., 2019a). However, with a large number of ASI-pairs, we consider a statistical method
315    promising, which analyzes the CBH samples received and, based on the known characteristics of each ASI-pair, determines

[Figure]

**Figure 5.** Inference procedure — Step 1: For each range $i$ of camera distance $CBH_i$ is computed as mean CBH from the respective ASI-pairs. Conditional probability is evaluated that $CBH_i$ would be received if true CBH (at the ceilometer) took on a value $\{0...0.1, 0.1...0.2,...,11.9...12\}$ km (red boxes). Step 1 yields a likelihood function for each range of camera distance. Step 2: Cumulative and complementary cumulative likelihood are calculated for each range of camera distance. Step 3: These functions are logarithmized and then summed over all ranges $i$ of camera distance yielding overall cumulative and complementary cumulative likelihood. Step 4: The Intersection of both functions gives the estimated likeliest CBH.

the CBH which is most likely to be present. The characteristics of each ASI-pair are in the following described by conditional probability distributions, which will by retrieved in Sect. 3.4. These distributions provide the probability of receiving a certain CBH reading from an ASI-pair, given that actually a specific reference CBH is present. Our estimation procedure then uses principles from Maximum Likelihood Estimation (MLE)  and modifies them for the specific case. To the best of our knowledge, the usage of a statistical method and in particular one relying on conditional probability distributions is novel to the task of estimating CBH from the observations of a multitude of ASIs.

To give an overview, Fig. 5 shows the inference process used to estimate CBH by the network based on the 42 CBH readings provided by the individual ASI-pairs. For each range $i$ of camera distance,  in Sect. 3.4, conditional probability distributions will be estimated. These conditional probabilities are translated into the likelihood that actually certain values of (reference) CBH are present (step 1) based on the readings of CBH received  from ASI-pairs in this range $i$ of camera distance. After calculating the cumulative likelihood for each range of camera distance (step 2), these are combined yielding the overall cumulative and complementary cumulative likelihood from all  ASI-pairs (step 3). Finally, the value of CBH which is most likely to be present at the site and at the evaluated time, given the readings from all involved ASI-pairs, is estimated (step 4). These steps are  presented in more detail in the following.

Step 1: For each ASI-pair, the median value of all valid CBH readings of the previous 10 min is calculated. If an ASI-pair does not provide any valid CBH within this period, it is excluded from the prediction for the instance in time evaluated. The ranges of camera distance $1...2.5$ km and $3...4$ km are represented by a larger number of ASI-pairs than the remaining distances.  Thus, the readings of ASI-pairs in these ranges of camera distance may prevail in the estimation of CBH. As the variety of camera distances is considered to bring a benefit to the procedure, we intend to represent all camera distances as uniformly as possible. For this, we define ranges of camera distance , using the range limits $\{0.5, 1, 1.5, ..., 6\}$ km . CBH readings of all ASI-pairs with camera distance in range $i$ are averaged to yield $\text{CBH}_i$. Consecutively, the conditional probability  $P(\text{CBH}_i \mid h_{true})$ is evaluated that the found $\text{CBH}_i$ would be received for a given true CBH  $h_{true}$ (red marked box prior to step 1 in Fig. 5). Note that  $P(\text{CBH}_i \mid h_{true})$ will be modeled in Sect. 3.4 measuring CBH $h_{Ref}$ by a ceilometer which  provides $h_{Ref} \approx h_{true}$. Thus, the likelihood  $\mathcal{L}_i(h_{true})$ is obtained (Fig. 5, output of step 1):

$$\mathcal{L}_i(\underline{\theta} h_{true}) = P(\text{CBH}_i \mid \underline{\theta} h_{true}). \tag{1}$$

Step 2:  We define cumulative likelihood $\mathcal{C}_i(\hat{h}_{true})$ as the likelihood of receiving the present reading $\text{CBH}_i$ given that $h_{true}$ is smaller or equal to an estimation of true CBH $\hat{h}_{true}$. Accordingly in the implementation, likelihood is summed cumulatively over all bins of reference CBH  $h_{true}$ (Fig. 5, step 2):

$$\mathcal{C}_i(\hat{h}_{true}) = \sum_{\underline{\theta \leq \hat{\theta}} h_{true} \leq \hat{h}_{true}} \mathcal{L}_i(\underline{\theta} h_{true}). \tag{2}$$

Likewise, a complementary cumulative likelihood is defined

$$\bar{\mathcal{C}}_i(\hat{\theta}) = \sum_{\theta > \hat{\theta}} \mathcal{L}_i(\theta).$$

as the likelihood of receiving the present reading $CBH_i$ given that $h_{true}$ is greater than an estimation of true CBH $\hat{h}_{true}$:

$$\bar{\mathcal{C}}_i(\hat{h}_{true}) = \sum_{h_{true} > \hat{h}_{true}} \mathcal{L}_i(h_{true}). \tag{3}$$

350 $\mathcal{C}_i(\hat{\theta})$ and $\bar{\mathcal{C}}_i(\hat{\theta})$ are used here as measures how likely it is that actual CBH $\theta$ is in the interval $]0$ km, $\hat{\theta}]$ or $]\hat{\theta}, 12$ km$]$ respectively. It is mainly the In particular, the use of these cumulative functions that and the estimation of likelihood functions from measurement data distinguishes the present approach from a regular Maximum-Likelihood-Estimation (MLE). This modification is used as in MLE typically smooth analytical  functions are assumed as likelihood function. In contrast, likelihood functions here  will be estimated based on empirical conditional probabilities. These approximated likelihood-

355 functions, derived from a dataset of finite size, may therefore be less smooth and may not be completely  representative. When using cumulative distributions, it is expected that the method still works robustly if the conditional probabilities are not estimated accurately for each grid cell of the discrete distribution if at least the cumulative value over a range of CBH is appropriate. In spite of the modification, the presented approach may adopt beneficial properties of MLE: The use of appropriate conditional probabilities ( determined in Sect. 3.4) reduces systematic deviations of estimated

360 CBH compared to the measurement of a single ASI-pair. Moreover, applied conditional probabilities are in general not specific to the studied site and its meteorological conditions which allows to apply the method at other sites.  Both functions $\mathcal{C}_i(\hat{h}_{true})$ and $\bar{\mathcal{C}}_i(\hat{h}_{true})$ are shown for three exemplary intervals of camera distance in Fig. 5 as output

365 of step 2.

Step 3:  We aim to determine the likelihood of receiving the combination of readings $CBH_i$ from all the intervals $i$ of camera distance  given that $h_{true} \leq \hat{h}_{true}$. This can be expressed as product of $\mathcal{C}_i(\hat{h}_{true})$ from all intervals $i$. As this product would often become zero in our numerical treatment, we instead calculate its natural logarithm, which we refer to as overall logarithmized cumulative likelihood $\log \mathcal{C}_n(\hat{h}_{true})$. This

370 operation also allows to replace the product by a sum (Fig. 5, step 3):

$$\log \mathcal{C}_n(\hat{h}_{true}) = \sum_i \log \mathcal{C}_i(\hat{h}_{true}). \tag{4}$$

Analogously, an overall complementary logarithmized cumulative likelihood is computed given all readings $CBH_i$ per interval $i$ of camera distance

375 $$\log \bar{\mathcal{C}}_n(\hat{h}_{true}) = \sum_i \log \bar{\mathcal{C}}_i(\hat{h}_{true}). \tag{5}$$

Both functions are visualized exemplarily as output of step 3 in Fig. 5.

Step 4:  $\log \mathcal{C}_n(\hat{h}_{true})$ and $\log \bar{\mathcal{C}}_n(\hat{h}_{true})$ are only known at discrete points. Linear interpolation yields continuous representations of these.  Then finally, we aim to select the true CBH $h_{likeliest}$, which makes it likeliest to receive the given combination of CBH$_i$. In our formulation of the problem, this means we intend to find a $\hat{h}_{likeliest}$ which simultaneously maximizes $\log \mathcal{C}_n(\hat{h}_{true})$ and $\log \bar{\mathcal{C}}_n(\hat{h}_{true})$. Consequently, we accept $h_{likeliest}$, for which $\log \mathcal{C}_n(\hat{h}_{true})$ and $\log \bar{\mathcal{C}}_n(\hat{h}_{true})$ are equal (Fig. 5, step 4):

$$\underline{\theta}h_{likeliest} = \operatorname*{argmin}_{\hat{\underline{\theta}}\hat{h}_{true}} \left| \log \bar{\mathcal{C}}_n(\hat{h}_{true}) - \log \mathcal{C}_n(\hat{h}_{true}) \right|. \tag{6}$$

Besides this estimation of CBH, a version of this procedure will be discussed that includes further refinements (in the following referred to as *refined* estimation).  As a first observation from the generation of conditional probabilities, ASI-pairs  with camera distance greater than 4.5 km  cause large deviations for CBH < 4 km and  exhibit only a moderate advantage at greater CBH.  These ASI-pairs are excluded from the refined estimation of $h_{likeliest}$. On the other hand, ASI-pairs with  small camera distance are already accurate if only small CBH occur, as we will discuss in Sect. 4. We inspected conditional probabilities of the ASI-pairs  (exemplarily viewed as input to step 1 in Fig. 5)  and identified the ASI-pairs which are most appropriate for an interval of CBH. Based on this, the refined estimation is  received from the arithmetic average of CBH measured by ASI-pairs with corresponding small camera distance, if the first iteration of $h_{likeliest}$ yielded a sufficiently small CBH. In summary, the refinement procedure to receive the final estimation of CBH  $h_{refined}$ reads

$$\underline{\theta}h_{refined} \begin{cases} h_{likeliest}, & h_{likeliest} \in ]3,12] \text{ km} \\ \min(3 \text{ km}, \text{mean}(h_{i \in \{i | d_i < 1.6 \text{ km}\}})), & h_{likeliest} \leq 3 \text{ km} \wedge \text{mean}(h_{i \in \{i | d_i < 1.6 \text{ km}\}}) > 1.5 \text{ km} \\ \min(1.5 \text{ km}, \text{mean}(h_{i \in \{i | d_i < 1.2 \text{ km}\}})), & h_{likeliest} \leq 3 \text{ km} \wedge \text{mean}(h_{i \in \{i | d_i < 1.6 \text{ km}\}}) \leq 1.5 \text{ km}. \end{cases} \tag{7}$$

**3.4 Estimation of conditional probabilities of CBH (ORDER OF SECTIONS 3.3 AND 3.4 WAS EXCHANGED)**

The procedure to combine CBH-measurements from independent ASI-pairs, which are organized as a network, requires knowledge of the (conditional) probability to receive a certain reading of CBH from an ASI-pair given the true CBH takes on some specific value. The  required distribution aims to answer the following question: If true CBH ranges in between 1.8...1.9 km, how large will be the probability that an ASI-pair with camera distance 2.2 km delivers a certain CBH e.g. within 0...0.1 km or 1.8...1.9 km or 11.9...12 km? In the following, these conditional probabilities are estimated not only for the range of true CBH between 1.8...1.9 km but

for each range $\{0...0.1, 0.1...0.2, 0.2...0.3, ..., 11.9...12\}$ km of true CBH. Conditional probability distributions of this kind are
410    not available so far for ASI-pairs. Therefore, we aim to approximate them from the measurement data of a modelling period. Estimations of CBH from the available ASI-pairs and measurements from the ceilometer during the period 01 April 2019 to 29 June 2019 are used. CBH measured by the ceilometer serves as reference CBH. It is considered not to be essential that the training period is representative of the period to which the method is applied. However, we expect that the method works best if the included ASI-pairs exhibit a similar distribution of measurement deviations given the same reference CBH in both
415    periods. For solar applications and the latitude of this study, we consider the used dataset and its split reasonable. The summer and shoulder months provide the main share of the annual solar yield at the site and are therefore in the focus of the nowcasting system under development. In that sense, the training dataset is considered to be for the large part representative of conditions relevant to solar applications at similar latitudes.

   The seven ASIs available in the urban area are arranged into 42 ASI-pairs. Each tuple of two ASIs, that is selected from the
420    set of seven ASIs, yields 2 independent ASI-pairs by swapping the ASI used as main camera (see Sect. 3.1).

   The procedure is developed based on periods in which valid measurements from ceilometer and the respective ASI-pair are available and in which the variability of CBH is moderate: For each time stamp a window of 30 min centered at this time stamp is defined. A time stamp is only included if standard deviation of reference CBH within the window is less than 30% of the mean value of reference CBH within the same window. As discussed before, ASI-pairs and ceilometer measure CBH
425    as spatial median and point-wise respectively. Therefore, this filter intends to assure that ceilometer and ASI-pair measure CBH of the same layer. CBH from the respective ASI-pair and from the ceilometer are processed by a moving-median filter with a window of 10 min. The joint frequency distribution of CBH measured by ceilometer $h_{Ref}$ and the respective ASI-pair $h_{ASI}$ is computed from these simultaneously acquired time series.  In other words, the domain of reasonable values, $[0, 12 \text{ km}[\times[0, 12 \text{ km}[$, which the pair $(h_{Ref}, h_{ASI})$ can take on, is discretized into a mesh of square grid cells with
430    side lengths $\Delta h$. Then the frequency is calculated with which $(h_{Ref}, h_{ASI})$ is observed in  each of the discrete grid cells. A bin size $\Delta h = 100$ m is chosen in a trade-off between sources of error. Finer bins will allow to represent the distributions at higher resolution and will thus allow for higher resolved measurements of CBH in the network. However, the size of the used data set is limited which makes it difficult to model these distributions at highest resolution. The bin size
435    chosen here is expected to limit the achievable uncertainty of the measurement to a minimum level of 100 m.

   Joint frequency distributions were inspected and found to be well reproduced among the studied independent ASI-pairs, if only the corresponding camera distances are similar. This meets the expectation from literature discussed in Sect. 3.1. Moreover, we conclude that the distributions modeled here will be transferable to other setups that use camera distances in the
440    studied range. Local climate is expected to influence the transferability to a minor extent  .

   The limited size and representativeness of the data set used in model development are expected to cause random features in the joint frequency distributions which are not useful to the estimation procedure, when it is applied to other setups, sites and

times  (such as represented by the validation data set). To suppress such random features of received joint frequency distributions~~. For this , the original joint frequency distribution $F_l$ of ASI-pair $l$ is transformed by a first filter into $F_{l,filter~1}$ and by a consecutively applied filter into $F_{l,filter~2}$.~~, we introduce a filtering procedure with two consecutive steps described here and in more detail in Appendix A. The parameter values set in the filtering procedure are approximate to this point and are based on a visual comparison of unfiltered and filtered distributions, evaluating the degree to which noise but also reasonable features were suppressed. The parameters values may be optimized in a future study.

First, a weighted mean filter is applied between the original joint frequency distributions  received for ASI-pairs with $d$

$$F_{l,filter~1} = \frac{\sum_j w_{l,m} F_m}{\sum_j w_{l,m}}.$$

 similar camera distance. As discussed above, ASI-pairs with similar camera distance

$$w_{l,m} = max(0, 1 - \Delta d_{l,m}/0.5 \text{ km}).$$

 are expected to perform similarly in the measurement of CBH and should consequently also exhibit similar joint frequency distributions of CBH. Thus, the filter aims to suppress differences between the joint frequency distributions of ASI-pairs which may result from disturbances in the estimation rather than from a difference in the systems' characteristics.

To each filtered distribution resulting from the prior step, a composite of three Gaussian filters is applied~~to $F_{l,filter~1}$ of each ASI-pair $l$~~. We first decompose each distribution $F_{l,filter~1}$ by conditional filters into three separate *modes*, which correspond to parts of the joint frequency distributions which are estimated with descending precision. Thereafter, we apply  a Gaussian filter $g_\sigma$  to each mode. The standard deviation of the Gaussian  filter applied to each mode corresponds qualitatively to the uncertainty with which the prior joint frequency distribution is estimated within grid cells of that mode. Consequently, the three filtered modes are summed to receive the smoothened joint frequency distribution.

The first mode is constituted by  grid cells for which the ASI-pair based measurement of CBH  deviates by more than 1.5 km from the ceilometer reading:

$$F_{l,outlier}(h_{Ref}, h_{ASI}) = \begin{cases} F_{l,filter~1}(h_{Ref}, h_{ASI}), & |h_{ASI} - h_{Ref}| > 1.5 \text{ km} \\ 0, & \text{else.} \end{cases}$$

. The large deviations represented by this mode occur less frequently which is why the joint frequency distribution will be estimated less precisely for the respective grid cells. On the other hand, apart from such

scattering effects, the joint frequency   distributions are found to be comparably smooth in the grid cells of this mode. A Gaussian filter with a large standard deviation of 1 km is applied to this mode  which is considered to be apt to preserve the expected distribution while suppressing random features.

The second mode is constituted by grid cells  for which the ASI-pair based measurement of CBH deviates by less than 1.5 km from the ceilometer reading and which feature a joint frequency   below the average of all grid cells

$$F_{l,inconfident}(h_{Ref}, h_{ASI}) = \begin{cases} F_{l,filter\ 1}(h_{Ref}, h_{ASI}), & |h_{ASI} - h_{Ref}| \leq 1.5 \text{ km} \\ & \wedge\ F_{l,filter\ 1}(h_{Ref}, h_{ASI}) < \text{mean}(F_{l,filter\ 1}) \\ 0, & \text{else.} \end{cases}$$

. These grid cells typically exhibit a larger joint frequency, i.e. more observations, than grid cells in the first mode. Still the comparably small number of observations in these grid cells is expected to cause an increased uncertainty of the estimated joint frequencies.  Consequently in a trade-off between suppressing random scattering and preserving meaningful variations a Gaussian filter with standard deviation 0.5 km is applied.

The third mode  makes up the complementary of the first and second mode. It contains grid cells that are observed with an at least average joint frequency and which are not classified as outliers

$$F_{l,confident}(h_{Ref}, h_{ASI}) = \begin{cases} F_{l,filter\ 1}(h_{Ref}, h_{ASI}), & |h_{ASI} - h_{Ref}| \leq 1.5 \text{ km} \\ & \wedge\ F_{l,filter\ 1}(h_{Ref}, h_{ASI}) \geq \text{mean}(F_{l,filter\ 1}) \\ 0, & \text{else.} \end{cases}$$

. Joint frequencies in these grid cells are considered to  be estimated with a comparably high accuracy. To avoid a loss of precision and ultimately a loss of accuracy in the estimation of CBH, a  Gaussian filter with a standard deviation of 0.1 km is used.

$$F_{l,filter\ 2} = g_{\sigma_{outlier}}(F_{l,outlier}) + g_{\sigma_{inconfident}}(F_{l,inconfident}) + g_{\sigma_{confident}}(F_{l,confident}).$$

Hence, only neighboring grid cells have a significant influence on this filter.

In many joint frequency distributions, there are grid cells with joint frequency close to zero. Especially for these grid cells, a greater  data set would be required to receive more representative values. For all grid cells, joint frequency is increased to a minimum value of 0.5 to avoid underestimations of joint frequency. This value corresponds to half of the joint frequency associated with a single actual observation in a grid-cell. For the estimation procedure of CBH,  such a minimum value leads to slightly reduced precision for most readings but increased robustness in the case that these grid cells $(h_{Ref}, h_{ASI})$ are indeed observed in the measurement.

 Finally, from each joint frequency distribution, the conditional probability $P(h_{ASI} \mid h_{Ref})$ to receive a certain CBH reading from an ASI-pair, given that the ceilometer measures some certain CBH, is

505 ~~dividing the respective probability mass function by the marginal distribution of CBH measured by the ceilometer. The latter distribution gives the probability to receive CBH from the ceilometer within a certain bin $h_{Ref}$ regardless of which CBH reading is simultaneously received from an ASI-pair. The distrbution can be derived from any of the probability mass functions by summing all grid cells of the probability mass function which correspond to the respective bin $h_{Ref}$ of CBH measured by the ceilometer.~~ derived (see Appendix A for a more detailed description).

510 ~~Inference procedure — Step 1: For each range $i$ of camera distance $CBH_i$ is computed as mean CBH from the respective ASI-pairs. Conditional probability is evaluated that $CBH_i$ would be received if true CBH (at the ceilometer) took on a value $\{0...0.1,0.1...0.2,...,11.9...12\}$ (red boxes). Step 1 yields a likelihood function for each range of camera distance. Step 2: Cumulative and complementary cumulative likelihood are calculated for each range of camera distance. Step 3: These functions are logarithmized and then summed over all ranges $i$ of camera distance yielding overall cumulative and complementary~~

515

The inference procedure, which  was introduced in Sect. 3.3, represents each range $i$ of camera distance bounded by the limits $\{0.5,1,1.5,...,6\}$ km by a single distribution of conditional probability. For each range of camera distance, the distribution of conditional probability, which corresponds to the camera distance closest to the center of this range, is selected

520  (example provided in Appendix A). Figure 5 (above Step 1) shows exemplary conditional probabilities for three ASI-pairs with camera distances 0.8, 2.2, 5.7 km representing the ranges of camera distance $i=1$, 4, 11 respectively. BIAS and precision, with which ASI pairs of distinct camera distances measure CBH, given a

525 certain reference CBH, are visible in these conditional probabilities. Such characteristics will be evaluated in more detail in the following, based on a separate validation data set.
* * *
Appendix A, pp. 35-37:

880 **Appendix A: Details on the retrieval of conditional probabilities**

**A1 Retrieval of raw joint frequency distributions**

CBH from the respective ASI-pair and from the ceilometer are processed by a moving-median filter with a window of $10\,\text{min}$. The joint frequency distribution of CBH measured by ceilometer $h_{Ref}$ and the respective ASI-pair $h_{ASI}$ is computed from these simultaneously acquired time series. That means, the frequency is calculated with which $(h_{Ref}, h_{ASI})$ is observed in

885 a discrete grid cell defined by the interval $[j\Delta h, (j+1)\Delta h[$ for $h_{Ref}$ and the interval $[k\Delta h, (k+1)\Delta h[$ for $h_{ASI}$, where $j,k \in \{0,1,2,...,N-1\}$, where N is the number of bins used for CBH in the analysis. A bin size $\Delta h = 100\,\text{m}$ is chosen in a trade-off between sources of error. Finer bins will allow to represent the distributions at higher resolution and will thus allow for higher resolved measurements of CBH in the network. However, the size of the used data set is limited which makes it difficult to model these distributions at highest resolution. The bin size chosen here is expected to limit the achievable uncertainty of

890 the measurement to a minimum level of $100\,\text{m}$. Joint frequency distributions modeled here are restricted to a maximum CBH of 12 km. This yields $N=120$.

**A2 Filtering operations applied**

First, a weighted mean filter is applied between original joint frequency distributions $F_l$ received from all ASI-pairs with camera distance $d$, this yields $F_{l,filter\,1}$:

$$F_{l,filter\,1} = \frac{\sum_m w_{l,m} F_m}{\sum_m w_{l,m}}.$$ (A1)

For the joint frequency distribution $F_l$ of each respective ASI-pair $l$, weights $w_{l,m}$ are used that include ASI-pairs with similar camera distance. More precisely, a triangular window, based on the difference of camera distance $\Delta d_{l,m}$ of ASI-pair $m$ compared to ASI-pair $l$, is used that is defined by

$$w_{l,m} = max(0, 1 - \Delta d_{l,m}/0.5 \text{ km}).$$ (A2)

We decompose each distribution $F_{l,filter\,1}$ by conditional filters into three separate *modes*. In the second step, we apply to each mode a Gaussian filter $g_\sigma$ with distinct standard deviation $\sigma_{mode}$ of the Gaussian kernel. The subscript *mode* indicates the specific mode for which $\sigma_{mode}$ is applied. The first mode is constituted by all outlier observations. Outliers are defined here as grid cells $(h_{Ref}, h_{ASI})$ for which ASI-pair measurement of CBH $h_{ASI}$ deviates by more than 1.5 km from the ceilometer reading $h_{Ref}$:

$$F_{l,outlier}(h_{Ref}, h_{ASI}) = \begin{cases} F_{l,filter\,1}(h_{Ref}, h_{ASI}), & |h_{ASI} - h_{Ref}| > 1.5 \text{ km} \\ 0, & \text{else.} \end{cases}$$ (A3)

Such outliers will contain a large random component. We expect that in a reproduction of the experiment, a similar number of outliers will be received, while the joint frequency found for a single grid cell $(h_{Ref}, h_{ASI})$ may vary significantly. Therefore, the strongest filter is applied to this mode using $\sigma_{outlier} = 1$ km.

The second mode is constituted by grid cells that are not part of the first mode and feature a joint frequency less than the average over all grid cells of the joint frequency distribution:

$$F_{l,inconfident}(h_{Ref}, h_{ASI}) = \begin{cases} F_{l,filter\,1}(h_{Ref}, h_{ASI}), & |h_{ASI} - h_{Ref}| \leq 1.5 \text{ km} \\ & \wedge F_{l,filter\,1}(h_{Ref}, h_{ASI}) < \text{mean}(F_{l,filter\,1}) \\ 0, & \text{else.} \end{cases}$$ (A4)

The comparably small number of observations in these grid cells is expected to cause an increased uncertainty of the estimated joint frequencies. For this mode, $\sigma_{inconfident} = 0.5$ km is applied.

The third mode $F_{l,confident}(h_{Ref}, h_{ASI})$ makes up the complementary of the first and second mode. It contains grid cells that are observed with an at least average joint frequency and which are not classified as outliers:

$$F_{l,confident}(h_{Ref}, h_{ASI}) = \begin{cases} F_{l,filter\ 1}(h_{Ref}, h_{ASI}), & |h_{ASI} - h_{Ref}| \leq 1.5 \text{ km} \\ & \wedge F_{l,filter\ 1}(h_{Ref}, h_{ASI}) \geq \text{mean}(F_{l,filter\ 1}) \\ 0, & \text{else.} \end{cases} \qquad (A5)$$

Joint frequencies in these grid cells are considered to have a comparably high accuracy. To avoid a loss of precision and ultimately a loss of accuracy in the estimation of CBH, a small value of $\sigma_{confident} = 0.1$ km is used. The three filtered modes $g_\sigma$ are summed to receive the smoothened joint frequency distribution

$$F_{l,filter\ 2} = g_{\sigma_{outlier}}(F_{l,outlier}) + g_{\sigma_{inconfident}}(F_{l,inconfident}) + g_{\sigma_{confident}}(F_{l,confident}). \qquad (A6)$$

For all grid cells, joint frequency is increased to a minimum value of 0.5 to avoid underestimations of joint frequency. This value is chosen to be half of the joint frequency associated with a single actual observation in a grid-cell.

Each joint frequency distribution is normalized with the sum of all joint frequency grid cells. In this way, a probability mass function $P(h_{Ref}, h_{ASI})$ (also known as discrete density function) to measure a certain CBH with the respective ASI-pair and to coincidentally measure a certain CBH with the ceilometer is yielded. The conditional probability $P(h_{ASI} | h_{Ref})$ to receive a certain CBH reading from an ASI-pair, given that the ceilometer measures some certain CBH, is calculated by dividing the respective probability mass function by the marginal distribution of CBH measured by the ceilometer. The latter distribution gives the probability to receive CBH from the ceilometer within a certain bin $h_{Ref}$ regardless of which CBH reading is simultaneously received from an ASI-pair. The distribution can be derived from any of the probability mass functions by summing all grid cells of the probability mass function which correspond to the respective bin $h_{Ref}$ of CBH measured by the ceilometer.

**A3 Representation of intervals of camera distance**

The inference procedure represents each range $i$ of camera distance bounded by the limits $\{0.5, 1, 1.5, ..., 6\}$ km by a single distribution of conditional probability. For each range of camera distance, the distribution of conditional probability, which corresponds to the camera distance closest to the center of this range, is selected. For example, for the range $i = 2$ representing camera distances $1...1.5$ km, the center of the range would be 1.25 km. For the camera distances 1.081, 1.247 and 1.352 km, conditional probabilities have been modeled. Consequently, for this range of camera distance, the distribution of conditional probability corresponding to the camera distance 1.247 km is used.

**Major comment 2**
Reviewer:

A similar comment goes also for the discussion part: It should be made much more concise, focusing on key results. Moreover, the stated aim of the proposed method is to assist nowcasting, and thus the authors should add an evaluation of the single measurement accuracy of the network. I.e. if the network outputs a CBH value of h, what is the uncertainty of this estimate? It is good that the network shows small overall biases in a three month period, but it wouldn't be of much use if the correct CBHs were measured at the wrong times.

Authors' response:

We reworked the validation part strongly, intending most of all to focus on the key results. Still, we also needed to add passages at some points as further discussions or clarifications were suggested by the reviewers.

As suggested, we included an additional subsection which evaluates the accuracy of an ASI-pair and of the ASI network for the nowcasting application as suggested above. We agree with the reviewer that this is an interesting aspect, which attests a certain advantage of the ASI network for this application.

In particular, we made the following larger changes to the discussion part (Sect. 4):

- The behavior of the ASI network during mostly clear periods has been detailed and described more precisely.
- The discussion of exemplary time series of CBH has been limited to a single day. Descriptions and visualizations of 06 August 2019 have been moved to Appendix B.
- To give the reader a faster overview of the results (and also based on Reviewer Comment 1, Minor Comment 7) scatter-density plots have been enhanced to include performance metrics and quantiles
- Discussion of minimum CBH has been condensed and has been placed in Sect. 4.2.1 also relating it to the expectation from geometry.
- Sect 4.3, evaluating accuracy in a nowcasting context, was added
- Based on Reviewer Comment 1, General comment 3, we described which portion of the validation data set was filtered out
- Based on Reviewer Comment 1, General comment 7, we discussed the relationship between the accuracy of CBH and the resolution of irradiance maps created by a nowcasting procedure.
- We merged the prior "Sect. 4.4" with the present Sect. 4.4, intending to shorten discussions as far as possible.

We hope to have found a reasonable trade-off regarding the length of this section. As another measure to shorten the discussion part, Sect. 4.1, which analyzes time series of CBH from the different sources, could still be moved to the Appendix. This section is majorly intended to give the reader concrete examples of the effects discussed thereafter by statistical tools.

Changes in manuscript:

pp. 20-34

**4 Validation of CBH measurement by the ASI network and comparison to CBH measurements by the ASI-pairs**

In this section, the accuracy of CBH measurement by the ASI network and by 42 independent ASI-pairs set up at a wide variety of camera distances and alignments is compared. This section is based on a validation data set including the days 530 from 30 June 2019 to 27 September 2019.  The analyzed quantity is 10 min-median CBH.

[Figure]

**Figure 6.** Time series of cloud base height for  an exemplary  day (02 September 2019) measured by 42 ASI-pairs (grey filled), by two exemplary ASI-pairs DON-MAR and CLO-FLE with respective camera distances 0.8 and 4.2 km, by the ASI network with refinements and by a ceilometer in the urban area of Oldenburg.

535

First, characteristics of CBH-measurements from the ASI network and from individual ASI-pairs are compared to the CBH-measurement of the reference ceilometer based on insightful days. Then, the  measurements of CBH by ASI network and ASI-pairs  are compared to the one of the ceilometer by scatter-density plots. Subsequently,
540  the accuracy of an ASI-pair and of the ASI network are analyzed for the application of nowcasting of solar irradiance. Finally,  deviation metrics of CBH received from the network and from all individual ASI-pairs per interval of CBH are discussed.

**4.1 Comparison of CBH measurements for  an exemplary  day**

 We first analyze the properties of the different procedures to measure CBH based on exemplary situations.
545 Fig. 6 visualizes time series of CBH for a variable day (02 September 2019) measured by ceilometer, by all available ASI-pairs and by the ASI network. The time series of two exemplary ASI-pairs DON-MAR and CLO-FLE with respective camera distances 0.8 and 4.2 km are plotted. The range of CBH-readings covered by all available ASI-pairs is shaded grey in the figure.

[Figure]

**Figure 7.** Sky images taken by ASI UOL representing  a multi-cloud-layer situation on  02 September 2019 7:20 (left) and an almost clear-sky situation on 02 September 2019 17:00 (right) respectively.

550   In the morning (06:00), both ceilometer and the ASI network recognize adequately a high cloud layer. The ASI-pairs with valid measurements deliver similar estimations of CBH. Around 07:00, the ceilometer still recognizes the high layer whereas many ASI-pairs as well as the ASI network recognize the approaching cumulus clouds. These already cover a significant fraction of the sky in the urban area (compare Fig. 7, left). The CBH estimation approach tends to react stronger to clouds in this area of the sky in which contrasts are typically pronounced. Around 10:20 a multilayer situation is present. In the

555   whole sky dome cumulus clouds are visible but a large fraction of the cloud cover is made up by the cirrus layer. Around this time the measurements of ceilometer and ASI network coincide well. All ASI-pairs recognize a rather low cloud layer while there are periods in which the ceilometer recognizes the cirrus layer. All of the ASI-based CBH estimations react stronger to the low layer and miss the high layer clouds. These two situations impress well why the ASI-based estimations of CBH are less accurate for higher clouds and tend to be negatively biased. On the other hand, for low clouds a high accuracy of the combined

560   CBH estimation is demonstrated.

Meanwhile, it is visible that, for low clouds, many ASI-pairs such as ASI-pair CLO-FLE, tend to overestimate CBH. In these conditions, the ASI network manages well to follow appropriate estimations.

Around 17:00, a nearly clear sky is visible (compare Fig. 7, right). Consequently, the ceilometer does not provide any valid CBH. The ASI-pairs provide a CBH that scatters over a wide range, while the ASI network provides

565    an intermediate CBH. A similar reading of CBH is also recognized by a fraction of the ASI-pairs. From around 17:05, the ASI network detects a CBH of 3 km. With 3.1 km, the following CBH measurements of the ceilometer around 17: 25 confirm the suggested CBH of the approaching cloud layer  (see Fig. B1 for a detail view of the CBH measurements during this almost clear sky period). This situation reflects the expected behavior of the ASI network under mostly clear conditions. However, for a completely clear sky, the ASI network partly produces invalid readings (NaN) and partly it detects a large CBH

570   of around 10 km. In this case, a consecutive image processing step detects the absence of clouds. This step is not part of the present study.

~~CBH of these varies in the range of 7...11 km according to the ceilometer. The range of CBH from ASI-pairs reflects this spread. Still, it is not obvious which of the ASI-pair based observations would be the most appropriate. From the ASI network a rather steady CBH estimation results which most of the time reflects the dominant CBH layer as recognized by the ceilometer. The combined estimation misses physically meaningful variations of CBH typically towards higher values recognized by the ceilometer. Also for this day time series of CBH and corresponding ASI images were compared. Again large underestimations~~

The time series of CBH from DON-MAR and CLO-FLE demonstrate the properties of ASI-pairs with respectively small and large camera distance. DON-MAR is typically close to the reference CBH if it actually takes on a value below 4 km (e.g. 02 September 2019 9:00...13:00) while this ASI-pair tends to take on large deviations and a negative BIAS for larger CBH (e.g. 02 September 2019 6:00...9:00). ASI-pair CLO-FLE typically misses the CBH of low clouds and provides a significantly overestimated CBH (e.g. 02 September 2019 9:00...13:00). For high clouds, however, CBH measured by CLO-FLE often coincides well with the reference.  To give further insight, in Appendix B2, timeseries of CBH from the different sources are compared for another exemplary day.

**4.2 Comparison of CBH measurements by relative frequencies**

 Deviations found for the exemplary ASI-pairs DON-MAR and UOL-HOL with camera distances of 0.8 km and 5.7 km as well as for the ASI network,  without and with the refinements described in Sect. 3.3, are now analyzed with the help of scatter-density plots provided in Fig. 8. The plots visualize the relative frequency of CBH measured by the respective ASI-based systems given a CBH measured by ceilometer. Thus, relative frequencies in each of the columns add to one. The plots also include the median (red dotted), limits to the interquartile range (IQR, red dashed) and 5−, 95−percentiles (red solid line) based on floating 1000 m-bins of CBH from the ceilometer. Each of the subplots further indicates performance metrics of the individual systems: Root mean squared deviation (RMSD), BIAS and coefficient of correlation ($\rho$).

**4.2.1 ASI-pairs**

The readings of ASI-pair DON-MAR, (Fig. 8 upper row, left)  are well aligned with the main diagonal up to a reference CBH of around 4 km. As reference CBH increases further, the ASI-pair increasingly underestimates CBH, indicated e.g. by the median. On the contrary, ASI-pair UOL-HOL (Fig. 8 upper row, right), overestimates CBH massively if reference CBH decreases below 3 km. Whereas based on the median-value, its readings are well aligned with the reference at larger CBH.

[Figure]

**Figure 8.** Relative frequency of ASI-based CBH estimation for given CBH from ceilometer~. Evaluation for two of the ASI-pairs DON-MAR (upper row, left) and UOL-HOL (upper row, right) with respective camera distances of 0.8 and 5.7 km, and from the ASI network without (bottom row, left) and with refinements (bottom row, right). Relative frequency in each column adds up to 1. Additionally, median (50%-quartile, red dotted), limits to the interquartile range (IQR, red dashed) and 5−,95−percentiles (red solid line) based on floating 1000 m-bins of CBH from ceilometer are plotted.

605     Both ASI-pairs exhibit a strong scattering of the measurements, clearly visible from the wide spread of the quartiles as well as of the 5−, 95−percentiles. In agreement with the prior finding, DON-MAR is rather precise at low CBH ($\leq 3$ km), whereas UOL-HOL is notably more precise at greater CBH. CBH from the ASI-pairs often deviates towards very low CBH. This

feature is in part also seen for the ASI network (Fig. 8 bottom row)low CBH, when the ceilometer measures CBH in the range 3...12 km. In this range, the 5-percentile of ASI-based CBH increases only slightly with reference CBH and comparably large relative frequencies are found close to the 5-percentile. As discussed beforein Sect. 4.1, this can in part result from low cloud layers which are actually present in the ASI-pairs' field of view but not at the ceilometer's location. Towards high readings of the reference (≥ 8 km) DON-MAR underestimates CBH for most readings.

CBH measured by ASI-pair

Qualitatively, the effects seen meet the expectation from the literature (Nouri et al., 2019a; Kuhn et al., 2019; Nguyen and Kleis\, ASI-pairs with large camera distance are expected to be more accurate when measuring the CBH of high clouds. On the other hand, ASI-pairs with large camera distance are expected to be less accurate for small CBH values and are expected to exhibit a larger *minimum CBH*, below which no physically meaningful readings are received. From the geometric considerations in Sect. 3.1, a minimum CBH of about $0.18 \times d$ was expected. Where $d$ is the camera distance. For UOL-HOL, which has a camera distance of 5.7 km, is visualized in Fig. 8 upper row, right. CBH measured by UOL-HOL scatters a significantly larger minimum CBH of about 2 km is evident. If reference CBH is smaller than 2 km, the ASI-pair yields measurements of CBH which scatter randomly around a modus value of 3.8 km for reference CBH < 1.8 km. If reference CBH ranges between 1.8...3 km, this behavior is still observed for a significant part of the readings. For UOL-HOL nearly no reading of less than 1.5 km is recognized. In general, strong scattering is seen for this ASI-pair. However, towards large values of reference CBH the measurement appears to scatter to a smaller extent and especially for very large CBH (> 8 km) a satisfying agreement of CBH from ASI-pair and ceilometer is seen. median value of 4 km. This behavior can be explained as the matching procedure fails if pattern are matched which are located at a larger zenith angle than a maximum value. Consequently, random features observed under a zenith angle smaller than the maximum value are often matched erroneously which yields a too large estimation of CBH. Similarly for DON-MAR a minimum CBH of around 0.3 km is suggested.

The measurement of CBH by the ASI network without refinements is shown in Overall, the ASI-pairs are characterized by a minimum CBH in the range of $0.32 \times d$. As described above, this suggests that the matching procedure of the ASI-pairs almost always fails if matched windows cover zenith angles larger than 67°. Further, also for reference CBH close to this minimum CBH, the ASI-pairs yield increased deviations, e.g. below 0.5 km and 3 km for DON-MAR and UOL-HOL.

**4.2.2 ASI network**

Based on Fig. 8 bottom row, left. The modus of the relative frequency distributions is , the ASI-network without refinements succeeds to combine the preferred properties of ASI-pairs with distinct camera distances. The median values of the ASI network are well aligned with the main diagonal for most reference CBH a reference CBH in the range 0.5...10 km. As indicated by the quartiles, the ASI network's precision is similar to that of an ASI-pair with small camera distance, such as DON-MAR, for reference CBH ≤ 4 km. For larger CBH, the network's precision is closer to the one of an ASI-pair with large camera distance, such as UOL-HOL. Additionally, outliers are less frequent and occur with smaller deviations compared to the ASI-pairs discussed before. The ASI networkreturns no reading of CBH of more than 10.9 km. Thus, CBHis underestimated if a corresponding reference CBH is present

In the range of reference CBH > 10 km, the ASI network constantly returns CBH of around 10 km. In the studied climate, reference CBH in this range are comparably rare (see Fig. 4). Therefore, corresponding grid cells of the conditional probability distributions, used by the estimation procedure, were approximated coarsely based on a small number of observations. The ASI network's combination method using cumulative likelihood is intended to avoid deviations resulting from these inaccuracies and thus to yield a more conservative estimation. However, this approach also suppresses the estimation of extreme CBH readings, which causes a BIAS under these conditions. For the analyzed site, deviations found in this range of CBH are of minor importance.

For very low values of reference CBH (especially CBH < 0.3 km) the ASI network without refinements overestimates CBH drastically. None of the ASI-pairs used has a sufficiently small minimum CBH for this range. We expect that the ASI network's accuracy would be enhanced significantly, especially in this range, if ASI-pairs with smaller camera distance than 0.8 km were added.

To improve shortcomings connected to conditions with very low clouds (CBH < 1 km), the refinements introduced in Sect. 3.3 are applied. As indicated by Fig. 8 bottom row, right, these refinements significantly improve the ASI network's performance for reference CBH < 2 km. In this range, the ASI network behaves for the greatest part like ASI-pairs DON-MAR and MAR-DON. The refinements do not affect the statistics notably for reference CBH > 2 km. Overall, this evaluation indicates that the ASI network performs significantly better than an individual ASI-pair, especially if the whole range of studied reference CBH 0...12 km should be covered. This is also indicated by the performance metrics shown in Fig. 8.

**4.3 CBH accuracy under nowcasting conditions**

The procedure to estimate CBH, developed here, will be used as part of a nowcasting system. In this application, it is of special interest to be aware at any time which accuracy can be expected from a specific reading provided by the ASI-network. For this purpose, Fig. 9 shows the relative frequency of CBH measured by the ceilometer given a specific ASI-based estimation of CBH. In each row, the frequencies add up to one. It should be noted, that the performance indicated by this evaluation is more dependent on the local cloud conditions than the one in Sect. 4.2. We analyze

[Figure]

**Figure 9.** Relative frequency of CBH from ceilometer for given ASI-based CBH estimation. Evaluation for ASI-pair DON-MAR (left) and for the ASI network with refinements (right). Subplots (left,right) are created analogously to Fig. 8 (top, left and bottom, right). However, relative frequencies add up to one in each *row* not column.

the systems which are best in class: ASI-pair DON-MAR (Fig. 9, left) and the ASI-network with refinements (Fig. 9, right). As in the previous section, the plots also include the median (red dotted), limits to the interquartile range (IQR, red dashed) and $5-$, $95-$percentiles (red solid line) based on floating 1000 m-bins of ASI-based CBH.

Under most conditions included in Fig. 9, median and interquartile range indicate a good alignment of the CBH estimation
680 from the ASI-network and of CBH from the ceilometer. For ASI-pair DON-MAR, a notable negative BIAS is indicated if the ASI-pair returns a CBH of 9 km or more. Also, if a CBH of more than 4 km is detected, the interquartile range indicates a notably increased precision of the ASI network . The range between the $5-$, $95-$percentiles is wide for both systems. For a wide range of CBH-readings, $5\%$ of the estimations of CBH may deviate by more than 4 km and 3 km from the ceilometer measurement in the case of the ASI-pair and the ASI network, respectively. Still, this range is notably narrower
685 for the  ASI network

Based on Fig. 9, both systems are considered suited for an application in nowcasting at the studied site, while a considerable uncertainty is present. The ASI-network provides a notably improved accuracy in particular in cases when clouds at a CBH $> 4$ km are detected.

690 **4.4 Comparison of CBH accuracy for a three-month data set**

**Table 1.** Frequency of measurements from the validation data set (period 30 June 2019 to 27 September 2019) per range of cloud base height (CBH) used in the evaluations described in Sect. 4.4 (retained) and frequency of those filtered from the evaluation due to increased variability of CBH (rejected).

| CBH range [km] | Observations retained | Observations rejected |
|---|---|---|
| $0 < \text{CBH} \leq 1$ | 11844 | 13255 |
| $1 < \text{CBH} \leq 2$ | 14130 | 9120 |
| $2 < \text{CBH} \leq 4$ | 9962 | 5923 |
| $4 < \text{CBH} \leq 8$ | 5559 | 3570 |
| $8 < \text{CBH} \leq 12$ | 4935 | 1355 |

[Figure]

**Figure 10.** RMSD (left) and BIAS (right) for five ranges of CBH received for all individual ASI-pairs (dots), for the ASI network without (circles), with refinements (diamonds) and for a basic average of CBH measured by all ASI-pairs (horizontal line).

The statistical evaluations are now restricted to times in which the variability of CBH is small. More precisely, the standard deviation of CBH within a window 15 min before and after the analyzed time is required to be less than 30% of mean CBH

within the same window. As discussed above, the ASI-pairs and the ASI network are expected to measure a spatial median CBH whereas the ceilometer measures CBH at the point of its installation. This restriction aims to assure a good comparability of both measurements. Further, this way our results are more comparable to a prior study by Kuhn et al. (2019).

Accuracies of CBH measurement by ASI-pairs and ASI network are analyzed separately for five ranges of reference CBH defined by the bounds $\{0, 1, 2, 4, 8, 12\}$ km. The number of CBH measurements included in this evaluation is given in Table 1 for each of these ranges. The interval bounds are spaced irregularly to correspond better to the distribution of CBH at the site (see also Fig. 4). Table 1 also shows the number of observations excluded from the validation as significant temporal variability of CBH was detected for these observations. While a significant fraction of the readings is sorted out, the representation of the CBH ranges remains widely comparable to the original data set (see Fig. 2, left). Only the range of lowest CBH $< 1000$ m is represented by a notably smaller share of the validation data set.

**4.4.1 Accuracy of the ASI network and ASI-pairs**

Figure 10 compares RMSD (left) and BIAS (right) for CBH estimated by the ASI network, with (diamonds) and without refinements (circles) described in Sect. 3.3, to the one estimated by all ASI-pairs (dots).  The ASI network with refinements provides measurements of CBH that are the most accurate or at least among the most accurate ones for all conditions. In terms of RMSD the estimation from the ASI network is the most accurate for the range of CBH $\in [1,8[$ km (see Fig. 10 left). For CBH $< 1$ km it is slightly outperformed by two ASI-pairs (DON-MAR, MAR-DON) as well as for CBH $> 8$ km by two other ASI-pairs (UOL-CLO, CLO-UOL). ASI network-based measurement of CBH provides among the smallest BIAS for CBH $< 8$ km (see Fig. 10 right). The magnitude of BIAS ranges constantly below 100 m. Only for CBH $> 8$ km the ASI network independently from applied corrections yields a BIAS of roughly $-1050$ m that corresponds to the average BIAS of all used ASI-pairs for these conditions. This deviation  is probably related to situations in which the ASI-based  estimation of CBH recognizes a low cloud layer whereas the ceilometer also recognizes a high layer when gaps in the low layer appear. Therefore, this deviation is rather related to the different nature of the measurements (spatial-median compared to point-wise).

 The distance between the cameras used by an ASI-pair and the reference ceilometer were considered as an influence on the accuracy of an ASI-pair. However, for the ASI-pairs  studied, this distance to the validation site is not confirmed as a significant influence on received accuracy. This was expected in part from the  assumption that the ASI-pairs  measure the median CBH of the most dominant cloud layer in terms of features, driven by area and optical thickness.

 As shown in Fig. 10, without the refinements, in the range CBH < 1 km 12 ASI-pairs with camera distance up to 1.6 km perform better than the ASI network in terms of RMSD and BIAS. As discussed in Sect. 4.2, in this range of reference CBH the ASI network could be improved by ASI-pairs with even smaller camera distance. The  applied refinements improve the accuracy notably. Figure 10 includes the error metrics received when simply averaging CBH measurements of all ASI-pairs.  The ASI network in both variants, with and without refinements, provides a significantly more accurate estimation of CBH in terms of RMSD and BIAS in most ranges of CBH compared to the simple approach.

The individual ASI-pairs and also the ASI-network exibit an RMSD of more than 180 m for all ranges of CBH. Based on this, we do not expect that the bin size of 100 m chosen for the distributions of conditional probability in Sect. 3.4 is a limiting factor to the accuracy of the  ASI-based estimation of CBH in this study. Meanwhile, the underlying ASI-pairs ~~measure CBH of the most dominant cloud layer in the sense of optical thickness and area in the analyzed field of view of the sky. When for example, a CBH of 10 km is present the corresponding spatial area included has side lengths of 15.7 km. For multi-cloud-layer conditions it is likely that within this window lower clouds are present which are recognized by an ASI-pair instead.More aggressive filtering of such multilayer situations included in the evaluation could reduce this influence but would further limit the database. The distance between the cameras used by an ASI-pair and the reference ceilometer are not found to have a significant influence on received accuracy in the evaluated data set. This was expected in part from the previously discussed effect that themeasure CBH of the most dominant cloud layer~~ and also for the ASI-network. This deviation is much larger than the spatial resolution of these maps of solar irradiance. For certain applications, e.g. to control solar power plants (Nouri et al., 2020a), it may still be advantageous to provide maps of solar irradiance at a resolution finer than the uncertainty of cloud shadow edge positions, as the statistical properties of spatial variability may still be captured in these maps.

**4.4.2 Influence of the camera distance on performance metrics**

Lastly we discuss how camera distance influences the performance metrics of the ASI-pairs

[Figure]

**Figure 11.** RMSD (top) and BIAS (bottom) received by 42 ASI-pairs utilizing camera distances in the range of 0.8...5.7 km and by the ASI network with refinements (no camera distance applicable) for the period 30 June 2019 to 27 September 2019.

different nature of both measurements, the accuracy of CBH in different ranges of CBH and compare these results those of Kuhn et al. (2019) who studied the accuracy of ASI-pairs with camera distances in the range of 0.5...2.56 km. Figure 11 provides RMSD and BIAS received from the ASI network is expected to improve further if an average CBH over a range of hours is of interest.

765 **4.5** Discussion of deviations in CBH measurement

For low CBH (CBH < 2 km) accuracy of the measurement by and ASI-pairs decreases with and distinguishes the latter by camera distance. This meets the expectation from Kuhn et al. (2019). Kuhn's study was limited to a maximum camera distance of 2.56 km. Additionally, Metrics of the ASI network, with refinements, are given by horizontal lines. Kuhn et al. (2019) analyzed the accuracy of CBH measurement was only analyzed for three ranges of CBH defined by the limits {0, 3, 8, 12} km.

770 We noticed that a finer classification of CBH as used Overall, in the present study yields more insights for small CBH. Figure 8 upper row provides the relative frequency of CBH readings from two exemplary ASI-pairs given a reference CBH. The camera distances of the ASI-pairs are 0.8 km (left) and 5.3 km (right) respectively. For reference CBH below a minimum value of around 2 km the ASI-pair with camera distance 5.3 km in most cases provides unreasonable readings scattering around 3.8 km. For ASI-pairs with smaller camera distance a similar behavior is observed while the respective minimum CBH reduces with

775 reduced the magnitudes of RMSD and BIAS range well below the values found by Kuhn et al. (2019).

For the CBH ranges 0...1 km and 1...2 km, Fig. 11 shows that BIAS is very small for ASI-pairs with small camera distance. Accordingly, both RMSD and BIAS steadily increase However, beginning at a camera distance of around 1.1 km and 2.5 km respectively, BIAS increases linearly with camera distance as shown in Fig. 11 for CBH ∈ ]0,1[ km. Even for the . Consequently the same trend is visible for RMSD in these ranges of CBH. From the analysis in Sect. 4.2, this effect is clearly connected to the

780 minimum CBH specific to an ASI-pair with camera distance 0.8 km a significant minimum CBH of 0.5 km is found (compare Fig. 8, left) . In line with the discussion above for CBH ∈ [1,2[ km both metrics only increase from a camera distance of 2.5 km on 's camera distance. While the in study of Kuhn et al. (2019) the lowest CBH range covered 0...3 km, which reduces the influence of minimum CBH, a qualitatively similar relationship of camera distance and accuracy was found. Minimum CBH of

785 For intermediate and large CBH (4...12 km) the ASI-pair for this camera distance is identified to be 1.3 km correlation of camera distance and accuracy is less clear – a slight trend seen in RMSD and BIAS is overlaid by strong scattering. The variation of error metrics found between these systems may indicate further influences of the setup on accuracy apart from camera distance. On the other hand, the limited set of observations of high clouds may not be sufficiently representative to identify the influence of camera distance in the presence of other disturbances present in this benchmark, such as low clouds

790 which may be present in spite of the applied filter.

For intermediate and large CBH ( Overall, in the range of CBH > 4 km), increased camera distance slightly improves the accuracy of CBH estimation. On average a reduction in RMSD of 500 m is suggested over the interval of studied camera distances. No significant influence is noticed for BIAS. From Kuhn et al. (2019) the influence of camera distance on accuracy was expected to be more significant in this range of CBH. The influence of CBH on accuracy of the measurement coincides

795

 Further, the orientation of the  ASI-pair's axis to the present direction of cloud movement was considered as an influence on accuracy in Kuhn et al. (2019).  ASI-pairs may measure CBH more

800 accurately if  the ASI-pair's axis is aligned with the direction of cloud motion.  The direction of cloud motion was retrieved from ASI UOL as  described in Sect. 3.2  and the dataset was  filtered to timestamps with cloud motion from west to east. Accuracies of ASI-pairs with similar camera distance but different orientation of the  ASI-pair's axis were compared. In this comparison no correlation of  accuracy and

805 the alignment of the ASI-pair's axis over the direction of cloud motion was recognized.

~~The behavior seen for CBH below a minimum value can be understood as follows. For small CBH and large camera distance the overlapping area (i.e. the fraction of the sky captured by both cameras) becomes small and corresponds to clouds located between both ASIs (Nguyen and Kleissl, 2014). These clouds are observed from very different perspectives by both ASIs. The difference in perspective may be expressed by the angular distance between a cloud's depiction in both ASIs' views. In~~

810 ~~hemispherical ASI images the similarity of a specific cloud observed by both ASIs reduces with this angular distance. Likewise, the representation of two clouds, that are randomly selected from the paired ASIs' sky images respectively, will appear more similar if they are observed at a small angular distance to each other. Therefore, erroneously matched cloud edges will typically be separated by a moderate angular distance. Thus, the likelihood to match cloud objects correctly which are observed at a large angular distance by the paired ASIs is small. If actual CBH relative to camera distance is small the fraction of invalid readings~~

815 ~~(indicating not any match) increases and concurrently a large share of the valid readings goes back to mismatches. Estimated cloud height scales inversely with angular distance of matched cloud patterns for stereoscopic approaches to measure CBH. Consequently, the negative bias of angular distance in the matching translates into a positive bias of estimated CBH. Except for this distinct effect the error metrics of all studied ASI-pairs are very similar for CBH < 4 km.~~

Based on these findings we recommend to chose camera distance of a single ASI-pair, that is not part of an ASI net-

820 work, based on the smallest CBH (CBH$_{min}$) which is of interest at a site. This consideration differs from previous studies by Nguyen and Kleissl (2014) and Kuhn et al. (2019) which suggest, based on theoretical and experimental findings respectively, to optimize camera distance for the most frequent or most relevant CBH. Our experimental results suggest that camera distance of a single ASI-pair should if possible not be chosen larger than  $1.4 \times$ CBH$_{min}$ and in no case larger than $3 \times$ CBH$_{min}$. For the meteorological conditions studied here, ASI-pairs with even smaller camera distances than 0.8 km

825 would be beneficial to cover the range CBH < 0.5 km.

~~Figure 10 provides error metrics for the ASI network both with and without refinements described in Sect. 3.3. Without the refinements, in the range CBH < 1 km 12 ASI-pairs with camera distance up to 1.6 km perform better than the ASI network in terms of RMSD and BIAS. In this range of CBH the ASI network suffers strongly from overestimation of CBH related to the found minimum CBH of involved ASI-pairs. For sites like Oldenburg at which low cloud conditions are dominant (see Sect.~~

830 ~~3.2) the presented approach without refinements would require a larger share of ASI-pairs with small camera distances of even less than 0.8 km. However, the refinements succeed to improve these shortcomings. Figure 10 also includes the error metrics received when simply averaging CBH measurements of all ASI-pairs. The ASI network in both variants, with and without refinements, provides a significantly more accurate estimation of CBH in terms of RMSD and BIAS in most ranges of CBH compared to the simple approach.~~

Appendix B, pp. 38-39, ll. 943-953:

**B2 Comparison of CBH measurements for another exemplary day**

Figure B2 shows CBH on 06 August 2019 again measured by ceilometer, by all available ASI-pairs and by the ASI network.

945 This day, similar to 02 September 2019, discussed previously, includes multi-layer conditions with high layers overlaid by low layers, resulting in similar observations. In the morning and evening high cloud layers are dominant. The CBH of these varies in the range of 7...11 km according to the ceilometer. The range of CBH from ASI-pairs reflects this spread. Still, it is not obvious which of the ASI-pair based observations would be the most appropriate. From the ASI network a rather steady CBH estimation results which most of the time reflects the dominant CBH layer as recognized by the ceilometer. The combined

950 estimation misses physically meaningful variations of CBH typically towards higher values recognized by the ceilometer. Also for this day time series of CBH and corresponding ASI images were compared. Again large underestimations of CBH by the ASI network (at 05:30, 08:15, 10:00, 12:30, 16:00) were traced back to the ASI-based estimations responding stronger to lower optically denser low cloud layers which pass the vicinity of the urban area (compare Fig. B3).

[Figure]

**Figure B2.** Time series of cloud base height for an exemplary day (06 August 2019) measured by 42 ASI-pairs (grey filled), by two exemplary ASI-pairs DON-MAR and CLO-FLE with respective camera distances 0.8 and 4.2 km, by the ASI network with refinements and by a ceilometer in the urban area of Oldenburg.

[Figure]

**Figure B3.** Sky image taken by ASI UOL representing a multi-cloudlayer situation on 06 August 2019 12:35

**Specific comment 1**

Reviewer:

Line 85: Is the 3-month period enough to monitor all available conditions? What would be a suggestion to other users about the range of conditions that needs to be captured for good training?

Authors' response:

One idea of our method was that it should not be necessary to train the model based on a dataset which represents the conditions during the operation or validation. The method should work best if the ASI-pairs exhibit the same behavior at a given reference CBH during model development and validation. I.e. distributions of conditional probabilities at a given reference CBH should be comparable for both data sets.

Apart from that the data sets used for modeling and validation are both considered to be qualitatively representative of the months which are of greatest interest to solar applications at the studied latitude, as they may provide the greatest energy yield, based on sky conditions and sun elevation.

Changes in manuscript:

p. 17, ll. 412-418:

29 June 2019 are used. CBH measured by the ceilometer serves as reference CBH. It is considered not to be essential that the training period is representative of the period to which the method is applied. However, we expect that the method works best if the included ASI-pairs exhibit a similar distribution of measurement deviations given the same reference CBH in both
415 periods. For solar applications and the latitude of this study, we consider the used dataset and its split reasonable. The summer and shoulder months provide the main share of the annual solar yield at the site and are therefore in the focus of the nowcasting system under development. In that sense, the training dataset is considered to be for the large part representative of conditions relevant to solar applications at similar latitudes.

**Specific comment 2**
Reviewer:

Line 108: "by arbitrary selecting a tuple of ASIs". From the text, you seem to be selecting all possible combinations of ASIs not only some arbitrary pairs. Moreover, I am not sure if tuple is the proper name as, in my mind, a tuple could include more than2 objects. Consider rephrasing.

Authors' response:

The description we used was misleading and should be understood as indicated by the reviewer. We replaced the term by "iteratively". Further we now pointed out in the same paragraph, that all 42 ASI-pairs are considered for the estimation procedure.

Indeed, the term tuple is not precise. Throughout the text, tuples are intended to have only two members. We replaced the term in general by "tuple of 2 ASIs".

Changes in manuscript:

p.1 l.10:

> 10    In this study, the deviations of 42 ASI-pairs are studied in comparison to a ceilometer and characterized by camera distance. The ASI-pairs are formed from seven ASIs and feature camera distances of 0.8...5.7 km. Each of the 21  tuples of two ASIs formed from seven ASIs yields two independent ASI-pairs as the ASI used as main and auxiliary camera respectively is swapped. Deviations found are compiled into conditional probabilities telling how probable it is to receive a certain reading

p. 5 ll. 127-131:

> For this study, these ASIs are arranged into several ASI-pairs as defined by  iteratively selecting a tuple of  21 tuples  available and forming two independent ASI-pairs from each tuple by swapping its main camera. The main camera of an ASI-pair is central to the measurement of CBH through an ASI-pair, described in
> 130    more detail in Sect. 3.1, and defines the center of the area for which CBH is estimated. From 21 tuples of 2 ASIs, 42 ASI-pairs are received. All 42 ASI-pairs are included in the estimation procedure. The paired cameras' distance and the orientation of the  ASI-pair's axis characterize the ASI-pairs. The orientation of  an ASI-pair's axis is defined

p. 6, caption of Fig. 6:

> **Figure 2.** Frequency distribution of  the bearing angles of the ASI-pairs' axes in the set of available ASI-pairs (over north, left) and of available camera distances (right) resulting when arranging the seven ASIs in the urban area into 42 ASI-pairs (from each  2-ASI-tuple two different ASI-pairs result by switching the main camera, counts of ASI-pairs with switched main camera are marked orange, striped)

p. 17 l. 419:

> The seven ASIs available in the urban area are arranged into 42 ASI-pairs. Each tuple of two ASIs, that is selected from the
> 420    set of seven ASIs, yields 2 independent ASI-pairs by swapping the ASI used as main camera (see Sect. 3.1).

**Specific comment 3**

Reviewer:

Line 111: "Camera axis". Does this refer to the line connecting the two ASIs that form a pair? If yes, then it should be called "pair-axis" or similar. "Camera axis" sound to me as the name for the direction that a single camera is looking.

Authors' response:

Indeed, the term is ambiguous and was adapted now to "ASI-pair's axis".

The nomenclature was originally motivated by the one used by Kuhn, P., B. Nouri, S. Wilbert, N. Hanrieder, C. Prahl, L. Ramirez, L. Zarzalejo, T. Schmidt, Z. Yasser, D. Heinemann, P. Tzoumanikas, A. Kazantzidis, J. Kleissl, P. Blanc and R. Pitz-Paal (2019). "Determination of the optimal camera distance for cloud height measurements with two all-sky imagers." Solar Energy 179: 74-88.

Changes in manuscript:

pp. 5-6 ll. 132-135, caption Fig. 2:

130    more detail in Sect. 3.1, and defines the center of the area for which CBH is estimated. From 21 tuples of 2 ASIs, 42 ASI-pairs are received. All 42 ASI-pairs are included in the estimation procedure. The paired cameras' distance and the orientation of the  ASI-pair's axis characterize the ASI-pairs. The orientation of  an ASI-pair's axis is defined as seen from the main ASI and given in degree north. Figure 2 shows the distribution of orientations of  ASI-pair's axes (left) and camera distances (right) in the set of available ASI-pairs. This set covers almost all possible orientations of

[Figure]

**Figure 2.** Frequency distribution of  the bearing angles of the ASI-pairs' axes in the set of available ASI-pairs (over north, left) and of available camera distances (right) resulting when arranging the seven ASIs in the urban area into 42 ASI-pairs (from each  2-ASI-tuple two different ASI-pairs result by switching the main camera, counts of ASI-pairs with switched main camera are marked orange, striped)

135    ASI-pair's axes. Available camera distances 0.8...5.7 km cover most of the range 0.02...5.5 km that is used in literature (Kuhn

p. 33 ll. 798, 800, 803, 805:

     Further, the orientation of the  ASI-pair's axis to the present direction of cloud movement was considered as an influence on accuracy in Kuhn et al. (2019).  ASI-pairs may measure CBH more

800    accurately if  the ASI-pair's axis is aligned with the direction of cloud motion.  The direction of cloud motion was retrieved from ASI UOL as  described in Sect. 3.2  and the dataset was  filtered to timestamps with cloud motion from west to east. Accuracies of ASI-pairs with similar camera distance but different orientation of the  ASI-pair's axis were compared. In this comparison no correlation of  accuracy and

805    the alignment of the ASI-pair's axis over the direction of cloud motion was recognized.

**Specific comment 4**
**Reviewer:**

Line 116:  For completeness, please provide some more information about the instrument: E.g. Is the instrument part of DWD network you mentioned before?  How is the CBH calculated from the data? Are you using the manufacturer's algorithm or a custom one? What is the minimum overlap height? What is the minimum height that CBH can be detected? References?

Authors' response:

We added a short description about the used instrument. It is operated by DLR since 2018. The manufacturer's algorithm is used with the default configuration. The algorithm is outlined in the instrument's manual. The firmware version is v0.747. A prior study stated that full overlap is given at a CBH of 1500 m and above. Based on an overlap correction, the manufacturer allows to set a minimum CBH down to 0 m. We use the default setting of 45 m. We also contacted the manufacturer for further information on the used algorithms in the meantime but did not receive a response, yet. If required, this can be handed in at a later time.

Changes in manuscript:

p. 6, ll. 137-141:

> The used ceilometer  of type Lufft CHM 15 k Nimbus (firmware v0.747) is operated by DLR since 2018. CBH is measured by the manufacturer's *Sky Condition Algorithm* (Lufft, 2018) in the default configuration. Heese et al. (2010) specifies for a ceilometer of the same type, that full overlap of the laser's and the receiver's field of view is reached at a height of 1500 m.
>
> 140 However relying on an overlap correction, the manufacturer specifies a minimum CBH of down to 0 m. In this study the manufacturer's default minimum CBH of 45 m is used.

**Specific comment 5**

Reviewer:

Line 116: Are you using the color or B&W version of Q25?

Authors' response:

We use the daylight version of Mobotix Q25 6MP. This is the RGB/color version. We now also attached a reference to the instrument's specification.

Changes in manuscript:

p. 6, l. 142:

> 140 However relying on an overlap correction, the manufacturer specifies a minimum CBH of down to 0 m. In this study the manufacturer's default minimum CBH of 45 m is used.
>
> The used ASIs are surveillance cameras of type Mobotix Q25 6MP color version (Mobotix, 2017) with a fisheye lens pro-

**Specific comment 6**

Reviewer:

Line 130: Please mention what is the total time required to get a processed image (including data transfer and processing)?

Authors' response:

We now specified the overall time required for image acquisition, transfer and processing, as suggested. Note that a further addition was made here based on Reviewer Comment 1, Minor comment 3.

Changes in manuscript:

p. 7, ll. 156-158:

> 155 avoiding redundant calculations. In this way, computational cost scales mostly linear with the number of ASIs used instead of with the number of ASI combinations so that execution in real time is possible. In total, including computation time, the estimation of CBH by the ASI network can be retrieved within 10 s after image acquisition. CBH is computed by the ASI-pairs and by the ASI network during daytime, i.e. if the sun elevation at the time of image acquisition is greater than 0°.

**Specific comment 7**

Reviewer:

Line 141: *optically* thick clouds.

Authors' response:

As suggested, we now specified this more accurately.

Changes in manuscript:

p. 7, l. 171:

> 170 As introduced in Sect. 2, a ceilometer of type Lufft CHM 15 k Nimbus is used as reference in the development and validation presented in this study. When low and optically thick clouds are present, only the lowest cloud layer is expected to be recognized

**Specific comment 8**

Reviewer:

Line 141-144: How exactly do you distinguish if the first cloud layer is thick, to exclude the other detected cloud layers? Do you always keep only the first layer when multiple layers are detected?

Authors' response:

This is the case. We now specified this more clearly.

Changes in manuscript:

p. 7, l. 172-173:

170    As introduced in Sect. 2, a ceilometer of type Lufft CHM 15 k Nimbus is used as reference in the development and validation presented in this study. When low and optically thick clouds are present, only the lowest cloud layer is expected to be recognized reliably by the ceilometer. Therefore, in the case of overlaid cloud layers, we only evaluate readings provided for the lowest layer. This approach applies to all evaluations presented in this publication.

**Specific comment 9a**

Reviewer:

Line 143-144: The accuracy discussion is not enough for an instrument sued as reference. The differences reported in Martucci et al. 2010 seem to be coming from different algorithm or even definition of CBH used by each instrument. Moreover, the bias they find is not only 160 meters, but also has a range component (Y=0.925X + 160). Finally, Martucci et al used a rather old model of the instrument you are using here. Therefore, you should give more details about the CBH algorithm used with Ceilometer data and discuss the possible differences in definition of CBH as used for ceilometer and for ASIs.

Authors' response:

Indeed, there have been several updates to the firmware after 2010. Some of these indicate changes to the algorithm of CBH measurement. We now summarized the results of two more recent studies which evaluated the CBH measurement by the ceilometer type used here. Based on these authors' findings, we also provided differences in the algorithms used by the manufacturers. Further, we explained that prior validations of the method used in this study to measure CBH by the ASI-pairs, were performed by an instrument of the same type. This may avoid inconsistencies when comparing the results of the present study to those prior ones.

Changes in manuscript:

pp. 7-9, ll. 174-203:

Regarding the accuracy of  ceilometers in general,
175 de Haij et al. (2016) and Görsdorf et al. (2016) noted that there is no generally excepted, quantifiable definition of CBH, yet. Further, due to a lack of reference measurements, benchmarks may typically focus on the consistency of CBH measurements by different types of ceilometers. In a benchmark performed by Martucci et al. (2010), the measurement of a Vaisala CL31 ceilometer $CBH_{CL}$ showed a significant deviation from the reading $CBH_{CHM}$ of the instrument
180 used here. This trend was given by $CBH_{CL} = 160.315\,\mathrm{m} + 0.925 * CBH_{CHM}$. However, the measurement procedure, of the instrument used here, was modified by firmware updates in the meantime. Görsdorf et al. (2016) presented results from a more recent measurement campaign, CeiLinEx2015, which took place in 2015. In this experiment the measurements of six types of ceilometers were compared. For stratus and stratocumulus clouds as well as for fog, deviations between the instruments of up to 70 m were observed. For each of these conditions, the CHM 15 k, used here, provided the smallest measurements of CBH

185 in terms of mean deviation from the median of all tested instruments. More severe deviations of several kilometers between the instrument types were observed during conditions with heavy rain.

In an acceptance test, de Haij et al. (2016) measured CBH by two CHM 15 k, by a Vaisala LD40 ceilometer, by a UV lidar (Leosphere ALS450) and by visibility sensors mounted in various altitudes on a tower of 213 m height. For CBH of up to 200 m, the CHM 15 k typically measured a CBH 30...50 m smaller than the one of the LD40. However, the CHM 15 k was in
190 better agreement with the estimate based on visibility sensors. Görsdorf et al. (2016) and de Haij et al. (2016) suggest, that the negative mean deviation of the CHM 15 k attested by all these studies, for clouds in the range $CBH < 3\,\mathrm{km}$, is mostly caused by the manufacturers' algorithms to detect CBH from backscatter profiles. Whereas, according to the manufacturer (Lufft, 2018), the CHM 15 k detects the rising edge of a backscatter peak that exceeds a threshold, other manufacturers' devices may rather recognize the peak's maximum.

195 For the range of CBH in 3...12 km, an inspection of timeseries depicted by de Haij et al. (2016) indicates very good agreement of the measurements from CHM 15 k and the UV lidar, used there. As a further test of de Haij et al. (2016), performed at a resolution of 60 s, high clouds, detected by the UV lidar in a range of 6...7.5 km, were to be detected by the CHM 15 k within a tolerance of $\pm 3$ classes in hh code (WMO Table 1677). This tolerance corresponds to a CBH-range of $\pm 1050\,\mathrm{m}$ centered around the discretized reference CBH. CHM 15 k was attested a probability of detection of $> 98\%$ and
200 a false alarm rate of 0%. Based on these studies, the accuracy of the reference instrument is expected to be adequate for the range of $CBH < 3\,\mathrm{km}$ and also for the range of $CBH > 3\,\mathrm{km}$, a rather good performance of the instrument is indicated. The experimental results of this study will in particular be compared to prior studies which used a ceilometer of the same type. This is expected to avoid possible inconsistencies related to the used reference.

pp. 9, l. 210:

210 less than 1 km occured. The studies of Kuhn et al. (2018b) and Nouri et al. (2019a) were performed in Almería, Spain. Both studies validated the ASI-based measurement of CBH using a ceilometer of type Lufft CHM 15 k as reference. At this point

**Specific comment 9b**
Reviewer:

Lines 161 - 176: The description of the algorithm is not very clear. Please add a new figure (or add a panel in Fig.) showing the image of the second ASI, highlighting the matched window. Also, a small flowchart could be helpful.

Authors' response:

As suggested by the reviewer, we added another row to Fig. 3 showing the raw and processed image simultaneously recorded by ASI FLE. For a flow chart of the method we would like to refer to

Nouri, B., P. Kuhn, S. Wilbert, N. Hanrieder, C. Prahl, L. Zarzalejo, A. Kazantzidis, P. Blanc and R. Pitz-Paal (2019). "Cloud height and tracking accuracy of three all sky imager systems for individual clouds." Solar Energy **177**: 213-228.

We revised the description of the algorithm and hope that it is clearer now. Further, we aimed to point out clearer that the CBH measurement of the ASI-pairs is only modified very slightly over the one described and validated in the publication given above. We further provided validation results of that study.

Changes in manuscript:

p. 8, Fig. 3:

[Figure]

**Figure 3.** Sky  areas evaluated in the measurement of CBH exemplary for ASI-pair FLE-UOL, with ASI UOL in the top row and FLE in the bottom row. Maximum extent (solid green shape) and area used by the main camera in the default case (red dashed shape) in the distorted ASI image (left), in the undistorted ortho-image (center), in the binary red-channel difference image of two consecutive exposures (right). The binary red-channel difference image (right) shows areas considered as features in the cross-correlation for the comparison to the second camera as yellow shapes. A rejected match between the ASI images is marked orange, a valid match is marked light blue.

pp. 9-10, ll. 204-251

From all ASIs available in the urban area, we form independent ASI-pairs that measure CBH by a stereoscopic triangulation
205  which was introduced by Kuhn et al. (2018b) and further refined by Nouri et al. (2019a). The algorithm used here to estimate CBH by the individual ASI-pairs has been described and validated in the latter publication. Nouri et al. (2019a) evaluated an ASI-pair with a camera distance of 495 m. For four ranges of reference CBH, defined by the bin edges 0, 3, 6, 9, 12 km, RMSDs of 0.6, 1.4, 3.2, 3.1 km were found for 10 min average CBH. The study did not provide information on BIAS. Further, in that validation, higher clouds were more frequent and no observations at a reference CBH of
210 less than 1 km occured. The studies of Kuhn et al. (2018b) and Nouri et al. (2019a) were performed in Almería, Spain. Both studies validated the ASI-based measurement of CBH using a ceilometer of type Lufft CHM 15 k as reference. At this point we recapitulate aspects of the procedure which are important for the remaining publication. For a more detailed description, we refer to Nouri et al. (2019a).

Images from both  ASIs (e.g. UOL and FLE, see Fig. 3, left) are first projected into horizontal planes yielding
215 orthogonal images (Fig. 3, center) by a well established method described e.g. by Luhmann (2000). Then, the difference in the red-channel  compared to the image recorded 30 s before is calculated for  the image of each ASI. Areas in the difference images of the two cameras, in which the red-channel changes most significantly (98-percentile) within the 30 s between consecutive images, are used as features (illustrated in Fig. 3, right) to be matched by block-wise correlation. With the known camera distance, a shift received in cross-correlation is translated into a height of the feature over
220 ground.

In practice, the triangulation relies on cloud edges which are visible from both perspectives and provide sufficient contrast. Therefore, the method responds stronger to optically dense clouds, especially in the proximity of the sun , as found by Kuhn et al. (2018b). Moreover, we do not exactly measure CBH but the height of these distinct cloud edges. We expect to introduce a small bias when using this cloud height as CBH. Nouri et al. (2019a) analyzed sources of deviations
225 when estimating CBH by an ASI-pair. In accordance with that study, we expect this bias to be acceptable compared to other uncertainties and to be in the order of 100 m.

In  accordance with the system used by Nouri et al. (2019a), we use a cascading procedure to estimate CBH robustly also in conditions with low sky coverage.  First, the main ASI's orthogonal image is restricted to a square-shaped area (Fig. 3, red dashed shape) defined by a maximum zenith angle of 67°,
230 measured  in the center of each  side of the square. In a cross-correlation, each of the nine squares confined by dotted or dashed lines (also known as windows, Fig. 3, bottom, right) from the orthoimage of the main ASI is matched with an area of identical shape from the orthoimage of the second ASI (Fig. 3,  top, right). With the known camera distance, the shift is converted into a measurement of CBH.

235 If the estimation of CBH failed for one of the windows, valid readings from neighboring ones are averaged ignoring any window for which the estimation failed. In cases with no valid measurement in any of the windows, the orthogonal images of both ASIs are evaluated up to a maximum zenith angle of 77.8° (measured at the center of each image side, green shapes in Fig. 3  ). These orthoimages from both cameras are matched in the cross-correlation and the ASI-pair returns a uniform CBH. This second step can yield a valid measurement of CBH in cases when only few clouds are
240 present to be matched. This step mainly intends to increase the robustness of the CBH measurement. This step is not expected to increase the capability of an ASI-pair to detect very low clouds in relation to the camera distance, as the window size used in this step is very large.

As a modification of the  method by Nouri et al. (2019a), we only use CBH provided for the central point of the orthoimage of the
245 main ASI, corresponding to a zenith angle of 0°. This procedure is followed for both the ASI-pairs and for the ASI network using these ASI-pairs. We expect that ASI-based measurement of CBH is most accurate for this central point. This point receives CBH primarily from matches involving the central window of the  main ASI's orthoimage, which is less affected by image distortion. The central window of the main  ASI's orthoimage covers zenith angles up to 38.1°, measured at the center of each window side.  Thus, a CBH measurement for a square-shaped area around the main
250  ASI's location is yielded. For example, the area's side lengths measure 1.6, 4.7, 7.8, 15.7 km for a respective CBH of 1, 3, 5, 10 km.

**Specific comment 10**

Reviewer:

Lines 161 – 176: Have you compared the results from the three method (center box, side boxes, full image) to validate your expectation that they yield similar results?

Authors' response:

Unfortunately, we did not validate these sub-algorithms separately. As described in our response to Specific Comment 9 we now pointed out clearer that the method to estimate CBH, used by the ASI-pairs, is only modified very slightly over the publication which introduced this method and implementation:

Nouri, B., P. Kuhn, S. Wilbert, N. Hanrieder, C. Prahl, L. Zarzalejo, A. Kazantzidis, P. Blanc and R. Pitz-Paal (2019). "Cloud height and tracking accuracy of three all sky imager systems for individual clouds." Solar Energy **177**: 213-228.

Therefore, we would like to refer to this study for further validation results. As part of a future study, it would be interesting to investigate the characteristics of these sub-algorithms. In our expectation, CBH will be measured more accurately by matches which are detected at small zenith angles.

Further, CBH measurement received for this central image area, which is used here, were also in the focus of the validation carried out in the publication named above, due to the cloud conditions at that site and due to the positions chosen for ASIs and ceilometer.

Changes in manuscript:

p. 10, ll. 243-248:

As a modification of the  method by Nouri et al. (2019a), we only use CBH provided for the central point of the orthoimage of the
245   main ASI, corresponding to a zenith angle of 0°. This procedure is followed for both the ASI-pairs and for the ASI network using these ASI-pairs. We expect that ASI-based measurement of CBH is most accurate for this central point. This point receives CBH primarily from matches involving the central window of the  main ASI's orthoimage, which is less affected by image distortion. The central window of the main  ASI's orthoimage covers zenith angles up to 38.1°, measured at the center of each window side. Thus, a CBH measurement for a square-shaped area around the main
250   ASI's location is yielded. For example, the area's side lengths measure 1.6, 4.7, 7.8, 15.7 km for a respective CBH of 1, 3, 5, 10 km.

p. 11, l. 266-269:

265   windows and $0.64 \times d$ using the central window.
This central point of the orthoimage, used here, was also in the focus of the validation presented by Nouri et al. (2019a) as the ceilometer was placed at one ASI's location and as observed CBH values were not smaller than 1 km. Overall, we expect that, by applying cross-correlation to binary difference images, our

**Specific comment 11**

Reviewer:

Lines 161 – 176: Please provide the relations connecting a) the ASI-pair distance with b) the minimum altitude that each method can be applied, due to purely geometric considerations.

Authors' response:

We calculated the expected minimum CBH which can be detected relying on the central window as well as when relying on all of the nine windows inside the cropped image of the main ASI. We additionally calculated the minimum CBH which is achieved if matches only succeed if matched windows cover zenith angles not larger than 67°. We further stated that the third iteration of the matching procedure in which the ASI image is evaluated up to a zenith angle of 77.8° is not expected to reduce minimum CBH as this step matches a very large windows.

Changes in manuscript:

p. 10, ll. 236-242:

window for which the estimation failed. In cases with no valid measurement in any of the windows, the orthogonal images of both ASIs are evaluated up to a maximum zenith angle of 77.8° (measured at the center of each image side, green shapes in Fig. 3, central red dotted box)from ). These orthoimages from both cameras are matched in the cross-correlation and the ASI-pair returns a uniform CBH. This second step can yield a valid measurement of CBH in cases when only few clouds are
240    present to be matched. This step mainly intends to increase the robustness of the CBH measurement. This step is not expected to increase the capability of an ASI-pair to detect very low clouds in relation to the camera distance, as the window size used in this step is very large.

p. 10-11, ll. 252-265:

If the estimation of CBH fails for Only based on geometry and the evaluated image areas, this central window , we use the CBH that is measured by matching the peripheral windows (Fig. 3, peripheral red dotted boxes) of the same orthoimage with the orthoimage of the second camera. These peripheral windowsof an orthoimage have the same shape as the central window
255    (see could provide readings down to a minimum CBH of $0.25 \times d$. Where $d$ is the camera distance. However, under such extreme conditions the matching procedure may fail very frequently. The central peripheral windows, shown in Fig. 3, center, peripheral red dotted boxes). If a valid estimation of CBH is received for multiple peripheral windows, we use the average CBHfrom these windows.

For cases with still no valid measurement, images of both cameras are evaluated up to a maximum zenith angle of approximately
260    cover zenith angles 38.1..67°. The matched area from the auxiliary ASI's orthogonal image has identical shape and can cover a zenith angle up to 77.8°(measured at . Based on this, we estimate the minimum CBH, which an ASI-pair can measure, to be $0.18 \times d$. However, from our experience, a large fraction of clouds observed at zenith angles larger than 67° are not matched successfully between the ASIs and typically rejected. If the matching procedure could only be successful, if also the window of

the second ASI included zenith angles not larger than 67°, then CBH could be measured down to $0.32 \times d$ using the peripheral
265    windows and $0.64 \times d$ using the central window.

**Specific comment 12**

Reviewer:

Line 186: Specify that this analysis is based on the ceilometer. Is the CBH analysis based only on the lowest layer detected by the ceilometer?

Authors' response:

We added the statement as suggested. As in the complete study, we carried out this analysis only based on the lowest recognized cloud layer.

Changes in manuscript:

p. 12, ll. 284-285:

> The distribution of CBH at the site of Oldenburg for the full measuring period is given in Fig. 4 right. As in general in this
> 285 study, the analysis is based only on the lowest cloud layer detected by the ceilometer. The majority of all ceilometer readings

**Specific comment 13**

Reviewer:

Line 200: How are TanDEM-X data used in this study? This seems the wrong place of the manuscript to introduce a new dataset.

Authors' response:

TanDEM-X data are needed by the nowcasting system to create irradiance maps. For this study the data set is only relevant for this estimation of the maximum elevation of the topography. We rephrased as shown below.

Changes in manuscript:

p. 12, ll. 299-300:

> tance of roughly 70 km. Eye2Sky and especially Oldenburg are situated in a plane with a maximum elevation over sea level of
> less than 160 m including vegetation and human infrastructure(TanDEM-X topographic data used in this study is described by Wes
> 300 , as we calculated from the TanDEM-X elevation model (Wessel et al., 2018). The flat topography is expected to support a tem-

**Specific comment 14**

Reviewer:

Line 187-206: How is this analysis of CBH stability relevant to this study? Does your algorithm work only in these conditions? Maybe the stability excludes some possible errors in transition periods? Please mention the context and usefulness of this part of the manuscript.

Authors' response:

The meteorological conditions described in this paragraph motivated the development of a method which aims to estimate CBH of the most dominant cloud layer more accurately. We added a conclusion to this paragraph which puts the analysis into the context of this study. Further as suggested by Reviewer Comment 1, General comment 2, we outlined at this point the scope of the method to estimate CBH and how it may be enhanced in the future.

Changes in manuscript:

p. 12, ll. 305-310:

305     characteristics are expected to cause greater temporal and spatial variability of CBH. To conclude, a procedure, which estimates CBH of the cloud layer most dominant in the urban area of Oldenburg accurately, is considered beneficial to assess and model clouds in the same area (depicted in Fig. 1, right). Still, if clouds over the whole region covered by Eye2Sky (depicted in Fig. 1, left) are assessed, this method alone may not be sufficient. In the future, local cloud conditions may be classified by image processing techniques (e.g. Fabel et al., 2021) and CBH may be assigned to local clouds from clouds of the same type, which

310     were recently observed in the urban area.

**Specific comment 15**

Reviewer:

Line 228: "..., where N is the number of vertical bins used for the analysis" or similar.

Authors' response:

We adapted this statement as suggested.

Changes in manuscript:

Appendix A, p. 35, l.886:

885     a discrete grid cell defined by the interval $[j\Delta h, (j+1)\Delta h[$ for $h_{Ref}$ and the interval $[k\Delta h, (k+1)\Delta h[$ for $h_{ASI}$, where $j, k \in \{0, 1, 2, ..., N-1\}$, where N is the number of bins used for CBH in the analysis. A bin size $\Delta h = 100\,\text{m}$ is chosen in a trade-off between sources of error. Finer bins will allow to represent the distributions at higher resolution and will thus allow for

**Specific comment 16**

Reviewer:

Line 303: Why use theta for true CBH and not a symbol based on h?

Authors' response:

We adapted the nomenclature as suggested, as it may be clearer (replacing $\theta, \hat{\theta}, \theta_{likeliest}, \theta_{refined}$ by $h_{true}, \hat{h}_{true}, h_{likeliest}, h_{refined}$). Theta was used as this symbol may be used frequently with maximum likelihood estimation for the true/ estimated parameter.

Changes in manuscript:

p. 14-15, ll. 337-350:

 $P(\mathrm{CBH}_i \mid h_{true})$ is evaluated that the found $\mathrm{CBH}_i$ would be received for a given true CBH  $h_{true}$ (red marked box prior to step 1 in Fig. 5). Note that  $P(\mathrm{CBH}_i \mid h_{true})$ will be modeled in Sect. 3.4 measuring CBH $h_{Ref}$ by a ceilometer which  provides $h_{Ref} \approx h_{true}$. Thus, the likelihood  $\mathcal{L}_i(h_{true})$ is obtained (Fig. 5, output of step 1):

$$\mathcal{L}_i(\sout{\theta}h_{true}) = P(\mathrm{CBH}_i \mid \sout{\theta}h_{true}). \tag{1}$$

Step 2:  We define cumulative likelihood $\mathcal{C}_i(\hat{h}_{true})$ as the likelihood of receiving the present reading $\mathrm{CBH}_i$ given that $h_{true}$ is smaller or equal to an estimation of true CBH $\hat{h}_{true}$. Accordingly in the implementation, likelihood is summed cumulatively over all bins of reference CBH  $h_{true}$ (Fig. 5, step 2):

$$\mathcal{C}_i(\hat{h}_{true}) = \sum_{\sout{\theta \leq \hat{\theta}} h_{true} \leq \hat{h}_{true}} \mathcal{L}_i(\sout{\theta}h_{true}). \tag{2}$$

Likewise, a complementary cumulative likelihood is defined

$$\bar{\mathcal{C}}_i(\hat{\theta}) = \sum_{\theta > \hat{\theta}} \mathcal{L}_i(\theta).$$

as the likelihood of receiving the present reading $\mathrm{CBH}_i$ given that $h_{true}$ is greater than an estimation of true CBH $\hat{h}_{true}$:

$$\bar{\mathcal{C}}_i(\hat{h}_{true}) = \sum_{h_{true} > \hat{h}_{true}} \mathcal{L}_i(h_{true}). \tag{3}$$

pp.15-16, ll. 364-401:

 Both functions $\mathcal{C}_i(\hat{h}_{true})$ and $\bar{\mathcal{C}}_i(\hat{h}_{true})$ are shown for three exemplary intervals of camera distance in Fig. 5 as output of step 2.

Step 3:  We aim to determine the likelihood of receiving the combination of readings $CBH_i$ from all the intervals $i$ of camera distance  given that $h_{true} \leq \hat{h}_{true}$. This can be expressed as product of $\mathcal{C}_i(\hat{h}_{true})$ from all intervals $i$. As this product would often become zero in our numerical treatment, we instead calculate its natural logarithm, which we refer to as overall logarithmized cumulative likelihood $\log \mathcal{C}_n(\hat{h}_{true})$. This operation also allows to replace the product by a sum (Fig. 5, step 3):

$$\log \mathcal{C}_n(\hat{h}_{true}) = \sum_i \log \mathcal{C}_i(\hat{h}_{true}). \tag{4}$$

Analogously, an overall complementary logarithmized cumulative likelihood is computed given all readings $CBH_i$ per interval $i$ of camera distance

$$\log \bar{\mathcal{C}}_n(\hat{h}_{true}) = \sum_i \log \bar{\mathcal{C}}_i(\hat{h}_{true}). \tag{5}$$

Both functions are visualized exemplarily as output of step 3 in Fig. 5.

Step 4:  $\log \mathcal{C}_n(\hat{h}_{true})$ and $\log \bar{\mathcal{C}}_n(\hat{h}_{true})$ are only known at discrete points. Linear interpolation yields continuous representations of these.  Then finally, we aim to select the true CBH $h_{likeliest}$, which makes it likeliest to receive the given combination of $CBH_i$. In our formulation of the problem, this means we intend to find a $\hat{h}_{likeliest}$ which simultaneously maximizes $\log \mathcal{C}_n(\hat{h}_{true})$ and $\log \bar{\mathcal{C}}_n(\hat{h}_{true})$. Consequently, we accept $h_{likeliest}$, for which $\log \mathcal{C}_n(\hat{h}_{true})$ and $\log \bar{\mathcal{C}}_n(\hat{h}_{true})$ are equal (Fig. 5, step 4):

$$\underline{\theta} h_{likeliest} = \operatorname{argmin}_{\underline{\hat\theta} \hat{h}_{true}} \left| \log \bar{\mathcal{C}}_n(\hat{h}_{true}) - \log \mathcal{C}_n(\hat{h}_{true}) \right|. \tag{6}$$

Besides this estimation of CBH, a version of this procedure will be discussed that includes further refinements (in the following referred to as *refined* estimation).  As a first observation from the generation of conditional probabilities, ASI-pairs  with camera distance greater than 4.5 km  cause large deviations for CBH < 4 km and  exhibit only a moderate advantage at greater CBH.  These ASI-pairs are excluded from the refined estimation of $h_{likeliest}$. On the other hand, ASI-pairs with  small camera distance are already accurate if only small CBH occur, as we will discuss in Sect. 4. We inspected conditional probabilities of the ASI-pairs  (exemplarily viewed as input to step 1 in Fig. 5)  and identified the ASI-pairs which are most appropriate for an interval of CBH. Based on this, the refined estimation is  received from the arithmetic average of CBH measured by ASI-pairs with corresponding small camera distance, if the first iteration of $h_{likeliest}$ yielded a sufficiently small CBH. In summary, the refinement procedure to receive the final estimation of CBH  $h_{refined}$ reads

$$\underline{\theta} h_{refined} \begin{cases} h_{likeliest}, & h_{likeliest} \in ]3,12]\ \text{km} \\ \min(3\ \text{km}, \text{mean}(h_{i \in \{i|d_i < 1.6\ \text{km}\}})), & h_{likeliest} \leq 3\ \text{km} \wedge \text{mean}(h_{i \in \{i|d_i < 1.6\ \text{km}\}}) > 1.5\ \text{km} \\ \min(1.5\ \text{km}, \text{mean}(h_{i \in \{i|d_i < 1.2\ \text{km}\}})), & h_{likeliest} \leq 3\ \text{km} \wedge \text{mean}(h_{i \in \{i|d_i < 1.6\ \text{km}\}}) \leq 1.5\ \text{km}. \end{cases} \tag{7}$$

**Specific comment 17**

Reviewer:

Lines 350-354: The uniformity constraint is very reasonable during algorithm training, not so during evaluation! It is very interesting to evaluate the algorithm in variable cases and understand what the outputs are, if it is biased towards the low or high clouds etc.

Authors' response:

First of all, we would like to apologize as a statement in the manuscript was misleading. The filter excluding variable situations is applied in the modelling of conditional probabilities (now Sect. 3.4) and in Sect. 4.4 to compare performance metrics from ASI-pairs and ASI network. In Sect. 4.1-4.3 this filter is not applied. We now corrected this statement and moved it from Sect. 4.1 to Sect. 4.4.

The scatter-density plots shown in Sect. 4.2 may provide insights regarding effects occurring in variable cases. Based on Reviewer Comment 1, minor comment 7 we also added performance metrics to these plots. To enable the reader to evaluate the performance of the ASI-based estimation of CBH under these conditions (e.g. concerning biases) more quickly, we also added percentiles to all scatter-density plots.

Changes in manuscript:

pp. 20-21, ll. 531-535:

530    from 30 June 2019 to 27 September 2019. This dataset was excluded from the model development described in Sect. 3. The
       analyzed quantity is 10 min-median CBH.

535

p. 28-29, ll. 691-695

       The statistical evaluations are now restricted to times in which the variability of CBH is small. More precisely, the standard
       deviation of CBH within a window 15 min before and after the analyzed time is required to be less than 30% of mean CBH

       within the same window. As discussed above, the ASI-pairs and the ASI network are expected to measure a spatial median
       CBH whereas the ceilometer measures CBH at the point of its installation. This restriction aims to assure a good comparability
695    of both measurements. Further, this way our results are more comparable to a prior study by Kuhn et al. (2019).
           Accuracies of CBH measurement by ASI-pairs and ASI network are analyzed separately for five ranges of reference CBH

p. 24, Fig. 8:

[Figure]

**Figure 8.** Relative frequency of ASI-based CBH estimation for given CBH from ceilometer—. Evaluation for two of the ASI-pairs DON-MAR (upper row, left) and UOL-HOL (upper row, right) with respective camera distances of 0.8 and 5.7 km, and from the ASI network without (bottom row, left) and with refinements (bottom row, right). Relative frequency in each column adds up to 1. Additionally, median (50%-quartile, red dotted), limits to the interquartile range (IQR, red dashed) and 5−,95−percentiles (red solid line) based on floating 1000 m-bins of CBH from ceilometer are plotted.

**Specific comment 18**

Reviewer:

Line 360: Why not reverse the two plots in Figure 6, to discuss them in order?

Authors' response:

We appreciate the suggestion. However, as suggested by Major Comment 2, to shorten Sect. 4.1, this figure and related descriptions have been moved to Appendix B.

Changes in manuscript:

p. 21, Fig. 6:

[Figure]

**Figure 6.** Time series of cloud base height for  an exemplary  day (02 September 2019) measured by 42 ASI-pairs (grey filled), by two exemplary ASI-pairs DON-MAR and CLO-FLE with respective camera distances 0.8 and 4.2 km, by the ASI network with refinements and by a ceilometer in the urban area of Oldenburg.

p. 21, ll. 544-545:

**4.1 Comparison of CBH measurements for  an exemplary day**

 We first analyze the properties of the different procedures to measure CBH based on exemplary situations. 545 Fig. 6 visualizes time series of CBH for a variable day (02 September 2019) measured by ceilometer, by all available ASI-

p. 22, Fig. 7:

[Figure]

**Figure 7.** Sky images taken by ASI UOL representing  a multi-cloud-layer situation on  02 September 2019 7:20 (left) and an almost clear-sky situation on 02 September 2019 17:00 (right) respectively.

p. 22, l. 553:

fraction of the sky in the urban area (compare Fig. 7, left). The CBH estimation approach tends to react stronger to clouds in this area of the sky in which contrasts are typically pronounced. Around 10:20 a multilayer situation is present. In the 555 whole sky dome cumulus clouds are visible but a large fraction of the cloud cover is made up by the cirrus layer. Around this

p. 23, ll. 572-581:

575 ~~CBH of these varies in the range of 7...11 km according to the ceilometer. The range of CBH from ASI-pairs reflects this spread. Still, it is not obvious which of the ASI-pair based observations would be the most appropriate. From the ASI network a rather steady CBH estimation results which most of the time reflects the dominant CBH layer as recognized by the ceilometer. The combined estimation misses physically meaningful variations of CBH typically towards higher values recognized by the ceilometer. Also for this day time series of CBH and corresponding ASI images were compared. Again large underestimations~~

580

p. 23, ll. 588-589:

 To give further insight, in Appendix B2, timeseries of CBH from the different sources are compared for another exemplary day.

590 **4.2 Comparison of CBH measurements by relative frequencies**

Appendix B, pp. 38-39, ll. 943-953:

**B2 Comparison of CBH measurements for another exemplary day**

Figure B2 shows CBH on 06 August 2019 again measured by ceilometer, by all available ASI-pairs and by the ASI network.

945 This day, similar to 02 September 2019, discussed previously, includes multi-layer conditions with high layers overlaid by low layers, resulting in similar observations. In the morning and evening high cloud layers are dominant. The CBH of these varies in the range of 7...11 km according to the ceilometer. The range of CBH from ASI-pairs reflects this spread. Still, it is not obvious which of the ASI-pair based observations would be the most appropriate. From the ASI network a rather steady CBH estimation results which most of the time reflects the dominant CBH layer as recognized by the ceilometer. The combined

950 estimation misses physically meaningful variations of CBH typically towards higher values recognized by the ceilometer. Also for this day time series of CBH and corresponding ASI images were compared. Again large underestimations of CBH by the ASI network (at 05:30, 08:15, 10:00, 12:30, 16:00) were traced back to the ASI-based estimations responding stronger to lower optically denser low cloud layers which pass the vicinity of the urban area (compare Fig. B3).

[Figure]

**Figure B2.** Time series of cloud base height for an exemplary day (06 August 2019) measured by 42 ASI-pairs (grey filled), by two exemplary ASI-pairs DON-MAR and CLO-FLE with respective camera distances 0.8 and 4.2 km, by the ASI network with refinements and by a ceilometer in the urban area of Oldenburg.

[Figure]

**Figure B3.** Sky image taken by ASI UOL representing a multi-cloudlayer situation on 06 August 2019 12:35

**Specific comment 19**

Reviewer:

Line 377: As shown from the two pairs, in cloud-free conditions some ASI-pairs output the value of 12km, while others 2km (probably due to local low clouds). Why do you suggest that the 4km output of the network is a reasonable prediction of a layer coming at least 30 minutes later? Is this layer captured by any pair in the network? It could also be a lucky combination of these two extreme values? In general, how does the network handle cloud-free conditions?

Authors' response:

Our description may not have been precise in this point. We now added a plot (Fig. B1) in the appendix which shows the measurements of CBH from the ASI-pairs and from the ASI network as well as from the ceilometer during this clear period in more detail. We also added a short passage in Sect. 4.1 to describe closer which period we referred to. From Fig. B1 it is visible that the ASI-pairs measure a broad range of values between the extreme values of 2 km and 12 km, before around

17:00. Most ASI-pairs measure an intermediate CBH. After 17:00 the spread between the measurements of the ASI-pairs reduces. From around 17:05 the ASI network and some of the ASI-pairs measure a CBH of around 3 km. This CBH (3.1 km) is later also measured by the ceilometer. During this period the approaching cloud layer may be detected before its arrival in the urban area.

During very clear periods, the ASI network is likely to return a CBH which is very large, in the range of 10 km. For an application this is not problematic, in our opinion, because another image processing step is used which is able to detect the absence of clouds. We added a short explanation on this.

Changes in manuscript:

p. 22, ll. 564-571:

Around 17:00, a nearly clear sky is visible (compare Fig. 7, right). Consequently, the ceilometer does not provide any valid CBH. The ASI-pairs provide a CBH that scatters over a wide range, while the ASI network provides a CBH that is assumed to be
565 reasonable. The an intermediate CBH. A similar reading of CBH is also recognized by a fraction of the ASI-pairs. From around 17:05, the ASI network detects a CBH of 3 km. With 3.1 km, the following CBH measurements of the ceilometer around 17:30 25 confirm the suggested CBH of the approaching cloud layer -(see Fig. B1 for a detail view of the CBH measurements during this almost clear sky period). This situation reflects the expected behavior of the ASI network under mostly clear conditions. However, for a completely clear sky, the ASI network partly produces invalid readings (NaN) and partly it detects a large CBH
570 of around 10 km. In this case, a consecutive image processing step detects the absence of clouds. This step is not part of the present study.

Appendix B, p. 38, ll. 940-942, Fig B1:

[Figure]

**Figure B1.** Detail view of CBH measured by ASI-pairs (grey dots), by the ASI network (blue triangles) and ceilometer (red circles) during a period with low sky coverage. Around 17:00 approaching clouds are viewed close to the horizon by all ASIs.

Figure B1 provides a detail view of CBH measured by ASI-pairs, by the ASI network and by the ceilometer during a mostly clear period on 02 September 2019. The period is discussed in Sect. 4.1.

**Specific comment 20**

Reviewer:

Line 395: The main ASI-based CBH retrieval limits the instrument to a maximum zenith angle of 67 degrees. For the CLO-FLE pair, given the 4.2km distance of the instruments, the minimum detectable clouds should be around 1.4 km (if I calculate correctly).  In the September cases many clouds are below this limit, so probably the second or third sub-algorithm was used (using e.g. the complete FOV of the camera). Could this be the reason of the overestimation? If yes, does the full-FOV retrieval add anything to the estimate or could just be skipped?

Authors' response:

We share the reviewer's opinion, that the behavior seen for CLO-FLE in situations with CBH much smaller than 2 km is connected to the minimum CBH which this system can detect. This minimum CBH may indeed be determined by the sub-algorithm relying on the main ASI's cropped orthogonal image. The usage of the full FOV to retrieve CBH is not expected to improve an ASI-pairs capability to

detect very low clouds noticeably. We now pointed out in Sect. 3.1, that this sub-algorithm is mainly intended to increase the robustness of the method. It may yield a valid measurement in some cases when the first sub-algorithms failed. We condensed the discussion of the minimum CBH in Sect. 4.2 as shown below. See also our response to Specific Comment 22.

Changes in manuscript:

p. 10, ll. 236-242:

> window for which the estimation failed. In cases with no valid measurement in any of the windows, the orthogonal images of both ASIs are evaluated up to a maximum zenith angle of 77.8° (measured at the center of each image side, green shapes in Fig. 3, central red dotted box)from ). These orthoimages from both cameras are matched in the cross-correlation and the ASI-pair returns a uniform CBH. This second step can yield a valid measurement of CBH in cases when only few clouds are
> 240 present to be matched. This step mainly intends to increase the robustness of the CBH measurement. This step is not expected to increase the capability of an ASI-pair to detect very low clouds in relation to the camera distance, as the window size used in this step is very large.

p. 10-11, ll. 252-265:

> If the estimation of CBH fails for Only based on geometry and the evaluated image areas, this central window , we use the CBH that is measured by matching the peripheral windows (Fig. 3, peripheral red dotted boxes) of the same orthoimage with the orthoimage of the second camera. These peripheral windows of an orthoimage have the same shape as the central window
> 255 (see could provide readings down to a minimum CBH of $0.25 \times d$. Where $d$ is the camera distance. However, under such extreme conditions the matching procedure may fail very frequently. The central peripheral windows, shown in Fig. 3, center, peripheral red dotted boxes). If a valid estimation of CBH is received for multiple peripheral windows, we use the average CBHfrom these windows.
> For cases with still no valid measurement, images of both cameras are evaluated up to a maximum zenith angle of approximately
> 260 cover zenith angles 38.1..67°. The matched area from the auxiliary ASI's orthogonal image has identical shape and can cover a zenith angle up to 77.8°(measured at . Based on this, we estimate the minimum CBH, which an ASI-pair can measure, to be $0.18 \times d$. However, from our experience, a large fraction of clouds observed at zenith angles larger than 67° are not matched successfully between the ASIs and typically rejected. If the matching procedure could only be successful, if also the window of
>
> the second ASI included zenith angles not larger than 67°, then CBH could be measured down to $0.32 \times d$ using the peripheral
> 265 windows and $0.64 \times d$ using the central window.

p. 25 ll. 614-632:

Qualitatively, the effects seen meet the expectation from the literature (Nouri et al., 2019a; Kuhn et al., 2019; Nguyen and Kleis

615 . ASI-pairs with large camera distance are expected to be more accurate when measuring the CBH of high clouds. On the other hand, ASI-pairs with large camera distance are expected to be less accurate for small CBH values and are expected to exhibit a larger *minimum CBH*, below which no physically meaningful readings are received. From the geometric considerations in Sect. 3.1, a minimum CBH of about $0.18 \times d$ was expected. Where $d$ is the camera distance. For UOL-HOL,  a significantly

620 larger minimum CBH of about 2 km is evident. If reference CBH is smaller than 2 km, the ASI-pair yields measurements of CBH which scatter ~~randomly around a modus value of 3.8 km for reference CBH < 1.8 km. If reference CBH ranges between 1.8...3 km, this behavior is still observed for a significant part of the readings. For UOL-HOL nearly no reading of less than 1.5 km is recognized. In general, strong scattering is seen for this ASI-pair. However, towards large values of reference CBH the measurement appears to scatter to a smaller extent and especially for very large CBH (> 8 km) a satisfying agreement of CBH~~

625  median value of 4 km. This behavior can be explained as the matching procedure fails if pattern are matched which are located at a larger zenith angle than a maximum value. Consequently, random features observed under a zenith angle smaller than the maximum value are often matched erroneously which yields a too large estimation of CBH. Similarly for DON-MAR a minimum CBH of around 0.3 km is suggested.

 Overall, the ASI-pairs are characterized by a

630 minimum CBH in the range of $0.32 \times d$. As described above, this suggests that the matching procedure of the ASI-pairs almost always fails if matched windows cover zenith angles larger than 67°. Further, also for reference CBH close to this minimum CBH, the ASI-pairs yield increased deviations, e.g. below 0.5 km and 3 km for DON-MAR and UOL-HOL.

**Specific comment 21**

Reviewer:

Line 405: What I understand from the plot is that the low clouds are detected by the ceilometer and not by the ASI-pair, not the other way around. If this is true, the ceilometer site should have persistent low not present over the ASI-pair. Is this reasonable from the local meteorological conditions? What seems more reasonable is that ASI-pair cannot detect low clouds, e.g. due to geometric and algorithm considerations. Please provide more details.

Authors' response:

We assume that the reviewer refers to the areas on the far left of the scatter-density plots (e.g. reference CBH < 0.5 km for DON-MAR and reference CBH < 2 km for UOL-HOL) and we agree with the analysis of the reviewer. At this point, we intended to discuss another area of these plots and now indicated these areas more precisely in the manuscript. When reference CBH ranges around 3…12 km, the ASI-based systems frequently detect low clouds close to the 5-percentile line, i.e. far below the main diagonal of the plot. In these cases, the ASI-based systems provide a CBH which is too small. As described in previous sections, we expect that in these cases the ASI-based systems recognize low clouds present in their field of view. At the same time there might be a gap in the low cloud layer at the location of the ceilometer. Therefore, the ceilometer may recognize a larger CBH.

Changes in manuscript:

pp. 24-25, ll. 605-610, Fig. 8:

[Figure]

**Figure 8.** Relative frequency of ASI-based CBH estimation for given CBH from ceilometer~. Evaluation for two of the ASI-pairs DON-MAR (upper row, left) and UOL-HOL (upper row, right) with respective camera distances of 0.8 and 5.7 km, and from the ASI network without (bottom row, left) and with refinements (bottom row, right). Relative frequency in each column adds up to 1. Additionally, median (50%-quartile, red dotted), limits to the interquartile range (IQR, red dashed) and 5−,95−percentiles (red solid line) based on floating 1000 m-bins of CBH from ceilometer are plotted.

605     Both ASI-pairs exhibit a strong scattering of the measurements, clearly visible from the wide spread of the quartiles as well as of the 5−,95−percentiles. In agreement with the prior finding, DON-MAR is rather precise at low CBH ($\leq 3$ km), whereas UOL-HOL is notably more precise at greater CBH. CBH from the ASI-pairs often deviates towards

 low CBH, when the ceilometer measures CBH in the range 3...12 km. In this range, the 5-percentile of ASI-based CBH increases only slightly with reference CBH and comparably large
610     relative frequencies are found close to the 5-percentile. As discussed  in Sect. 4.1, this can  result from low cloud layers which are actually present in the ASI-pairs' field of view but not at the ceilometer's location.

**Specific comment 22**
Reviewer:

Line 408-410:  This doesn't sound very surprising since the minimum altitude where your ASI have overlapping images at 67deg FOV should be around 1.7km. Please discuss such issues, preferable in a previous section, before presenting the results.

Authors' response:

As suggested by Specific Comment 11, we now calculated the minimum CBH which may be related to the sub-algorithms in Sect. 3.1. We reworked the discussion of minimum CBH in Sect. 4.2. and adapted it to refer to these values of minimum CBH expected from geometry.

Changes in manuscript:

p. 10, ll. 236-242:

window for which the estimation failed. In cases with no valid measurement in any of the windows, the orthogonal images of both ASIs are evaluated up to a maximum zenith angle of 77.8° (measured at the center of each image side, green shapes in Fig. 3, central red dotted box)from ). These orthoimages from both cameras are matched in the cross-correlation and the ASI-pair returns a uniform CBH. This second step can yield a valid measurement of CBH in cases when only few clouds are present to be matched. This step mainly intends to increase the robustness of the CBH measurement. This step is not expected to increase the capability of an ASI-pair to detect very low clouds in relation to the camera distance, as the window size used in this step is very large.

p. 10-11, ll. 252-265:

If the estimation of CBH fails for Only based on geometry and the evaluated image areas, this central window , we use the CBH that is measured by matching the peripheral windows (Fig. 3, peripheral red dotted boxes) of the same orthoimage with the orthoimage of the second camera. These peripheral windowsof an orthoimage have the same shape as the central window (see could provide readings down to a minimum CBH of $0.25 \times d$. Where $d$ is the camera distance. However, under such extreme conditions the matching procedure may fail very frequently. The central peripheral windows, shown in Fig. 3, center, peripheral red dotted boxes). If a valid estimation of CBH is received for multiple peripheral windows, we use the average CBHfrom these windows.

For cases with still no valid measurement, images of both cameras are evaluated up to a maximum zenith angle ofapproximately cover zenith angles 38.1..67°. The matched area from the auxiliary ASI's orthogonal image has identical shape and can cover a zenith angle up to 77.8°(measured at . Based on this, we estimate the minimum CBH, which an ASI-pair can measure, to be $0.18 \times d$. However, from our experience, a large fraction of clouds observed at zenith angles larger than 67° are not matched successfully between the ASIs and typically rejected. If the matching procedure could only be successful, if also the window of the second ASI included zenith angles not larger than 67°, then CBH could be measured down to $0.32 \times d$ using the peripheral windows and $0.64 \times d$ using the central window.

p. 25 ll. 614-632:

Qualitatively, the effects seen meet the expectation from the literature (Nouri et al., 2019a; Kuhn et al., 2019; Nguyen and Kleis
615  . ASI-pairs with large camera distance are expected to be more accurate when measuring the CBH of high clouds. On the other
hand, ASI-pairs with large camera distance are expected to be less accurate for small CBH values and are expected to exhibit
a larger *minimum CBH*, below which no physically meaningful readings are received. From the geometric considerations in
Sect. 3.1, a minimum CBH of about $0.18 \times d$ was expected. Where $d$ is the camera distance. For UOL-HOL,
 a significantly
620  larger minimum CBH of about 2 km is evident. If reference CBH is smaller than 2 km, the ASI-pair yields measurements of
CBH which scatter

625   median value of 4 km. This behavior can be explained as the matching procedure fails if
pattern are matched which are located at a larger zenith angle than a maximum value. Consequently, random features observed
under a zenith angle smaller than the maximum value are often matched erroneously which yields a too large estimation of
CBH. Similarly for DON-MAR a minimum CBH of around 0.3 km is suggested.

 Overall, the ASI-pairs are characterized by a
630  minimum CBH in the range of $0.32 \times d$. As described above, this suggests that the matching procedure of the ASI-pairs almost
always fails if matched windows cover zenith angles larger than 67°. Further, also for reference CBH close to this minimum
CBH, the ASI-pairs yield increased deviations, e.g. below 0.5 km and 3 km for DON-MAR and UOL-HOL.

**Specific comment 23**

Reviewer:

Line 417: "in the dataset used for modelling"?

Authors' response:

As we understand the comment, it is not clear at this point why the "dataset used modelling" is discussed in this context. As also suggested by the following Special Comment 24, we rephrased this passage. We hope this makes the intended statement clearer, also in this perspective.

Changes in manuscript:

p. 26, ll. 642-650:

In the range of reference CBH > 10 km, the ASI network constantly returns CBH of around 10 km. In the studied cli-
mate, reference CBH in this range are comparably
rare
645   (see Fig. 4). Therefore, corresponding grid
cells of the conditional probability distributions, used by the estimation procedure, were approximated coarsely based on a
small number of observations. The ASI network's combination method using cumulative likelihood is intended to avoid de-
viations resulting from these inaccuracies and  thus to yield a more conservative
 estimation. However, this approach also suppresses the estimation of extreme CBH readings, which causes a BIAS under
650  these conditions. For the analyzed site, deviations found in this range of CBH are of minor importance.

**Specific comment 24**

Reviewer:

Line 418-422:  The text is not well written, and it is not clear what you mean.  Please rephrase.

Authors' response:

We rephrased the passage as shown below.

Changes in manuscript:

p. 26, ll. 642-650:

> In the range of reference CBH > 10 km, the ASI network constantly returns CBH of around 10 km. In the studied climate (see Fig. 4) and accordingly in the dataset used for modelling readings of , reference CBH in this range are comparably rare . Therefore, conditional probabilities used in the estimation are modeled inaccurately. The estimation procedure uses cumulative . Compared to the usage of likelihood , this avoids frequent strong (see Fig. 4). Therefore, corresponding grid cells of the conditional probability distributions, used by the estimation procedure, were approximated coarsely based on a small number of observations. The ASI network's combination method using cumulative likelihood is intended to avoid deviations resulting from these inaccuracies and yields thus to yield a more conservative but in this case biased estimation of CBH estimation. However, this approach also suppresses the estimation of extreme CBH readings, which causes a BIAS under these conditions. For the analyzed site, deviations found in this range of CBH are of minor importance. For low values of reference CBH ASI network and ASI-pair DON-MAR both appear to perform similarly at high accuracy. Only for

**Technical comment 1**

Reviewer:

Lines 42-54: As written now, the paragraph starts as if to present ceilometers but ends up presenting various CBH estimation techniques and ends up with ASI-based forecasting requirements. A slight editing is needed to make the text clearer.

Authors' response:

We rewrote this paragraph in part to put a stronger focus on possible sources of CBH to be considered for nowcasting. We moved this specification of nowcasts up.

Changes in manuscript:

p. 2, ll. 34-35:

The method to measure CBH, presented in this study, is used as part of an ASI-based nowcasting system of the solar resource. ASI-based nowcasting is typically applied if variations of irradiance have to be predicted for lead times immediately ahead

35 (0...20 min) and at highest temporal and spatial resolution (e.g. 30 s and 5 m respectively as used by Nouri et al., 2020b). Such nowcasts can reduce the uncertainty of supply from solar power plants and can support efficient balancing of energy supply

p. 2-3, ll. 45-58:

45      CBH, required in ASI-based nowcasting, can be estimated in multiple ways. Most commonly, CBH is measured by ceilometers or other LiDARs. In Germany, the meteorological service Deutscher Wetterdienst (DWD) operates a network of ceilometers which has a distance between stations of approximately 60 km in the region of the measurement site Oldenburg (Chan et al., 2018). Ceilometers are specialized instruments that come at a high price and provide CBH zenith-wise for the location of their installation. Therefore, we do not consider ceilometers as an option to provide CBH in real time for most

50   solar power plants or cities with many roof top installations. Further  common approaches to measure CBH, which could be applied for operational use in nowcasting, include weather balloons and the estimation of CBH based on a recognized cloud genus (World Meteorological Organization, 2018). Satellites can measure CTH of the highest cloud layer (Hamann et al., 2014) but require estimations of cloud vertical extent (see e.g. Noh et al., 2017) to provide cloud base height (CBH). ASIs can directly measure CBH but require estimations of cloud vertical extent if CTH is of interest.

55  In ASI-based nowcasting, the double use of ASIs for the estimation of CBH besides cloud recognition is considered advantageous in a trade-off between system costs and accuracy.

    ASI-based estimation of CBH may follow different principles. Some approaches first measure the angular velocity of clouds

60   in the sky-image of a single ASI and estimate CBH with an external source of cloud velocity. Wang et al. (2016) derives cloud velocity by three photocells placed at known distances from each other. Kuhn et al. (2018b) measures cloud velocity by a cloud speed sensor based on nine photocells and by a shadow camera system and compares the accuracy of received CBH. Tomographic reconstruction approaches (Mejia et al., 2018) or similarly voxel carving approaches (Nouri et al., 2018) first model 3-dimensional representations of clouds from which their base height can be retrieved.

**Technical comment 2**

Reviewer:

Line 71: Better use "Most ASI-based monitoring systems..." or similar.

Authors' response:

We adapted ASI system to ASI-based nowcasting system

Changes in manuscript:

p. 3, l. 79:

80 Most  ASI-based nowcasting systems described in the literature feature one (Schmidt et al., 2016), two (Allmen and Kegelmeyer Jr, 1996; Beekmans et al., 2016; Blanc et al., 2017; Savoy et al., 2016) or three (Peng et al., 2015) ASIs. Four

**Technical comment 3**

Reviewer:

Line 202: "For example, Tabernas,..."

Authors' response:

We inserted accordingly.

Changes in manuscript:

p. 12, l. 302:

300 , as we calculated from the TanDEM-X elevation model (Wessel et al., 2018). The flat topography is expected to support a temporally and spatially low variability of CBH within cloud layers. For other sites, a focus on measuring CBH for every cloud object is of higher priority. For example, Tabernas, the site studied by Nouri et al. (2019a), features a cold-arid steppe climate (BSk according to Kottek et al., 2006) and is surrounded by mountains with elevations up to 2168 m over sea level within a radius of 25 km. As shown by (Nouri et al., 2019c), CBH at the site is distributed almost uniform in the range 0...11 km. These

**Technical comment 4**

Reviewer:

Line 355: "Then, the coincidence,...". The sentence needs rewording.

Authors' response:

We reformulated as shown below.

Changes in manuscript:

p. 21, ll. 537-539:

First, characteristics of CBH-measurements from the ASI network and from individual ASI-pairs are compared to the CBH-measurement of the reference ceilometer based on insightful days. Then, the  measurements of CBH by ASI network and ASI-pairs  are compared to the one of the ceilometer by scatter-density plots. Subsequently,

540 the accuracy of an ASI-pair and of the ASI network are analyzed for the application